# Direct Preference-Based Evolutionary Multi-Objective Optimization with Dueling Bandits

**Tian Huang**[1][†]**, Shengbo Wang**[1][†]**, Ke Li**[2][∗]
[1] School of Computer Science and Engineering ,
University of Electronic Science and Technology of China
[2] Department of Computer Science, University of Exeter
tianhuang.uestc@gmail.com   shnbo.wang@foxmail.com   k.li@exeter.ac.uk

## Abstract

The ultimate goal of multi-objective optimization (MO) is to assist human decision-makers (DMs) in identifying solutions of interest (SOI) that optimally reconcile multiple objectives according to their preferences. Preference-based evolutionary MO (PBEMO) has emerged as a promising framework that progressively approximates SOI by involving human in the *optimization-cum-decision-making* process. Yet, current PBEMO approaches are prone to be inefficient and misaligned with the DM's true aspirations, especially when inadvertently exploiting mis-calibrated reward models. This is further exacerbated when considering the stochastic nature of human feedback. This paper proposes a novel framework that navigates MO to SOI by *directly* leveraging human feedback without being restricted by a predefined reward model nor cumbersome model selection. Specifically, we developed a clustering-based stochastic dueling bandits algorithm that strategically scales well to high-dimensional dueling bandits. The learned preferences are then transformed into a unified probabilistic format that can be readily adapted to prevalent EMO algorithms. This also leads to a principled termination criterion that strategically manages human cognitive loads and computational budget. Experiments on 48 benchmark test problems, including the RNA inverse design and protein structure prediction, fully demonstrate the effectiveness of our proposed approach.

## 1 Introduction

Multi-objective optimization (MO) represents a fundamental challenge in artificial intelligence [49], with profound implications that span virtually every sector—from scientific discovery [67] to engineering design [22], and societal governance [23]. In MO, there is no single *utopian solution* that optimizes all objectives; instead, the Pareto front (PF) comprises non-dominated solutions, each representing an efficient yet incomparable trade-off between objectives. The goal of MO is to assist human decision-makers (DMs) in identifying solutions of interest (SOI) that optimally reconcile these conflicting objectives according to their preferences. This field has been a subject of rigorous study within the multi-criterion decision-making (MCDM) community [43] for more than half a century. Over the past two decades, we have witnessed a seamless transition from purely analytical methodologies to a burgeoning interest in interactive evolutionary meta-heuristics, known as preference-based evolutionary MO (PBEMO) [39].

As in Figure 1(**a**), a PBEMO method involves three building blocks. The optimization module uses a population-based meta-heuristics to explore the search space. The preference information is progressively learned by periodically involving human DM in the consultation module to provide

---

[∗]Correspondence: k.li@exeter.ac.uk; [†] Equal contributions.

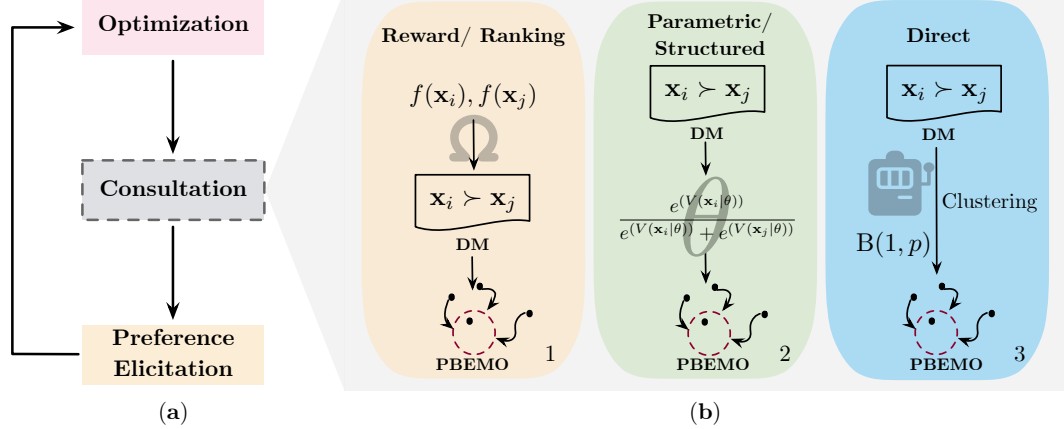

**Figure 1:** (**a**) Flow chart of a conventional PBEMO. (**b**) Conceptual illustration of reward-based, model-based, and direct preference learning strategies.

preference feedback. The learned preference representation is then transformed into the format that guides the evolutionary search progressively towards SOI in the `preference elicitation` module. The overall PBEMO process is DM-oriented and is an *optimization-cum-decision-making* process. While PBEMO has been extensively studied in the literature in the past three decades (e.g., [44, 15, 8, 36, 59, 38]), there are several fundamental issues unsolved, especially in the `consultation` and `preference elicitation` modules, that significantly hamper the further uptake in real-world problem-solving scenarios.

- The `consultation` module serves as the interface for DM interaction with the `optimization` module. It queries the DM about their preferences to effectively recommend solutions. This closely aligns with preference learning in the machine learning community. There are three strategies to accomplish this. The first, as shown in Figure 1(**b**-1), involves reward models [51] or ranking mechanisms [29, 27, 38]. In this approach, DMs often provide scores or rankings for a large set of solutions. However, this overburdens the DM and risks introducing errors into the optimization process due to overlooked human cognitive limitations. The other two strategies are grounded in the dueling bandits settings [66], utilizing pairwise comparisons in consultation but with different assumptions about human feedback. Specifically, the second strategy relies on a parameterized transitivity model [5] or a structured utility function for dueling feedback, such as the Bradley-Terry-Luce model [11] in Figure 1(**b**-2). While theoretically appealing, this method faces a challenge in model selection that can be as complex as the original problem, posing difficulties in practical applications. Different from the previous two strategies, the last one tackles human feedback as stochastic events such as Bernoulli trials [63], and it learns DM's preferences from their feedback as shown in Figure 1(**b**-3). However, a key bottleneck here is the souring number of DM queries required when considering a population of solutions in PBEMO.

- The `preference elicitation` module acts as a catalyst, transforming the preference information learned in the `consultation` module—usually not directly applicable—into a format usable in the underlying EMO algorithm. The stochastic nature of human feedback [1] can lead to a misuse of learned preferences that adversely disturb search processes. This issue is pronounced in PBEMO contexts, where the number of consultations is constrained to reduce the human cognitive burden. Additionally, there is no thumb rule for determining the frequency of DM interactions or terminating such interactions, where current methods are often heuristics (e.g., setting a fixed number of interactions [36]).

In this paper, we propose a novel direct PBEMO framework (dubbed `D-PBEMO`) that directly leverages DM's feedback to guide the evolutionary search for SOI. Note that it neither relies on any reward model nor cumbersome model selection. Our `D-PBEMO` framework consists of two key features.

- Given the stochastic nature of human feedback, we develop a novel clustering-based stochastic dueling bandits algorithm in the `consultation` module. It is model-free and its regret

is $\mathcal{O}(K^2 \log T)$, where $K$ is the number of clusters and $T$ is the number of rounds. This overcomes the challenge of substantial queries inherent in conventional dueling bandits [4].

- The `preference elicitation` module transforms the learned preferences from the `consultation` module into a unified probabilistic format, in which the associated uncertainty represents the stochasticity involved in preference learning. This not only streamlines the incorporation of learned preferences into the `optimization` module to guide EMO to search for SOI, but also constitutes a principled termination criterion that strategically manages human cognitive burden and the computational budget.

## 2 Preliminaries

### 2.1 Multi-Objective Optimization Problem

The MO problem is formulated as: $\min\limits_{\mathbf{x} \in \Omega} \mathbf{F}(\mathbf{x}) = \big(f_1(\mathbf{x}), \ldots, f_m(\mathbf{x})\big)^\top$, where $\mathbf{x} = (x_1, \ldots, x_n)^\top$ is an $n$-dimensional decision vector and $\mathbf{F}(\mathbf{x})$ is an $m$-dimensional objective vector whose $i$-th element is the objective mapping $f_i : \Omega \to \mathbb{R}$, where $\Omega$ is the feasible set in the decision space $\mathbb{R}^n$. Without considering the DM's preference information, given $\mathbf{x}^1, \mathbf{x}^2 \in \Omega$, $\mathbf{x}^1$ is said to dominate $\mathbf{x}^2$ (denoted as $\mathbf{x}^1 \preceq \mathbf{x}^2$) iff $\forall i \in \{1, \ldots, m\}$ we have $f_i(\mathbf{x}^1) \leq f_i(\mathbf{x}^2)$ and $\mathbf{F}(\mathbf{x}^1) \neq \mathbf{F}(\mathbf{x}^2)$. A solution $\mathbf{x} \in \Omega$ is said to be Pareto-optimal iff $\nexists \mathbf{x}' \in \Omega$ such that $\mathbf{x}' \preceq \mathbf{x}$. The set of all Pareto-optimal solutions is called the Pareto-optimal set (PS) and their corresponding objective vectors constitute the PF.

The ultimate goal of MO is to identify the SOI from the PS satisfying DM's preference. It consists of two tasks: ① searching for Pareto-optimal solutions that cover SOI, and ② steering these solutions towards the SOI. PBEMO addresses the task ① by employing an EMO algorithm as a generator of an evolutionary population of non-dominated solutions $\mathcal{S} = \{\mathbf{x}^i\}_{i=1}^N$, striking a balance between convergence and diversity for coverage. For the task ②, PBEMO actively queries DM for preference information regarding these generated solutions, then it leverages the learned preferences to guide the EMO algorithm to approximate the SOI.

### 2.2 Preference Learning as Dueling Bandits

Since human feedback from relative comparisons is considerably more reliable than absolute labels [48], we focus on pairwise comparisons as a form of indirect preference information. In PBEMO, a DM is asked to evaluate pairs of solutions $\langle \mathbf{x}^i, \mathbf{x}^j \rangle$ selected from $\mathcal{S}$, where $i, j \in \{1, \ldots, N\}$ and $i \neq j$. The DM's task is to decide, based on her/his preferences, whether $\mathbf{x}^i$ is better, worse, or equivalent to $\mathbf{x}^j$, denoted as $\mathbf{x}^i \succ_{\mathrm{p}} \mathbf{x}^j$, $\mathbf{x}^i \prec_{\mathrm{p}} \mathbf{x}^j$, or $\mathbf{x}^i \simeq_{\mathrm{p}} \mathbf{x}^j$. Regarding stochastic preference, there is a preference matrix for $K$-armed dueling bandits defined as $\mathrm{P} = [p_{i,j}]_{K \times K}$, where $p_{i,j}$ is the winning probability of the $i$-th arm over the $j$-th arm [65]. In particular, we have $p_{i,j} + p_{j,i} = 1$ with $p_{i,i} = 0.5$. The $i$-th arm is said to be superior to the $j$-th one iff $p_{i,j} > 0.5$. Simply considering each solution as an individual arm will yield $K = N$, which suffers efficiency in targeting the SOI when the evolutionary population is large [35]. The ranking of all arms is determined by their Copeland scores, where the SOI should be the Copeland winners that have the biggest Copeland scores.

**Definition 2.1** ([60]). The normalized Copeland score of the $i$-th arm, $i \in \{1, \ldots, K\}$, is given by:

$$\zeta_i = \frac{1}{K-1} \sum_{j \neq i, j \in \{1, \ldots, K\}} \mathbb{I}(p_{i,j} > 0.5), \tag{1}$$

where $\mathbb{I}(\cdot)$ is an indicator function. Arm $k^\star$ satisfying $k^\star = \underset{i \in \{1, \ldots, K\}}{\operatorname{argmax}} \zeta_i$ is the Copeland winner.

The goal of dueling bandits algorithm is to identify the Copeland winner among all candidate arms with no prior knowledge of $\mathrm{P}$. To this end, a winning matrix is introduced as $\mathrm{B} = [b_{i,j}]_{K \times K}$ to record the pairwise comparison labels, where $b_{i,j}$ is the number of time-slots when the $i$-th arm is preferred from pairs of $i$-th and $j$-th arms. Consequently, we can approximate the preference probability with mean $\tilde{p}_{i,j} = \frac{b_{i,j}}{b_{i,j} + b_{j,i}}$, whose upper confidence bound $u_{i,j}$ and lower confidence bound $l_{i,j}$ can be quantified as:

$$u_{i,j} = \frac{b_{i,j}}{b_{i,j} + b_{j,i}} + \sqrt{\frac{\alpha \log t}{b_{i,j} + b_{j,i}}}, \quad l_{i,j} = \frac{b_{i,j}}{b_{i,j} + b_{j,i}} - \sqrt{\frac{\alpha \log t}{b_{i,j} + b_{j,i}}}, \tag{2}$$

where $\alpha > 0.5$ controls the confidence interval, and $t$ is the total number of comparisons so far. The performance of a dueling bandits algorithm is often evaluated by the regret defined as follows.

**Definition 2.2.** The expected cumulative regret for a dueling bandits algorithm is given as:

$$R_T = \sum_{t=1}^{T} \frac{(\zeta^\star - \zeta'(t)) + (\zeta^\star - \zeta''(t))}{2} = \zeta^\star T - \frac{1}{2} \sum_{t=1}^{T} (\zeta'(t) + \zeta''(t)), \qquad (3)$$

where $T$ is the total number of pairwise comparisons, $\zeta'(t)$ and $\zeta''(t)$ denote the pair to be compared at the $t$-th ($1 \leq t \leq T$) round, $\zeta^\star$ represents the Copeland score of the Copeland winner.

## 3 Proposed Method

Our proposed `D-PBEMO` framework follows the conventional PBEMO flow chart as in Figure 1(**a**). In the following paragraphs, we mainly focus on delineating the design of `D-PBEMO` with regard to both `consultation` and `preference elicitation` modules, while leaving the design of the `optimization` module open.

### 3.1 Consultation Module

As the interface by which the DM interacts with an EMO algorithm, the `consultation` module mainly aims to collect the DM's preference information from their feedback upon $S$ to identify the SOI. We employ the stochastic dueling bandits [74], to directly derive preferences from human feedback without relying on further assumptions such as contextual priors [41] or structured models [11]. In this setting, a natural choice is to consider each candidate solution as an arm to play. However, since the size of $S$ is usually as large as over 100 in the context of EMO, the conventional dueling bandits algorithms will suffer from a large amount of preference comparisons to converge [73, 35]. This is impractical in PBEMO when involving DM in the loop. To address this problem, we propose clustering-based stochastic dueling bandits algorithm that consist of the following three steps.

**Step** 1: **Partition $S$ into $K$ subsets $\{\tilde{S}^i\}_{i=1}^{K}$ based on solution features in the context of EMO.** Such partitioning is implemented as a clustering method based on the Euclidean distances between solutions of $S$ in the objective space. Instead of viewing each solution as an individual arm, we consider each subset $\tilde{S}^i$ as an arm in our proposed dueling bandits. We denote the solution-level preference matrix as $P_s = [p_{i,j}^s]_{N \times N}$, where $p_{i,j}^s$ represents the probability that $\mathbf{x}^i \succ_{\mathrm{p}} \mathbf{x}^j$. Then, the preference matrix $P$ in the subset-level can be calculated by $p_{i,j} = \frac{1}{|\tilde{S}^i||\tilde{S}^j|} \sum_{\mathbf{x}^u \in \tilde{S}^i} \sum_{\mathbf{x}^v \in \tilde{S}^j} p_{u,v}^s$, where $|\tilde{S}^i|$ stands for the size of $\tilde{S}^i$. This probability is well-defined since it satisfies $p_{i,j} + p_{j,i} = 1$. So far, we have reformulated a subset-level dueling bandits problem. Accordingly, the subset-level Copeland winner is the subset $\tilde{S}^\star$ that beats others on average.

**Step** 2: **Subset-level dueling sampling and solution-level pairwise comparisons.** We employ the double Thompson sampling algorithm [62] to determine the subset pairs, and then select solutions from the pairs to query DM preferences. We introduce two vectors $\mathbf{v} = (v_1, \ldots, v_N)^\top$ and $\boldsymbol{\ell} = (\ell_1, \ldots, \ell_N)^\top$ to record the winning and losing times of each solution respectively, initialized by $v_i = 0, \ell_i = 0, i = \{1, \ldots, N\}$. We perform the following steps within a given budget $T$.

**Step** 2.1: **Determine the subset $\tilde{S}'$ that most likely covers the SOI.** We first narrow candidates to the subsets having the highest upper confidence Copeland scores, denoted as $\mathcal{C}^1 = \{\tilde{S}^i | i = \mathrm{argmax}_i \tilde{\zeta}_i\}$, where $\tilde{\zeta}_i = \frac{1}{K-1} \sum_{j \neq i} \mathbb{I}(u_{i,j} > 0.5)$, $i, j \in \{1, \ldots, K\}$. Then, $\forall \tilde{S}^i \in \mathcal{C}^1$, we apply Thompson sampling as $\theta_{i,j}^{(1)} \sim \mathrm{Beta}(b_{i,j} + 1, b_{j,i} + 1)$ to sample the winning probability of $\tilde{S}^i$ over other subsets $\tilde{S}^j$, where $j \in \{1, \ldots, K\}$ and $j \neq i$. Finally, we apply the majority voting strategy to determine the candidate by $\tilde{S}' \leftarrow \mathrm{argmax}_{\tilde{S}^i \in \mathcal{C}^1} \sum_{j \neq i} \mathbb{I}(\theta_{i,j}^{(1)} > 0.5)$, where ties are broken randomly.

**Step** 2.2: **Select the subset $\tilde{S}''$ that can be potentially preferred over $\tilde{S}'$.** To promote exploration, we narrow candidates to the subsets whose lower-confident winning probability over $\tilde{S}'$ is at most 0.5, denoted as $\mathcal{C}^2 = \{\tilde{S}^i | l_{i,\prime} \leq 0.5\}$. Note that $\tilde{S}' \in \mathcal{C}^2$ because $l_{\prime,\prime} \leq p_{\prime,\prime} = 0.5$.

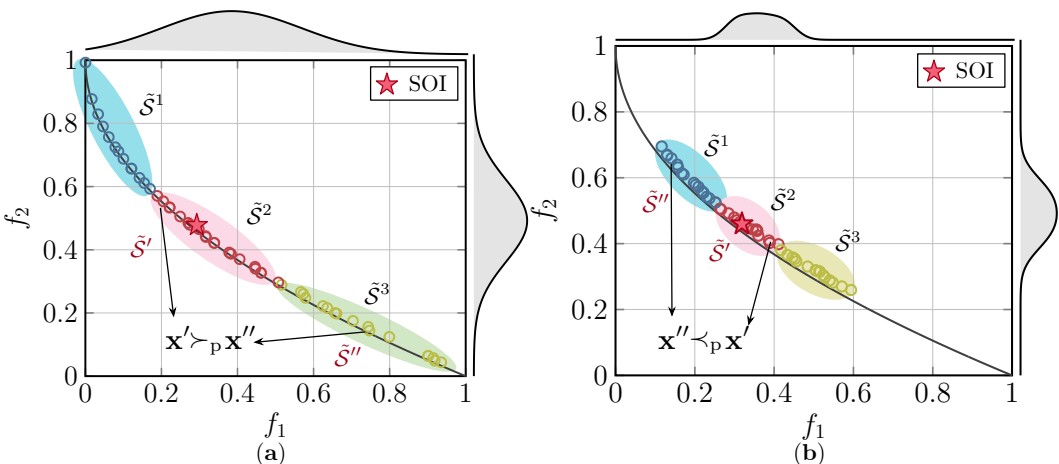

**Figure 2: (a)** The evolutionary population of an EMO algorithm is divided into three subsets, where $\tilde{\mathcal{S}}^2$ covers the SOI (denoted as a ⋆). **(b)** After a PBEMO round, in the next `consultation` session, all solutions are steered towards the SOI and their spreads become more tightened towards the SOI.

Then, $\forall \tilde{\mathcal{S}}^i \in \mathcal{C}^2$, we apply Thompson sampling as $\theta_{i,\prime}^{(2)} \sim \text{Beta}(b_{i,\prime}+1, b_{\prime,i}+1)$ to sample the winning probability of $\tilde{\mathcal{S}}^i$ over $\tilde{\mathcal{S}}'$, and fix $\theta_{\prime,\prime}^{(2)} = 0.5$ according to the definition of preference matrix. Finally, the candidate is determined by $\tilde{\mathcal{S}}'' \leftarrow \text{argmax}_{\tilde{\mathcal{S}}^i \in \mathcal{C}^2} \theta_{i,\prime}^{(2)}$.

**Step** 2.3: **Select two representative solutions $\mathbf{x}' \in \tilde{\mathcal{S}}'$ and $\mathbf{x}'' \in \tilde{\mathcal{S}}''$ to query DM.** We conduct uniform sampling to obtain solutions from the *least-informative* perspective. The DM is asked to evaluate the pair of solutions $\langle \mathbf{x}', \mathbf{x}'' \rangle$. If we observe $\mathbf{x}' \succ_{\mathrm{p}} \mathbf{x}''$, we update $b_{\prime,\prime\prime} \leftarrow b_{\prime,\prime\prime} + 1$, $v_{\prime} \leftarrow v_{\prime} + 1$, and $\ell_{\prime\prime} \leftarrow \ell_{\prime\prime} + 1$, and vice versa. Note that other strategies to obtain solutions $\mathbf{x}'$ and $\mathbf{x}''$ can be used to improve the query efficiency.

**Step** 3: **Output the learned preferences.** The output is a triplet $\{\tilde{\mathcal{S}}^\star, \mathbf{v}, \boldsymbol{\ell}\}$, where $\tilde{\mathcal{S}}^\star = \text{argmax}_{\tilde{\mathcal{S}}^i} \tilde{\zeta}_i$ is the optimal subset that most likely covers SOI, and candidate solutions are considered to be the SOI with uncertainty encoded by their winning and losing times.

The pseudo codes of the above algorithmic implementation are detailed in Appendix C. Figure 2 gives an illustrative example of the consultation process.

**Remark 1.** *PBEMO is a optimization-cum-decision-making process. Instead of having a set of Pareto-optimal candidate solutions upfront, PBEMO starts with a coarse-grained representation of the PF. Then, it gradually steers incumbent solutions towards the learned SOI, which may be inaccurate initially. Subsequent consultations then serve as a refinement process. In this context, different from the dueling bandits, which are designed for identifying the single best solution in each round, our proposed method aims to recognize the SOI with a progressively refined fidelity.*

**Remark 2.** *Based on the Remark 1, we intend to explore the dependency among solutions [47, 53]. This involves performing pairwise comparisons for solution groups, rather than for a single candidate, to enable more efficient interaction. Consequently, in the preference elicitation stage, it becomes essential not only to rely on the learned preference but also to consider the uncertainty introduced by the coarse-grained representation.*

**Theoretical Analysis.** We present a rigorous analysis of the regret bound of our proposed algorithm. To this end, we introduce the following two assumptions derived from the current literature.

**Assumption 3.1** ([62]). The winning probability between two arms satisfies $p_{i,j} \neq 0.5, \forall i \neq j$.

**Assumption 3.2** (Tight clustering [33]). All solutions in $\tilde{\mathcal{S}}^\star$ are the Copeland winners over other solutions in the sub-optimal subsets.

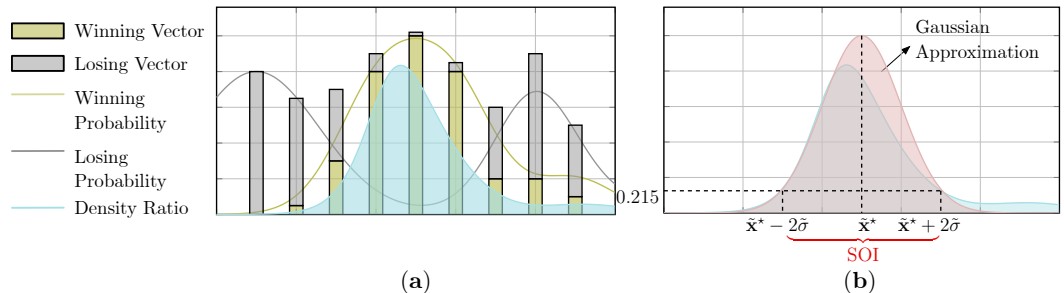

(a)                                           (b)

**Figure 3:** The density ratio between $p_\nu(\tilde{\mathbf{x}})$ and $p_\ell(\tilde{\mathbf{x}})$ is shaded in blue, while its estimation is shaded in red. The SOI falls within the estimated Gaussian distribution for $95\%$ confidence interval.

**Theorem 3.3.** *Under the Assumptions 3.1 and 3.2, for any $\epsilon \in (0,1]$ and $\alpha > 0.5$, the regret of our clustering-based stochastic dueling bandits algorithm is bounded by:*

$$R_T(T) \leq \sum_{i \neq j,\ \underline{p}_{i,j} < 0.5} \left( (1+\epsilon) \frac{\log T}{D_{\mathrm{KL}}(\underline{p}_{i,j} \parallel 0.5)} + \frac{4\alpha \log T}{(\underline{p}_{i,j} - 0.5)^2} \right) + \mathcal{O}\left( \frac{K^2}{\epsilon^2} \right).$$

Definitions of $\underline{p}_{i,j}$ and proof can be found in Appendix B.1.

**Remark 3.** *Theorem 3.3 reveals that the expected regret of our algorithm is bounded by $\mathcal{O}(K^2 \log T)$. Compared with the dueling bandits algorithms based on Thompson sampling and their extensions to large arms [62, 35], whose regret is bounded by $\mathcal{O}(N^2 \log T)$, our proposed clustering-based stochastic dueling bandits algorithm is more efficient in searching for the SOI, given $K \ll N$.*

### 3.2 Preference Elicitation Module

The `preference elicitation` module plays as a bridge that connects consultation and optimization. It transforms the preference learned from the `consultation` module into the configurations used in the underlying EMO algorithm, thus steering the EMO process progressively towards the SOI. In addition, this module maintains the preference information to inform a strategic termination criterion of consultation, thus reduces DM's workload. We present the design of this module by addressing the following three questions.

**How to leverage the preference learned from the consultation session?** Let us assume the DM's feedback collected in the `consultation` session is drawn from a preference distribution as the density ratio $p_v(\tilde{\mathbf{x}})/p_\ell(\tilde{\mathbf{x}})$, where $p_v(\tilde{\mathbf{x}})$ and $p_\ell(\tilde{\mathbf{x}})$ is respectively the winning and losing probability of a solution $\tilde{\mathbf{x}}$ sampled from the PS, see an illustrative example in Figure 3(**a**). After the $\tau$-th round of the `consultation` session, we perform density-ratio estimation based on $\mathbf{v}$ and $\boldsymbol{\ell}$. By using moment matching techniques (see Appendix B.2), we obtain a Gaussian distribution with mean $\tilde{\mathbf{x}}_\tau^*$ and covariance $\tilde{\Sigma}_\tau$. Let $\tilde{\sigma}_\tau$ take the largest value of diagonal elements of $\tilde{\Sigma}_\tau$. The `preference elicitation` module maintains a Gaussian mixture distribution by a convex combination of Gaussian distributions from multiple consultation sessions:

$$\widetilde{\Pr}(\mathbf{x}) = \frac{1}{Z} \sum_{\tau=1}^{N_{\mathrm{consult}}} \frac{1}{\tilde{\sigma}_\tau} \mathcal{N}(\mathbf{x} \mid \tilde{\mathbf{x}}_\tau^*, \tilde{\Sigma}_\tau), \tag{4}$$

where $\mathbf{x} \in \Omega$, $N_{\mathrm{consult}}$ is the total number of consultation sessions conducted so far, and $Z = \sum_{\tau=1}^{N_{\mathrm{consult}}} 1/\tilde{\sigma}_\tau$ is the normalization term. Figure 3(**b**) gives an illustrative example of a Gaussian distribution approximated by the DM's feedback collected at one consultation session.

**How to adapt $\widetilde{\Pr}(\mathbf{x})$ to EMO algorithms?** We believe $\widetilde{\Pr}(\mathbf{x})$ can be applied in any EMO algorithms with few adaptation in their environmental selection. For proof-of-concept purpose, this paper takes NSGA-II [14] and MOEA/D [68], two most popular algorithms in the EMO literature, as examples and we design two D-PBEMO instances, dubbed `D-PBNSGA-II` and `D-PBMOEA/D`.

- At each generation of the original NSGA-II, it first uses non-dominated sorting to divide the combination of parents and offspring into several non-domination fronts $F_1, \ldots, F_l$. Starting from $F_1$, one front is selected at a time to construct a new population, until its size equals to or exceeds the limit. The exceeded solutions in the last acceptable front will be eliminated according to the crowding distance metric to maintain population diversity. In `D-PBNSGA-II`, we replace the crowding distance with $\widetilde{Pr}(\mathbf{x})$. As a result, the solutions close to the SOI will survive to the next generation.

- The basic idea of the classic MOEA/D is to decompose the original MOP into a set of subproblems using weight vectors. Then, these subproblems are tackled collaboratively using population-based meta-heuristics. In `D-PBMOEA/D`, we progressively transform the originally uniformly distributed weight vectors $W = \{\mathbf{w}^i\}_{i=1}^N$ used in the original MOEA/D to the preference distribution $\widetilde{\mathrm{Pr}}(\mathbf{x})$. In practice, the transformed weight vector is $\mathbf{w}^{i\prime} = \widetilde{\mathrm{Pr}}^{-1}(\mathbf{w}^i)$, where $\widetilde{\mathrm{Pr}}^{-1}(\cdot)$ is the weighted sum of inverse Gaussian distribution, defined in equation (19).

Detailed implementation of `D-PBNSGA-II` and `D-PBMOEA/D` are in Appendix C.

**When to stop querying DM?** There is no principled termination criterion in exiting PBEMO, but is often set as a pre-defined number of consultation sessions. This is not rationale and likely to incur unnecessary workloads to DM, even when the evolutionary population is either converged to the SOI or being trapped by local optima. Under our `D-PBEMO` framework, we have the following theoretical result about the convergence property of $\widetilde{\mathrm{Pr}}(\mathbf{x})$.

**Theorem 3.4.** *Assume the preference distribution $\widetilde{\mathrm{Pr}}(\mathbf{x})$ around the SOI follows a Gaussian mixture distribution. It becomes stable when $N_{\mathrm{consult}}$ increases.*

The proof of Theorem 3.4 is in Appendix B.3. In `D-PBEMO`, we apply Theorem 3.4 to adaptively terminate the `consultation` session when $\widetilde{\mathrm{Pr}}(\mathbf{x})$ becomes stable. In practice, this happens when the Kullback–Leibler (KL) divergence of $\widetilde{\mathrm{Pr}}(\cdot)$ between two consecutive consultation sessions is smaller than a threshold $\varepsilon$:

$$D_{\mathrm{KL}}\left(\widetilde{\mathrm{Pr}}_{\tau-1} \| \widetilde{\mathrm{Pr}}_\tau\right) = \sum_{i=1}^{N_{\mathrm{s}}} \widetilde{\mathrm{Pr}}_{\tau-1}\left(\hat{\mathbf{x}}_i\right) \frac{\log \widetilde{\mathrm{Pr}}_{\tau-1}\left(\hat{\mathbf{x}}_i\right)}{\log \widetilde{\mathrm{Pr}}_\tau\left(\hat{\mathbf{x}}_i\right)}, \tag{5}$$

where $1 < \tau \leq N_{\mathrm{consult}}$, $\hat{\mathbf{x}}_i$ is sampled from $\tilde{\mathcal{S}}^\star$. $N_{\mathrm{s}}$ is the number of samples when calculating the KL divergence. Here we use $\varepsilon = 10^{-3}$, and its parameter sensitivity is studied in Section 4.4.

## 4 Experiments

### 4.1 Experimental Setup

This section outlines some key experimental setup including benchmark test problems and performance metrics. More detailed information can be found in Appendix D.

**Benchmark problems**   Our experiments considers 33 *synthetic test instances* including ZDT1 to ZDT4 and ZDT6 [72] ($m = 2$), DTLZ1 to DTLZ6 [18] where $m = \{3, 5, 8, 10\}$, and WFG1, WFG3, WFG5, and WFG7 [30] ($m = 3$). These problems are with various PF shapes and challenges such as bias, plateau, and multi-modal. In addition, we also consider two *scientific discovery problems* including 10 two-objective RNA inverse design tasks [55, 70] and 5 four-objective protein structure prediction (PSP) task [69]. The problem formulations are detailed in Appendix D.1.

**Performance metrics**   As discussed in [37], quality assessment of non-dominated solutions is far from trivial when considering DM's preference information regarding conflicting objective. In our experiments, we consider two metrics to serve this purpose. One is approximation accuracy $\epsilon^\star(\mathcal{S})$ that evaluates the closest distance of $\mathbf{x} \in \mathcal{S}$ regarding the DM preferred solution in the objective space, denoted as 'golden point' $\mathbf{z}^\star \in \mathbb{R}^m$, while the other is average accuracy $\bar{\epsilon}(\mathcal{S})$ that evaluates the average distance to $\mathbf{z}^\star$. Note that $\mathbf{z}^\star$ is unknown to the underlying algorithm in practice, and the corresponding settings used in our experiments are in Appendix D.2. Due to the stochastic

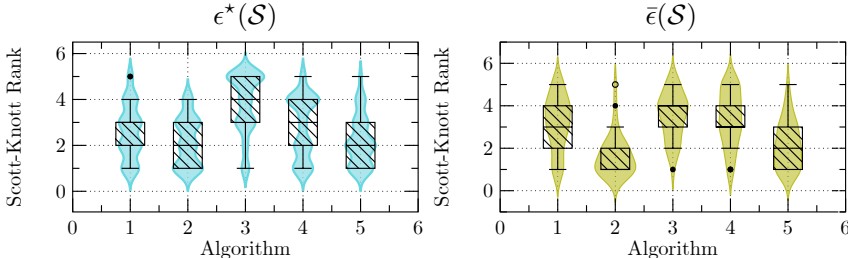

**Figure 4:** Box plot for the Scott-Knott test rank of `D-PBEMO` and peer algorithms achieved by 33 test problems running for 20 times. The index of algorithms are as follows: $1 \rightsquigarrow$ `D-PBNSGA-II`, $2 \rightsquigarrow$ `D-PBMOEA/D`, $3 \rightsquigarrow$ `I-MOEA/D-PLVF`, $4 \rightsquigarrow$ `I-NSGA-II/LTR`, $5 \rightsquigarrow$ `IEMO/D`.

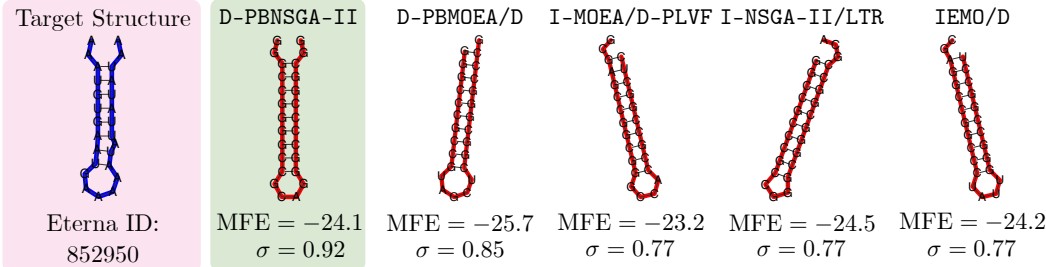

**Figure 5:** Comparison result of `D-PBNSGA-II` against the other three state-of-the-art PBEMO algorithms on a selected RNA inverse design task (Eterna ID: 852950). The target structure is shaded in blue color while the predicted structures obtained by different optimization algorithms are highlighted in red color. In this experiment, the preference is set to $\sigma = 1$. The closer $\sigma$ is to 1, the better performance achieved by the corresponding algorithm. When the $\sigma$ shares the same biggest value, the smaller $MFE$ the better the performance is. Full results can be found in Appendix F.

nature of evolutionary computation, each experiment is repeated 20 times with different random seeds. To derive a statistical meaning of comparison results, we consider Wilcoxon rank-sum test and Scott-Knott test in our experiments. They are briefly introduced in Appendix D.3.

### 4.2 Comparison Results with State-of-the-art PBEMO algorithms

To validate the effectiveness of our proposed `D-PBEMO` framework, we first compare the performance of `D-PBNSGA-II` and `D-PBMOEA/D` against three state-of-the-art PBEMO algorithms, `I-MOEAD-PLVF` [36], `I-NSGA2/LTR` [38], `IEMO/D` [58].

For the synthetic benchmark test problems, the comparison results of $\epsilon^\star(\mathcal{S})$ and $\bar{\epsilon}(\mathcal{S})$ in Tables A5 and A6 have demonstrated the competitiveness of our proposed `D-PBEMO` algorithm instances. In particular, `D-PBNSGA-II` and `D-PBMOEA/D` have achieved the best metric values in 77 out of 96 comparisons according to the Wilcoxon rank-sum test at the $0.05$ significant level. In Figure 4, the box plots of the ranks derived from the Scott-Knott test of `D-PBNSGA-II` and `D-PBMOEA/D` compared against the other three peer algorithms further consolidate our observation about the effectiveness of our `D-PBEMO` framework for searching for SOI within a given computational budget. From the plots of the final non-dominated solutions obtained by different algorithms shown in Figures A8 to A9, we can see that the solutions found by our `D-PBEMO` algorithm instances are more concentrated on the 'golden point' while the others are more scattered. Further, the superiority of our proposed `D-PBEMO` algorithm instances becomes more evident when tackling problems with many objectives, i.e., $m \geq 3$. Similar observations can be found in the scientific discovery problems, as the results shown in Tables A22, A23, A24 and A25. For the RNA inverse design tasks, the sequences identified by our proposed `D-PBEMO` algorithm instances have a good match regarding the targets as the selected results shown in Figure 5 (full plots in Figures A14 and A15). As for the PSP problems, from the results in Figure 6, it is clear to see that `D-PBEMO` algorithm instances significantly outperform the other peers. The protein structures predicted by our algorithms are more aligned with the native protein structure. Full results can be found in Appendix F.

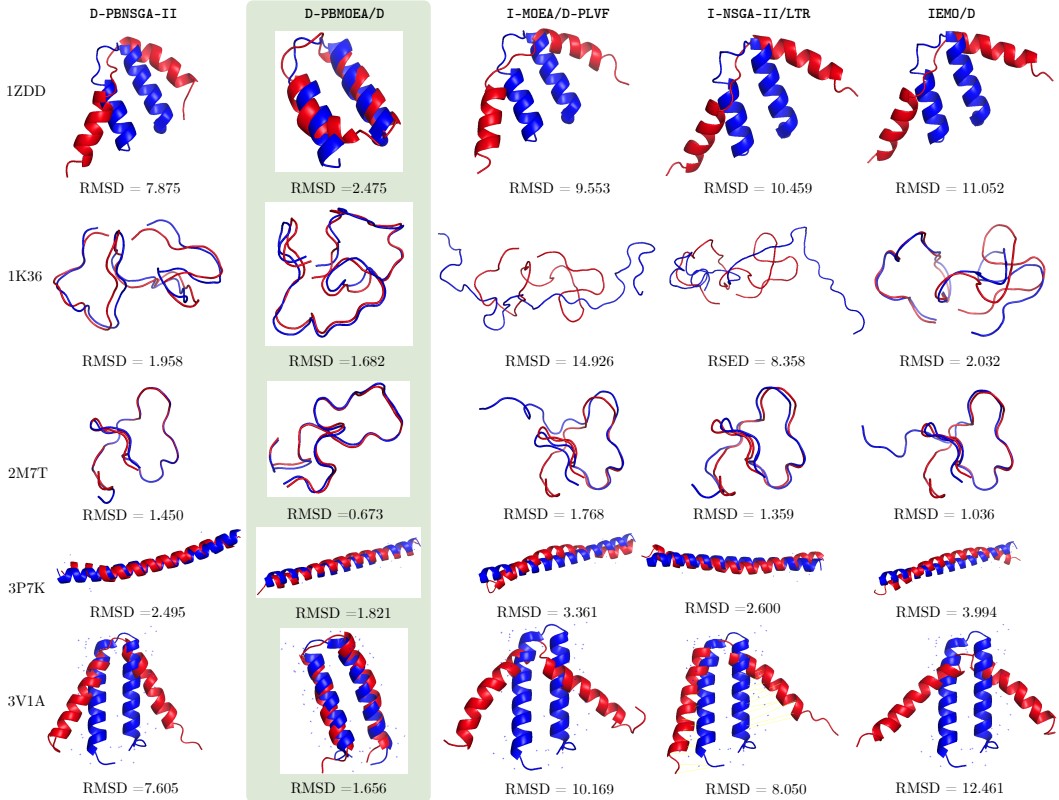

**Figure 6:** Experiments results for comparison results between D-PBEMO and the other three state-of-the-art PBEMO algorithms on the PSP problems. In particular, the native protein structure is represented in a blue color while the predicted one obtained by different optimization algorithms are highlighted in a red color. The smaller RMSD as defined in Equation (29) of appendix, the better performance achieved by the corresponding algorithm.

## 4.3 Investigation of the Effectiveness of Our `Consultation` Module

The `consultation` module, which learns the DM's preferences from their feedback, is one of the most important building blocks of our proposed D-PBEMO framework. To validate its effectiveness, we designed a D-PBEMO variant (denoted as D-PBEMO-DTS) that uses the double Thompson sampling (DTS) widely used in conventional stochastic dueling bandits [62] as an alternative of our proposed clustering-based stochastic dueling bandit algorithm. From the results in Tables A7 and A8, it is clear to see that D-PBEMO-DTS is always outperformed by our three D-PBNSGA-II and D-PBMOEA/D. This observation can be attributed to the ineffectiveness of the traditional dueling bandit algorithms for tackling a large number of arms. In our PBEMO context, DTS requires at least thousands comparisons when encountering more than 100 candidate solutions. This is not feasible under the limited amount of computational budget. We envisage the same results will be obtained when considering other dueling bandits variants such as [74, 34].

Further, we designed another variant (denoted as D-PBEMO-PBO) that uses a parameterized preference learning model in Bayesian optimization [25] as alternative in the `consultation` module. From the comparison results shown in Tables A7 and A8, we find that the performance of D-PBEMO-PBO is competitive on problems with a small number of objective ($m = 2$). However, its performance degenerate significantly with the increase of the number of objectives. This can be attributed to the exponentially increased search that renders D-PBEMO-PBO ineffective by suggesting too many solutions outside of the region of interest. In contrast, the clustering strategy in our proposed method strategically and significantly narrow down the amount of comparisons without compromising the learning capability. Additionally, the enlarged search space also makes the model selection in D-PBEMO-PBO difficult. Our proposed method, on the other hand, learns human preferences directly from their feedback, thus is scalable to a high-dimensional space.

### 4.4 Parameter Sensitivity Study

This section discusses the sensitivity study of two hyperparameters of the D-PBEMO framework.

$K$ is the parameter that controls the number of subsets used in our clustering-based stochastic dueling bandits algorithm. From the results shown from Tables A11 and A12 considering $K = \{2, 5, 10\}$, it is interesting to note that D-PBEMO is not sensitive to the setting of $K$ when the dimensionality is low (i.e., $m = 2$). This implies that we can achieve a reasonably good performance even when using a smaller $K$, i.e., a coarser-grained approximation of SOI. By doing so, we can further improve the efficiency of our D-PBEMO framework.

As the dimensionality increases (i.e., $m = \{3, 5, 8, 10\}$), the population size will increase. Generally, as is shown in Table A13 ~ A20, with larger populations, a higher $K$ tends to yield better results, aligning with our intuition. Furthermore, our significance analysis across 20 repeated experiment (Figure 4) reveals that the optimal $K$ does not show significant differences in performance. In summary, $K$ does not significantly impact the performance of our proposed D-PBEMO framework. For most problems, we do not recommend choosing a very small/large $K$ (e.g., $K = 2$, $K = N$), as it may inefficiently narrow down the ROI.

To a certain extent, the threshold $\varepsilon$ plays an important role for controlling the budget of consulting the DM. From the results shown in Tables A9 and A10, we find that all three settings of $\varepsilon = \{10^{-1}, 10^{-3}, 10^{-6}\}$ have shown comparable results. However, a too large $\varepsilon = 10^{-1}$ may lead to a premature convergence risk. On the other hand, a too small $\varepsilon = 10^{-6}$ may be too conservative to terminate. This renders more consultation iterations, thus leading to a larger amount of cognitive workloads to DMs. In contrast, we find that D-PBEMO algorithm instances can converge with less than 10 consultation iterations with $\varepsilon = 10^{-3}$.

## 5   Limitations

Our proposed D-PBEMO directly leverages DM's feedback to guide the evolutionary search for SOI, which neither relies on any reward model nor cumbersome model selection. However, there are several potential limitations of D-PBEMO that warrant discussion here:

- The regret analysis of our proposed clustering-based stochastic dueling bandits is for the optimal subset, i.e., the region of interest on the PF. It is not yet directly applicable to identify the exact optimal solution of interest. As part of our future work, we will work on efficient algorithms and theoretical study on the best arm identification in the context the preference-based EMO.

- This paper only analyzes the regret of the consultation module. How to further analyze the convergence of the D-PBEMOas a whole remains unknown. This will also lead to the next step of our research. In particular, if it is successful, we may provide a radically new perspective to analyze the convergence of evolutionary multi-objective optimization algorithms.

## 6   Conclusion

This paper introduced the D-PBEMO framework that is featured in a novel clustering-based stochastic dueling bandits algorithm. It enables learning DM's preferences directly from their feedback, neither relying on a predefined reward model nor cumbersome model selection. Additionally, we derived a unified probabilistic format to adapt the learned preference to prevalent EMO algorithms. Meanwhile, such probabilistic representation also contributes to a principled termination criterion of DM interactions. Experiments demonstrate the effectiveness of our proposed D-PBEMO algorithm instances. In future, we will investigate more principled approaches to obtain $K$ subsets, such as fuzzy clustering with overlapping. Further, we plan to extend this current $K$-armed bandits setup to a best arm identification. By doing so, we can directly obtain the solution of interest, with a potential explainability for MCDM. Moreover, we will extend our theoretical analysis of the termination criterion to a convergence analysis of PBEMO, even generalizable to the conventional EMO. Last but not the least, we will collaborate with domain experts to promote a emerging 'expert-in-the-loop' scientific discovery platform, contributing to the prosperity of AI for science.

## Acknowledgements

This paper presents work whose goal is to advance the field of Machine Learning. There are many potential societal consequences of our work, none which we feel must be specifically highlighted here. Li was supported by UKRI Future Leaders Fellowship (MR/X011135/1, MR/S017062/1), NSFC (62376056, 62076056), the Kan Tong Po Fellowship (KTP/R1/231017), Alan Turing Fellowship, and Amazon Research Award.

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

# A  Related Works

## A.1  Multi-objective optimization

In the context of MO, to align with DM's preference, existing methods fall into three main categories: the *a posteriori* methods, the *a priori* methods, and the *interactive* methods [12]. The *a posteriori* methods first generates extensive efficient solutions through MO algorithms, then ask DM to select SOI from these solutions. In this way, DM is involved only when solution generation from the optimization module is finished, where the decision making process can be finished according to multi-criterion decision analysis [26] or certain SOI analysis [13, 6, 40]. However, the absence of DM's preferences in solution generation can paradoxically complicate pinpointing the SOI, and the sheer number of generated solutions can be cognitively overwhelming for DM to choose from. The *a priori* method, leverages pre-defined DM's preference to guide the search of SOI [7, 42, 57, 37]. In other words, the decision making happens before optimization. This is more related to offline preference learning such as reinforcement learning from human feedback (RLHF) [38]. However, given the black-box nature of real-world problems, eliciting reasonable preferences a priori can be controversial. [39, 54] pointed out that the *a priori* method may lead to disruptive preferences and result in faulty decisions. Moreover, elaborating the DM labeled dataset used for offline preference learning such as RLHF is also a difficult task. In contrast, the *interactive* method also known as preference-based evolutionary multi-objective optimization (PBEMO) [39] presents a valuable opportunity for the DM to gradually comprehend the underlying black-box system and consequently refine user preference information [44, 15, 8, 36, 59, 38].

## A.2  Preference learning in PBEMO

The consultation module in PBEMO collects DM's preference by actively querying DM's preference towards recommended candidates. Based on the requirements of human feedback, methods for consultation can be categorized into the following three types, as also presented in Figure 1(**b**).

- The first type requires DM to provide scores or rankings for a bunch of (typically more than three) solutions. In [38], a rank-net was introduced to conduct consultation in PBEMO. In [27], a ranking model was required in the context of RLHF. In addition, as presented in [50], relative preference can only be leveraged by ranking on more than three candidates. However, providing reward or ranking on a bunch of solutions will not only increase the workload of DMs, but will also introduce uncertainty and randomness into optimization processes due to the limited capacity of DMs [48].

- Algorithms in the second type rely on a parameterized transitivity model or a structured utility function for dueling feedback [5], such as the Bradley-Terry-Luce model [11] which has been widely explored in the literature [32, 24, 11, 29, 71]. These algorithms can be traced back to 1982, when [32] introduced the UTA (UTilités Additives) method for deducing value functions based on a provided ranking of reference set. Following up work included [24] that employed pairwise preference to predict a ranking for potential labels associated with new training examples, [11] that utilized GP for pairwise preference learning (PGP) within a Bayesian framework, and [9] that proposed the ListNet which was an NN-based learning-to-rank method to model preference feedback. In addition, [29] extended the work in [9] to handle multi-user scenarios by introducing a weight vector for each user and combining multiple preference latent functions. More recently, [25] proposed a novel pairwise preference learning method using the concept of Bayesian optimization based on a structured surrogate model assumption, named preferential Bayesian optimization (PBO). Moreover, in [71], a comparative study towards four prominent preference elicitation algorithms was conducted, and results shown that ranking queries outperformed the pairwise and clustering approaches in terms of utility models ad human preference.

- Algorithms that belong to the third type consider human feedback as stochastic events such as Bernoulli trails, enabling quantifying randomness regarding the DM's capability. Dueling bandits algorithms are known for its efficiency towards interactive preference learning [65]. In recent years, work like [75, 62, 63] has improved dueling bandits algorithms in both querying efficiency and computational complexity. Further investigations on dueling bandits are presented in A.3.

Algorithms in the first two types burden the development of PBEMO, as reward/ranking method fails to take human capability into account and will navigate solutions to an inaccurate direction, while the parameterized method relies heavily on model selection. It is well known that, in real-world problems, selecting an appropriate model is as difficult as solving the original problems. We appreciate the model-free settings in dueling bandits algorithms.

### A.3 Bandit algorithms

The dueling bandits problem involves a sequential decision-making process where a learner selects two out of $K$ "arms" in each round and receives binary feedback about which arm is preferred. Following [4], dueling bandits algorithms can be classified into three categories: MAB-related, merge sort/quick sort, and tournament/challenge. In this paper, we focus on traditional dueling bandit algorithms falling in the MAB-related category. comprising four distinct methodologies for handling pairwise comparisons.

- The first method is known as explore then commit (ETC), which is utilized by algorithms such as interleaved filtering (IF) [65], beat the mean (BTM) [66] and SAVAGE [60]. ETC methods kick out solutions that are unlikely to win, but this approach may lead to lower accuracy of predictive probability.

- The second method involves using the upper confidence bound (UCB), for example relative upper confidence bound (RUCB) [74], MergeRUCB [73], and relative confidence sampling (RCS) [75]. MergeRUCB, an extension of RUCB, is particularly designed for scenarios with a large number of arms. RCS combines UCB and Beta posterior distribution to recommend one arm for each duel in each iteration step.

- The third method employs Thompson sampling, as demonstrated by double Thompson sampling (DTS) [62] and MergeDTS [35]. Similar to MergeRUCB, MergeDTS is designed for dealing with a substantial number of arms. It is worth nothing that UCB methods assume the existence of a Condorcet winner, whereas Thompson sampling methods assume a Copeland winner, representing a fundamental distinction between these two types.

- The fourth method involves using the minimum empirical divergence, as introduced by relative minimum empirical divergence (RMED) [34] and deterministic minimum empirical divergence (DMED) [28]. RMED and DMED employ KL divergence as a metric to evaluate candidate arms.

Overall, these four methods represent different approaches to pairwise comparison in traditional dueling bandit algorithms. However, the key bottleneck here is the sourcing number of DM queries required when considering a population solutions in PBEMO.

To our knowledge, there is no clustering-based dueling bandit algorithms. While in the field of multi-armed bandit (MAB), there has already exists clustered MABs which consider the correlation of arms [47, 46], they relied heavily on complex model selection that is not applicable for real-world scenarios.

Additionally, it is noted that multi-objective multi-armed bandit (MOMAB) is potential to address PBEMO problems. However, as pointed out in [31, 56], MOMABs belong to the *a posteriori* MO method, recalling A.1. Specifically, the goal of MOMABs is to find an optimal multi-objective arm by sampling potential winners. However, MOMABs do not involve optimization for PF generation nor active query for preference learning. Thus it's not feasible to use MOMABs as our peer algorithms or consultation module.

## B  Derivation and Theoretical Analysis

### B.1  Regret bound of clustering dueling bandits

*Proof of Theorem 3.3.* We follow the proof of double Thompson sampling for dueling bandits [62], different from which the clustering operation will lead to a non-stationary environment. As shown subsequently, the Assumption 3.2 is important to build an auxiliary problem that is stationary, facilitating to apply the results in [62].

According to the property of Beta distribution, the estimated mean of probability for subset $i$ beats subset $j$ is $\tilde{p}_{i,j}(t) = \frac{b_{i,j}(t)}{b_{i,j}(t-1)+b_{j,i}(t-1)}$. Unfortunately, the pairwise comparison feedback does not follow a stationary distribution. Specifically, $p_{i,j}$ represents winning probability between the best solutions in $\tilde{\mathcal{S}}^i$ and $\tilde{\mathcal{S}}^j$, while $\tilde{p}_{i,j}(t)$ is computed from comparison among different solutions in the two subsets due to limited knowledge of the best solutions at the beginning. As a result, the reward distribution is drifting [10]. To handle this, we introduce an auxiliary problem with stationary reward distributions. Based on comparison principles, the number of plays on each subset in the original problem can be upper bounded by the ones of the auxiliary problem. Note that the auxiliary problem are introduced merely for theoretical analysis.

Without loss of generality, we assume $\tilde{\mathcal{S}}^1$ to be $\tilde{\mathcal{S}}^*$. Since we are most interested in locating the optimal subset, we define the underestimation of winning probability of $p_{1,j}$ as $\underline{p}_{1,j}$, which is the winning probability between: $i$) the solution $\underline{\mathbf{x}}^1 \in \tilde{\mathcal{S}}^1$ with the lowest Copeland score over all solutions in $\tilde{\mathcal{S}}^j$, and $ii$) the solution $\overline{\mathbf{x}}^j \in \tilde{\mathcal{S}}^j$ with the highest Copeland score over all solutions in $\tilde{\mathcal{S}}^1$, $j = 2, \ldots K$. Likewise, we define $\underline{p}_{i,j}$ as the winning probability between: $i$) the solution $\underline{\mathbf{x}}^i \in \tilde{\mathcal{S}}^i$ with the lowest Copeland score over all solutions in $\tilde{\mathcal{S}}^j$, and $ii$) the solution $\overline{\mathbf{x}}^j \in \tilde{\mathcal{S}}^j$ with the highest Copeland score over all solutions in $\tilde{\mathcal{S}}^1$, $p_{i,j} > 0.5$, $ij = 1, \ldots K$. For $p_{i,j} < 0.5$, denote $\underline{p}_{i,j} = 1 - \underline{p}_{j,i}$. Lastly, we denote $\underline{p}_{i,i} = 0.5$, $\forall i = 1, \ldots, K$. According to Assumption 3.2, the optimality remains, i.e., the optimal subset in the context of $p_{ij}$ is also optimal in the context of $\underline{p}_{ij}$.

With the auxiliary problem, we introduce the virtual Thompson sampling strategy that generates samples as $\theta'_{j,i} \sim \text{Beta}(b'_{j,i}(t) - 1, b'_{i,j}(t) - 1)$, where $b'_{i,j}$ is the number of time-slots when subsets $\tilde{\mathcal{S}}^i$ beats $\tilde{\mathcal{S}}^j$ under $\underline{p}_{i,j}$. It is easy to verify two facts in expectation: $i$) $b'_{i,j} \leq b_{i,j}$, and $ii$) $b'_{j,i} \geq b_{j,i}$. Consequently, when $p_{i,j} > 0.5$ or $i = 1$, for samples $\theta_{i,j}(t) \sim \text{Beta}(b_{i,j} + 1, b_{j,i} + 1)$ and $\theta'_{i,j}(t) \sim \text{Beta}(b'_{i,j} + 1, b'_{j,i} + 1)$, we have $\theta_{i,j}$ dominates $\theta'_{i,j}$ in expectation [10].

In what follows, we are going to quantify the number of plays of subset pair $\tilde{\mathcal{S}}^i$ and $\tilde{\mathcal{S}}^j$ To avoid ambiguity, let $N_{i,j}(t)$ denote the case where $\tilde{\mathcal{S}}' = \tilde{\mathcal{S}}^i$ and $\tilde{\mathcal{S}}'' = \tilde{\mathcal{S}}^j$.

When the first subset for comparison has been determined as $\tilde{\mathcal{S}}^i$, the number of plays for $\tilde{\mathcal{S}}^j$, $j \neq i$, is

$$
\begin{aligned}
\mathbb{E}[N_{i,j}(T)] &= \sum_{t=1}^{T} \mathbb{P}\left(\tilde{\mathcal{S}}'' = \tilde{\mathcal{S}}^j\right) \\
&= \sum_{t=1}^{T} \mathbb{P}\left(\tilde{\mathcal{S}}'' = \tilde{\mathcal{S}}^j, \underline{p}_{j,i} < 0.5\right) + \sum_{t=1}^{T} \mathbb{P}\left(\tilde{\mathcal{S}}'' = \tilde{\mathcal{S}}^j, \underline{p}_{j,i} > 0.5\right). \quad (6)
\end{aligned}
$$

When $\underline{p}_{i,j} < 0.5$, we consider the following facts: $i$) $\theta_{j,i}$ are sampled from $\text{Beta}(b_{j,i} + 1, b_{i,j} + 1)$ when $j \neq i$; $ii$) $\theta_{i,i} = 0.5$; $iii$) $\theta_{j,i} > \theta_{i,i}$ since subset $\tilde{\mathcal{S}}^j$ is selected by sampling for comparison. From these observations, the problem is equivalent to a multi-arm bandit problem where the best arm is indexed by $i$. Then, the left item of equation (6) can be further divided by

$$
\begin{aligned}
&\sum_{t=1}^{T} \mathbb{P}\left(\tilde{\mathcal{S}}'' = \tilde{\mathcal{S}}^j, \underline{p}_{j,i} < 0.5\right) \\
&= \sum_{t=1}^{T} \mathbb{P}\left(\tilde{\mathcal{S}}'' = \tilde{\mathcal{S}}^j, \underline{p}_{j,i} < c < 0.5\right) + \sum_{t=1}^{T} \mathbb{P}\left(\tilde{\mathcal{S}}'' = \tilde{\mathcal{S}}^j, c < \underline{p}_{j,i} < 0.5\right) \\
&= \sum_{t=1}^{T} \mathbb{P}\left(\tilde{\mathcal{S}}'' = \tilde{\mathcal{S}}^j, \underline{p}_{j,i} < c, N_{j,i}(t-1) \leq \frac{\log T}{D_{\text{KL}}(c\|0.5)}\right) \\
&\quad + \sum_{t=1}^{T} \mathbb{P}\left(\tilde{\mathcal{S}}'' = \tilde{\mathcal{S}}^j, \underline{p}_{j,i} < c, N_{j,i}(t-1) > \frac{\log T}{D_{\text{KL}}(c\|0.5)}\right) + \sum_{t=1}^{T} \mathbb{P}\left(\tilde{\mathcal{S}}'' = \tilde{\mathcal{S}}^j, c < \underline{p}_{j,i} < 0.5\right).
\end{aligned}
$$
$$(7)$$

When $\underline{p}_{j,i} < 0.5$, if $p_{j,i} > 0.5$, we have $p_{j,i} > \underline{p}_{j,i}$, thus $\theta_{j,i}$ dominates $\theta'_{j,i}$ in expectation. Therefore, we have $\mathbb{P}\left(N_{j,i}(t-1) \leq \frac{\log T}{D_{\mathrm{KL}}(c\|0.5)}\right) \leq \mathbb{P}\left(\underline{N}_{j,i}(t-1) \leq \frac{\log T}{D_{\mathrm{KL}}(c\|0.5)}\right)$, where $\underline{N}_{j,i}(t-1)$ is the number of plays of pair $\tilde{\mathcal{S}}^j$ against $\tilde{\mathcal{S}}^i$ using the virtual Thompson sampling strategy. Otherwise, if $p_{j,i} < 0.5$, we have $p_{j,i} < \underline{p}_{j,i}$, thus $\mathbb{P}(p_{j,i} < c) \geq \mathbb{P}\left(\underline{p}_{j,i} < c\right)$ which follows the regular form as Lemma 1 in [62]. According to the concentration property of Thompson samples [2], we have

$$
\sum_{t=1}^{T} \mathbb{P}\left(\tilde{\mathcal{S}}'' = \tilde{\mathcal{S}}^j, \underline{p}_{j,i} < 0.5\right)
$$

$$
\leq \sum_{t=1}^{T} \mathbb{P}\left(\tilde{\mathcal{S}}'' = \tilde{\mathcal{S}}^j, \underline{p}_{j,i} < c, \underline{N}_{j,i}(t-1) \leq \frac{\log T}{D_{\mathrm{KL}}(c\|0.5)}\right)
$$

$$
+ \sum_{t=1}^{T} \mathbb{P}\left(\tilde{\mathcal{S}}'' = \tilde{\mathcal{S}}^j, \underline{p}_{j,i} < c, N_{j,i}(t-1) > \frac{\log T}{D_{\mathrm{KL}}(c\|0.5)}\right) + \sum_{t=1}^{T} \mathbb{P}\left(\tilde{\mathcal{S}}'' = \tilde{\mathcal{S}}^j, c < \underline{p}_{j,i} < 0.5\right)
$$

$$
\leq \frac{\log T}{D_{\mathrm{KL}}(c\|0.5)} + \frac{1}{D_{\mathrm{KL}}(c\|\underline{p}_{j,i})} + 1. \tag{8}
$$

As shown in [2], for any $\epsilon \in (0,1]$, $c$ can be chosen such that $D_{\mathrm{KL}}(c\|0.5) = D_{\mathrm{KL}}(\underline{p}_{j,i}\|0.5)/(1+\epsilon)$, and $1/D_{\mathrm{KL}}(c\|\underline{p}_{j,i}) = O(1/\epsilon^2)$. Therefore the left item in equation (6) is bounded from above by

$$
\sum_{t=1}^{T} \mathbb{P}\left(\tilde{\mathcal{S}}'' = \tilde{\mathcal{S}}^j, \underline{p}_{j,i} < 0.5\right) \leq (1+\epsilon)\frac{\log T}{D_{\mathrm{KL}}(\underline{p}_{j,i}\|0.5)} + \mathcal{O}\left(\frac{1}{\epsilon^2}\right). \tag{9}
$$

To account for the second term in equation (6), the concentration property of relative upper confidence bound [74, 62] will be used. For $\underline{p}_{j,i} > 0.5$, define $\Delta_{j,i} = \underline{p}_{j,i} - 0.5 > 0$, thus the right side of equation (6) can be divided into

$$
\sum_{t=1}^{T} \mathbb{P}\left(\tilde{\mathcal{S}}'' = \tilde{\mathcal{S}}^j, \underline{p}_{j,i} > 0.5\right)
$$

$$
= \sum_{t=1}^{T} \mathbb{P}\left(\tilde{\mathcal{S}}'' = \tilde{\mathcal{S}}^j, \underline{p}_{j,i} > 0.5, N_{j,i}(t-1) < \frac{4\alpha \log T}{\Delta_{j,i}}\right)
$$

$$
+ \sum_{t=1}^{T} \mathbb{P}\left(\tilde{\mathcal{S}}'' = \tilde{\mathcal{S}}^j, \underline{p}_{j,i} > 0.5, N_{j,i}(t-1) \geq \frac{4\alpha \log T}{\Delta_{j,i}}\right). \tag{10}
$$

Note that the subset $\tilde{\mathcal{S}}^j$ can be selected as $\tilde{\mathcal{S}}''$ only if $l_{j,i} < 0.5$. For the right item in equation (10), $N_{j,i}(t-1) \geq \frac{4\alpha \log T}{\Delta_{j,i}}$ is sufficient to yield $l_{j,i} + \Delta_{j,i} \geq u_{j,i}$, finally we have $u_{j,i} \leq p_{j,i}$. Applying concentration properties in Lemma 6 of [62], we have

$$
\sum_{t=1}^{T} \mathbb{P}\left(\tilde{\mathcal{S}}'' = \tilde{\mathcal{S}}^j, \underline{p}_{j,i} > 0.5, N_{j,i}(t-1) \geq \frac{4\alpha \log T}{\Delta_{j,i}}\right) \leq \sum_{t=1}^{T} \mathbb{P}(u_{i,j} \leq p_{i,j}) = \mathcal{O}(1).
$$

As a result, we can also have a upper bound of the left item in equation (10) as

$$
\sum_{t=1}^{T} \mathbb{P}\left(\tilde{\mathcal{S}}'' = \tilde{\mathcal{S}}^j, \underline{p}_{j,i} > 0.5, N_{j,i}(t-1) < \frac{4\alpha \log T}{\Delta_{j,i}}\right) \leq \frac{4\alpha \log T}{\Delta_{j,i}}. \tag{11}
$$

Altogether, for $j \neq i$, we have

$$
\mathbb{E}[N'_{j,T}] = \sum_{t=1}^{T} \mathbb{P}\left(\tilde{\mathcal{S}}'' = \tilde{\mathcal{S}}^j\right) \leq (1+\epsilon)\frac{\log T}{D_{\mathrm{KL}}(\underline{p}_{j,i}\|0.5)} + \frac{4\alpha \log T}{\Delta_{j,i}} + \mathcal{O}\left(\frac{1}{\epsilon^2}\right).
$$

For the case where $\tilde{\mathcal{S}}'$ and $\tilde{\mathcal{S}}''$ are selected to be the same subsets, an upper bound of $\mathcal{O}(K)$ has been proven in Lemma 3 of [62]. Therefore we omit the proof of this case for brevity.

$\square$

## B.2 Density-ratio estimation of preference distribution

Consider the DM's preference distribution is determined by the density ratio of winning distribution over losing distribution, in other word, preference on solution $\tilde{\mathbf{x}}$ is determined by $p_v(\tilde{\mathbf{x}})/p_\ell(\tilde{\mathbf{x}})$. From consultation module, we have obtained the winning and losing time vectors $\mathbf{v}$ and $\boldsymbol{\ell}$ of all candidate solutions, which in turn gives us a density estimation of $p_v$ and $p_\ell$. We are going to perform density-ratio estimation based upon $\mathbf{v}$ and $\boldsymbol{\ell}$ to obtain the density ratio, namely the preference distribution $p_r(\tilde{\mathbf{x}})$, which can be mathematically formulated as

$$p_v(\tilde{\mathbf{x}}) = p_r(\tilde{\mathbf{x}})p_\ell(\tilde{\mathbf{x}}). \tag{12}$$

We use finite-order moment matching approach without conducting complex density estimation of each $p_v(\tilde{\mathbf{x}})$ and $p_\ell(\tilde{\mathbf{x}})$. Denote a vector-valued nonlinear function $\boldsymbol{\phi} : v \to \mathbb{R}^k$. Consider the first $k$-th order moments, we have $\boldsymbol{\phi}(\tilde{\mathbf{x}}) = \left(\tilde{\mathbf{x}}^{(1)}, \ldots, \tilde{\mathbf{x}}^{(k)}\right)$, where $\tilde{\mathbf{x}}^{(k)}$ stands for element-wise $k$-th power of $\tilde{\mathbf{x}}$. Then, the moment matching problem for density-ratio estimation can be formulated as

$$\arg\min_{p_r} \left\| \int \boldsymbol{\phi}(\tilde{\mathbf{x}})p_r(\tilde{\mathbf{x}})p_\ell(\tilde{\mathbf{x}}) \, \mathrm{d}\tilde{\mathbf{x}} - \int \boldsymbol{\phi}(\tilde{\mathbf{x}})p_v(\tilde{\mathbf{x}}) \, \mathrm{d}\tilde{\mathbf{x}} \right\|^2. \tag{13}$$

Equivalently, we can reformulate the problem by

$$\arg\min_{p_r} \left\| \int \boldsymbol{\phi}(\tilde{\mathbf{x}})p_r(\tilde{\mathbf{x}})p_\ell(\tilde{\mathbf{x}}) \, \mathrm{d}\tilde{\mathbf{x}} \right\|^2 - 2 \left\langle \int \boldsymbol{\phi}(\tilde{\mathbf{x}})p_r(\tilde{\mathbf{x}})p_\ell(\tilde{\mathbf{x}}) \, \mathrm{d}\tilde{\mathbf{x}}, \int \boldsymbol{\phi}(\tilde{\mathbf{x}})p_v(\tilde{\mathbf{x}}) \, \mathrm{d}\tilde{\mathbf{x}} \right\rangle, \tag{14}$$

where $\langle \cdot \rangle$ denotes the inner product. Viewing each component in equation (14) as the expectation operation of function $\boldsymbol{\phi}(\tilde{\mathbf{x}})p_r(\tilde{\mathbf{x}})$ over distribution $p_v(\tilde{\mathbf{x}})$ and $p_\ell(\tilde{\mathbf{x}})$, we use sample averages to estimate the expectation [52]. To this end, we construct two sample vectors $\Phi_v = \left(\boldsymbol{\phi}(\tilde{\mathbf{x}}_v^1), \ldots, \boldsymbol{\phi}(\tilde{\mathbf{x}}_v^{N_v})\right)$ and $\Phi_\ell = \left(\boldsymbol{\phi}(\tilde{\mathbf{x}}_\ell^1), \ldots, \boldsymbol{\phi}(\tilde{\mathbf{x}}_\ell^{N_\ell})\right)$ based on the wining and losing time vectors $\mathbf{v}$ and $\boldsymbol{\ell}$. Specifically $\tilde{\mathbf{x}}_v^i$ and $\tilde{\mathbf{x}}_\ell^i$ are solutions that has won or lost one time during consultation. Note that a solution may appear in $\Phi_v$ or $\Phi_\ell$ more than one time since its winning or losing time is greater than 1. $N_v$ and $N_\ell$ are the number of winning and losing solutions, calculated by the sum of all winning or losing times for all solutions. Correspondingly, we denote $\boldsymbol{p}_r = \left(p_r(\tilde{\mathbf{x}}_\ell^1), \ldots, p_r(\tilde{\mathbf{x}}_\ell^{N_\ell})\right)$. An estimator $\hat{\boldsymbol{p}}_r$ for $\boldsymbol{p}_r$ can be calculated by solving the following problem:

$$\hat{\boldsymbol{p}}_r = \arg\min_{\boldsymbol{p}_r \in \mathbb{R}^{N_\ell}} \frac{1}{N_\ell^2} \boldsymbol{p}_r^\top \Phi_\ell^\top \Phi_\ell \boldsymbol{p}_r - \frac{2}{N_\ell N_\ell} \boldsymbol{p}_r^\top \Phi_\ell^\top \Phi_v \mathbf{1}_{N_v}, \tag{15}$$

where $\mathbf{1}_{N_v}$ is the $N_v$-dimensional vector whose elements are all valued by 1. The optimality is reached by when the derivative of equation (15) is zero, i.e.,

$$\frac{2}{N_\ell^2} \Phi_\ell^\top \Phi_\ell \boldsymbol{p}_r - \frac{2}{N_\ell N_\ell} \Phi_\ell^\top \Phi_v \mathbf{1}_{N_v} = \mathbf{0}_k, \tag{16}$$

where $\mathbf{0}_k$ is the $k$-dimensional vector whose elements are all valued by 0. Correspondingly, we have

$$\hat{\boldsymbol{p}}_r = \frac{N_\ell}{N_v} \left(\Phi_\ell^\top \Phi_\ell\right)^{-1} \Phi_\ell^\top \Phi_v \mathbf{1}_{N_v}. \tag{17}$$

Note that $\hat{\boldsymbol{p}}_r$ is the estimation of true density ratio $\boldsymbol{p}_r$, namely the probability density function, on solutions $\tilde{\mathbf{x}}_\ell^i$, $i = 1, \ldots, N_\ell$. By assuming Gaussian distribution around SOI, we can further approximate the mean and variance by

$$\tilde{\mathbf{x}}_\tau^* = \sum_{i=1}^{N_\ell} \tilde{\mathbf{x}}_\ell^i \hat{\boldsymbol{p}}_r^i, \quad \tilde{\Sigma}_\tau = \mathrm{diag}\left\{ \sum_{i=1}^{N_\ell} \tilde{\mathbf{x}}_\ell^{i\,2} \hat{\boldsymbol{p}}_r^i - \tilde{\mathbf{x}}_\tau^{*\,2} \right\}, \tag{18}$$

where we dismiss the covariance between any two dimensions in solution space for brevity.

**Remark 4.** The sample averages can also be performed by first estimating the winning and losing density distributions based on $v$ and $\ell$, then sample solutions on them to increase the sample numbers. This will improve the estimation in equation (14) at the cost of complex density estimation. In addition, perturbations on solutions can be introduced for legal inverse operation of $\Phi_\ell^\top \Phi_\ell$ in equation (17).

## B.3 Convergence of preference distribution

*Proof of Theorem 3.4.* Form Theorem 3.3, when $N_{\text{consult}}$ increases, the clustering dueling bandits can grasp DM's preference towards pairwise comparisons with increasing accuracy. Therefore, the convergence of preference distribution $\widetilde{\Pr}(\mathbf{x})$ will be mainly influenced by the behaviors of EMO algorithms as well as the landscape of MO problems. To this end, we consider two extreme cases: $i)$ the population distribution of candidate solutions from EMO generators remains unchanged, and $ii)$ the population of solutions are steered into shrinking regions by EMO algorithms.

For the first case, we let the unchanged distribution be $\mathcal{N}(\mathbf{x} \mid \tilde{\mathbf{x}}_u^*, \tilde{\Sigma}_u), \forall \tau > N_u$, where $N_u > 1$ denotes a threshold number of consultation sessions. The convergence is presented by the following truth:

$$
\begin{aligned}
\widetilde{\Pr}(\mathbf{x}) = & \frac{1}{Z} \sum_{\tau=1}^{N_u} \frac{1}{\tilde{\sigma}_\tau} \mathcal{N}(\mathbf{x} \mid \tilde{\mathbf{x}}_\tau^*, \tilde{\Sigma}_\tau) + \frac{1}{Z} \sum_{\tau=N_u+1}^{N_{\text{consult}}} \frac{1}{\tilde{\sigma}_u} \mathcal{N}(\mathbf{x} \mid \tilde{\mathbf{x}}_u^*, \tilde{\Sigma}_u) \\
= & \frac{1}{\sum_{\tau=1}^{N_u} \frac{1}{\tilde{\sigma}_\tau} + (N_{\text{consult}} - N_u)\frac{1}{\tilde{\sigma}_u}} \sum_{\tau=1}^{N_u} \frac{1}{\tilde{\sigma}_\tau} \mathcal{N}(\mathbf{x} \mid \tilde{\mathbf{x}}_\tau^*, \tilde{\Sigma}_\tau) \\
& + \frac{\frac{1}{\tilde{\sigma}_u}(N_{\text{consult}} - N_u)\mathcal{N}(\mathbf{x} \mid \tilde{\mathbf{x}}_u^*, \tilde{\Sigma}_u)}{\sum_{\tau=1}^{N_u} \frac{1}{\tilde{\sigma}_\tau} + (N_{\text{consult}} - N_u)\frac{1}{\tilde{\sigma}_u}}.
\end{aligned}
$$

Note that $\frac{1}{\tilde{\sigma}_u} > 0$ and is constant. When $N_{\text{consult}} \gg N_u$, we have $\widetilde{\Pr}(\mathbf{x}) \to 0 + \mathcal{N}(\mathbf{x} \mid \tilde{\mathbf{x}}_u^*, \tilde{\Sigma}_u)$, therefore $\widetilde{\Pr}(\mathbf{x})$ is convergent to $\mathcal{N}(\mathbf{x} \mid \tilde{\mathbf{x}}_u^*, \tilde{\Sigma}_u)$.

For the second case, shrinking regions with increasing $N_{\text{consult}}$ result in a uniform distribution in the context of Gaussian distribution around SOI. In this case winning and losing probability are close to each other. As a result, a large $\tilde{\sigma}_\tau$ will be computed. Therefore this distribution hardly contributes to the mixture distribution. Consequently, with shrinking regions, effect of consultation sessions decreases to zero, thus $\widetilde{\Pr}(\mathbf{x})$ convergent ultimately.

Based on these observations, two extreme cases can result in a convergent stable mixture distribution, covering all other cases in PBEMO between the two extreme ones. The proof is complete. $\qquad\square$

## C  Algorithmic Implementation of Our `D-PBEMO` Instances

The structural pseudocode for `D-PBEMO` is shown in Algorithm 1. The `D-PBEMO` framework is made up of three component, consultation, preference elicitation and optimization module. In this paper, our contribution is mainly in consultation and preference elicitation module (highlighted by $rhd$). In consultation module, we proposed a clustering-based stochastic dueling bandit algorithm (pseudocode can be referenced in Algorithm 2). While in this section, we will delineate the step-by-step process of three different preference elicitation module. The code of our algorithms and peer algorithms are available at `https://github.com/COLA-Laboratory/EMOC/`.

### C.1  `D-PBNSGA-II`

This section will discuss how the learned preference information can be used in NSGA-II [17]. Based on [16], solutions from the best non-domination levels are chosen front-wise as before and a modified crowding distance operator is used to choose a subset of solutions from the last front which cannot be entirely chosen to maintain the population size of the next population, the following steps are performed:

Step 1: Before the first consultation session, the NSGA-II runs as usual without considering the preference information.

Step 2: If it is time to consult user preferences (e.g., when we have evaluated the population for $50\%$ of the total generation), we will update the preference distribution $\widetilde{\Pr}(\mathbf{x})$ defined in equation (4).

Step 3: Between two interactions, the crowding distance of each solution will be evaluated by the predicted preference distribution $\widetilde{\Pr}(\mathbf{x})$ learned from the last consultation session.

**Algorithm 1** D-PBEMO structure

---

**Input:** $G$ maximum generation, $N$ solutions in population.
1: $\tau \leftarrow 0$; // $\tau$ represents the consultation session count.
2: $current\_generation \leftarrow 0$.
3: $\widetilde{\mathrm{Pr}}_0(\cdot) \leftarrow 0$.
4: **while** $current\_generation < G$ **do**
5:     // Phase 1:Consult user using C-DTS.
6:     **if** time to consult (e.g. $current\_generation \geq 0.5 * G$ ) and $D_{KL}(\widetilde{\mathrm{Pr}}_{\tau-1}||\widetilde{\mathrm{Pr}}_\tau) > \varepsilon$ **then**
7:         ▷run clustering-based stochastic dueling bandit (Algorithm 2);
8:         $\tau \leftarrow \tau + 1$;
9:     **end if**
10:     // Phase 2: Preference elicitation and optimization.
11:     **if** $s = 0$ **then**
12:         Optimization;
13:     **else**
14:         ▷ Preference Elicitation, update $\widetilde{\mathrm{Pr}}_\tau(\mathbf{x})$;
15:         Optimization;
16:     **end if**
17: **end while**
**Output:** Solutions $\mathbf{x}_i, i \in \{1, 2, \ldots, N\}$.

---

## C.2 D-PBMOEA/D

Following [36], MOEA/D [68] is designed to use a set of evenly distributed weight vectors $W = \{\mathbf{w}^i\}_{i=1}^N$ to approximate the whole PF. The recommendation point learned from the consultation module is to adjust the distribution of weight vectors. The following four-step process is to achieve this purpose.

Step 1: Before the first consultation session, the EMO algorithm runs as usual without considering any preference information.

Step 2: If time to consult (e.g., when we have evaluated the population for $50\%$ of the total generation), the whole population is fed to consultation module and the three outputs will be recorded and used to update the predicted preference distribution for the current population as $\widetilde{\mathrm{Pr}}(\mathbf{x})$ defined in Equation (4).

Step 3: The weight vectors $W = \{\mathbf{w}^i\}_{i=1}^N$ will be projected to the preference distribution $\widetilde{\mathrm{Pr}}$ by using the weighted sum of inverse Gaussian distribution. For a specific weight vector $\mathbf{w}^i$:

$$\mathbf{w}^{i\prime} = \widetilde{\mathrm{Pr}}^{-1}(\mathbf{w}^i) = \frac{1}{Z} \sum_{\tau=1}^{N_{\mathrm{consult}}} \frac{1}{\tilde{\sigma}_\tau} \mathcal{N}^{-1}(\mathbf{w}^i \mid \tilde{\mathbf{w}}_\tau^*, \tilde{\Sigma}_\tau), \tag{19}$$

where $i = 1, 2, \ldots, N$ and $\tilde{\mathbf{w}}_\tau^*$ is the corresponding weight vector of $\tilde{\mathbf{x}}_\tau^*$ in $\widetilde{\mathrm{Pr}}(\mathbf{x})$ (Equation (4)). Output the adjusted $\mathbf{w}^{i\prime}$ as new weight vectors $W'$ for next generation.

## C.3 D-PBEMO-DTS

This section will discuss how to utilize DTS as consultation module in ablation experiment. In each consultation session, the budget for pairwise comparison is limited to $100$. The optimization module is Pareto-based EMO algorithm, i.e., NSGA-II [17]. Based on [16], solutions from the best non-domination levels are chosen front-wise as before and a modified crowding distance operator is used to choose a subset of solutions from the last front which cannot be entirely chosen to maintain the population size of the next population, the following steps are performed:

Step 1: Before the first consultation session, the NSGA-II runs as usual without considering the preference information.

Step 2: If it is time to consult (e.g., when we have evaluated the population for $50\%$ of the total generation), then the whole population will be fed into the DTS and the optimal arm will be

**Algorithm 2** Clustering-based Stochastic Dueling Bandit (`C-DTS`)
___

**Input:** $T \in \{1, 2, \ldots\} \cup \{\infty\}$, $N$ arms, $K$ subset.

1: $\mathbf{v} \leftarrow \mathbf{0}_N, \boldsymbol{\ell} \leftarrow \mathbf{0}_N$;
2: $B \leftarrow \mathbf{0}_{K \times K}$, $u_{i,j} = 1, l_{i,j} = 0$ for $i, j \in \{1, \ldots, K\}$;
3: **for** $t = 1, \ldots, T$ **do**
4:     *// Step 1: Choose the first candidate subset $\tilde{\mathcal{S}}'$*
5:     $u_{i,j} = \frac{b_{i,j}}{b_{i,j} + b_{j,i}} + \sqrt{\alpha \log t / (b_{i,j} + b_{j,i})}$, $l_{i,j} = \frac{b_{i,j}}{b_{i,j} + b_{j,i}} - \sqrt{\alpha \log t / (b_{i,j} + b_{j,i})}$, if $i \neq j$,
    and $u_{i,i} = l_{i,i} = 1/2, \forall i$; *// let $x/0 := 1$ for any $x$, when $b_{i,j} + b_{j,i} = 0$.*
6:     $\tilde{\zeta}_i \leftarrow 1/(K-1) \sum_{j \neq i} \mathbb{I}(u_{i,j} > 1/2)$; *// Upper bound of the normalized Copeland score*
7:     $\mathcal{C}^1 \leftarrow \{i : \tilde{\zeta}_i = \max_i \tilde{\zeta}_i\}$;
8:     **for** $i, j = 1, \ldots, K$ with $i < j$ **do**
9:         Sample $\theta_{i,j}^{(1)} \sim \text{Beta}(b_{i,j} + 1, b_{j,i} + 1)$;
10:        $\theta_{j,i}^{(1)} \leftarrow 1 - \theta_{i,j}^{(1)}$;
11:     **end for**
12:     $\tilde{\mathcal{S}}' \leftarrow \arg\max_{i \in \mathcal{C}^1} \sum_{j \neq i} \mathbb{I}(\theta_{i,j}^{(1)} > 1/2)$; *//Choosing from $\mathcal{C}^1$ to eliminate likely non-winner arms; Ties are broken randomly.*
13:     *//Step 2: Choose the second candidate subset $\tilde{\mathcal{S}}''$*
14:     $\mathcal{C}^2 \leftarrow \{\tilde{\mathcal{S}}^i | l_{i,\prime} \leq 0.5\}$;
15:     Sample $\theta_{i,\prime}^{(2)} \sim \text{Beta}(b_{i,\prime} + 1, b_{\prime,i} + 1)$ for all $i \neq \tilde{\mathcal{S}}'$, and let $\theta_{\prime,\prime}^{(2)} = 1/2$;
16:     $\tilde{\mathcal{S}}'' \leftarrow \text{argmax}_{\tilde{\mathcal{S}}^i \in \mathcal{C}^2} \theta_{i,\prime}^{(2)}$;
17:     *// Step 3: Query and update*
18:     Randomly select an arm $\mathbf{x}'$ from $\tilde{\mathcal{S}}'$, and $\mathbf{x}''$ from $\tilde{\mathcal{S}}''$. And observe the result.
19:     **if** $\mathbf{x}' \succ_{\text{p}} \mathbf{x}''$ **then**
20:        $b_{\prime,\prime\prime} \leftarrow b_{\prime,\prime\prime} + 1$ and $v_\prime \leftarrow v_\prime + 1, \ell_{\prime\prime} \leftarrow \ell_{\prime\prime} + 1$;
21:     **else**
22:        $b_{\prime\prime,\prime} \leftarrow b_{\prime\prime,\prime} + 1$ and $v_{\prime\prime} \leftarrow v_{\prime\prime} + 1, \ell_\prime \leftarrow \ell_\prime + 1$;
23:     **end if**
24: **end for**
**Output:** $\{\tilde{\mathcal{S}}^\star, \mathbf{v}, \boldsymbol{\ell}\}$, where $\tilde{\mathcal{S}}^\star = \text{argmax}_{\tilde{\mathcal{S}}^i} \tilde{\zeta}_i$
___

    recorded and used to update the predicted preference distribution $\widetilde{\Pr}(\mathbf{x})$ as Equation (4) for the current population.

Step 3: Between two interactions, the crowding distance of each solution will be evaluated by the predicted preference distribution $\widetilde{\Pr}(\mathbf{x})$ learned from the last consultation session.

## C.4 `D-PBEMO-PBO`

In this section, PBO [25] is utilized as the consultation module for `D-PBEMO`. The PBO is initialized with 4 pairwise comparisons and its query budget is limited to 10. Also the acquisition function is set to Thompson sampling. The optimization module is chosen to be decomposition-based EMO algorithm, i.e., MOEA/D [68]. Following [36], the decomposition-based EMO algorithm (e.g., MOEA/D [68]) is designed to use a set of evenly distributed weight vectors $W = \{\mathbf{w}^i\}_{i=1}^N$ to approximate the whole PF. The recommendation point learned from the consultation module is to adjust the distribution of weight vectors. The following four-step process is to achieve this purpose.

Step 1: Before the first consultation session, the EMO algorithm runs as usual without considering any preference information.

Step 2: If time to consult (e.g., when we have evaluated the population for 50% of the total generation), the whole population is fed to consultation module and the recommendation solution will be recorded and used to update the predicted preference distribution for the current population $\widetilde{\Pr}(\mathbf{x})$ as Equation (4).

Step 4: The weight vectors $W = \{\mathbf{w}^i\}_{i=1}^N$ will be projected to the preference distribution $\widetilde{\Pr}$ by using the weighted sum of inverse Gaussian distribution as defined in Equation (19). The adjusted weight vectors $W' = \{\mathbf{w}^{i\prime}\}_{i=1}^N$ will be the new weight vectors for next generation.

We present the lookup table for key notations in the following table [2].

# D    Experimental Settings

## D.1    Benchmark Test Problems

This section introduces the problem definitions of two real-world scientific discovery problems considered in our experiments. Further, we summarize the key parameter settings in our experiments.

### D.1.1    Inverse RNA design

An RNA sequence $\mathbf{x}$ of length $n$ is specified as a string of base nucleotides $x_1 x_2 \ldots x_n$ where $x_i \in \{A, C, G, U\}$ for $i = 1, 2, \ldots, n$. A secondary structure $\mathcal{P}$ for $\mathbf{x}$ is a set of paired indices where each pair $(i, j) \in \mathcal{P}$ indicates two distinct bases $x_i x_j \in \{CG, GC, AU, UA, GU, UG\}$ and each index from 1 to $n$ can only be paired once. For example, given a target secondary structure "$(...)$", the predicted sequence $x_1 x_2 x_3 x_4 x_5$ should satisfied that the $1^{st}$ and $5^{th}$ nucleotides in paired set $\{CG, GC, AU, UA, GU, UG\}$.

Reference from [55, 70], our inverse RNA design problem adopts two objective functions.

**Stability**    The ensemble of an RNA sequence $\mathbf{x}$ is the set of all secondary structures that $\mathbf{x}$ can possibly fold into, denoted as $\mathcal{Y}(\mathbf{x})$. The free energy $\Delta G(\mathbf{x}, \mathbf{y})$ is used to characterize the stability of $\mathbf{y} \in \mathcal{Y}(\mathbf{x})$. The lower the free energy $\Delta G(\mathbf{x}, \mathbf{y})$, the more stable the secondary structure $\mathbf{y}$ for $\mathbf{x}$. The structure with the minimum free energy (MFE) is the most stable structure in the ensemble, i.e., MFE structure.

$$f_1 = MFE(\mathbf{x}) = \underset{\mathbf{y} \in \mathcal{Y}}{\operatorname{argmin}} \Delta G(\mathbf{x}, \mathbf{y}). \tag{20}$$

Note that ties for $\operatorname{argmin}$ are broken arbitrarily, thus there could be multiple MFE structures for given $\mathbf{x}$. Technically, $MFE(\mathbf{x})$ should be a set. In our experiment, the MFE is calculate by ViennaRNA[3] package.

**Similarity**    The second objective function is to assess the similarity between the best secondary structure of our predicted sequence and the target structure.

$$\sigma = (n - d)/n, \tag{21}$$

where $d$ is the Hamming distance between our predicted structure and target structure, and $\sigma \in [0, 1]$. For example, when the target structure is "$(...)$" and the predicted secondary structure is "$.....$", $d = 2$ and $\sigma = (5 - 2)/5 = 0.6$. The bigger the similarity $\sigma$, the more precise our predicted structure is. Since our MO problems are minimization problems, the second objective should be:

$$f_2 = 1 - \sigma = d/n. \tag{22}$$

### D.1.2    Protein Structure Prediction

Protein structure prediction (PSP) is the inference of the three-dimensional structure of a protein from its amino acid sequence — that is, the prediction of its secondary and tertiary structure from primary structure.

To computationally address PSP, the initial step involves constructing protein conformations using a suitable model. Traditional Protein Data Bank files contain Cartesian coordinates of all atoms in a protein, often totaling over 1000 atoms for a modest 70-amino acid sequence. This abundance of coordinates renders Protein Backbone Exploration with EMO algorithms nearly impracticable. To mitigate this, protein structures are represented using torsion and dihedral angles, obtained by

---

[2]In Table A1, we use the $i$-th arm to denote the $i$-th subset in the context of PBEMO.
[3]https://github.com/ViennaRNA/ViennaRNA

converting atom coordinates along consecutive bonds, effectively reducing the search space while retaining essential structural information. Protein backbones are primarily defined by three torsion angles: $\Phi$ (around the $-N-C_\alpha-$ bond), $\Psi$ (around the $-C_\alpha-C-$ bond), and $\Omega$ (around the $-C-N-$ peptide bond). Additionally, each amino acid residue may feature varying numbers of side-chain torsion angles ($\chi_i$, $i \in 1,2,3,4$) depending on its structure. Despite this, the conformational search space remains extensive. To address this, secondary structures and backbone-independent rotamer libraries [21] are utilized to further constrain backbone and side-chain angles, respectively. These libraries also integrate biological insights from protein secondary structures. Moreover, this approach necessitates less optimization space compared to those relying on contact maps. For instance, reconstructing the main and side chains of a 50-amino acid sequence requires only $250-300$ angles using dihedral angles, whereas contact maps demand 2500 distances for the main chain alone.

In this paper, we consider PSP tasks as a four-objective optimization problem. Each objective function is an energy function that determines a regulatory property of a protein.

**CHARMM** The first objective function $f_1$ is CHARMM that calculates several energy terms that determine protein properties. The corresponding energy function is formulated as:

$$
\begin{aligned}
f_1 = E_C = &\sum_{bounds} k_b(b-b_0)^2 + \sum_{angles} k_\theta(\theta-\theta_0)^2 + \\
&\sum_{dihedrals} k_\phi[1+\cos(n\phi-\delta)] + \\
&\sum_{improper} k_\omega(\omega-\omega_0)^2 + \sum_{Urey-Bradly} k_u(u-u_0)^2 + \\
&\sum_{Van-der-Vaals} \varepsilon_{ij}[(\frac{R_{ij}}{r_{ij}})^{12} - (\frac{R_{ij}}{r_{ij}})^6] + \sum_{charge} \frac{q_i q_j}{e \cdot r_{ij}},
\end{aligned}
\tag{23}
$$

where $b$ is the bound length, $b_0$ is the ideal length, $\theta$ is the angle formed by three atoms involved in two connected bonds, $\phi$ is the torsion angle, $\omega$ is the improper angle, $u$ is the distance between two atoms of nonbonded interactions in an angle, $k_b, k_\theta, k_\phi, \delta, n, k_\omega, \omega_0, k_u, u_0$ are constants. The bonds, angles dihedrals, improper, and Urey-Bradley terms belong to the bond term, while the non-bond term includes Van-der-Waals and charge terms. Apart from these, Van-der-Waals and charge terms are used to calculate the Van der Waals force and electrostatic energy between a pair of atoms $(i,j)$, respectively.

**dDFIRE** The second one $f_2$ is dDFIRE that follows two interactions of atom pairs, i.e., that between polar and nonpolar atoms, and that between polar and non-hydrogen-bonded polar atoms, to get excellent solution for potential. The energy of atoms $(p,q)$ pair is calculated as:

$$
\bar\mu_D(r,\theta_p,\theta_q,\theta_{pq}) = \begin{cases} -RT\ln \frac{N_o(\theta_p,\theta_q,\theta_p q,r)}{(\frac{r}{r_{cut}})^\alpha \frac{\Delta r}{\Delta r_{cut}} N_o(\theta_p,\theta_q,\theta_{pq},r_{cut})}, & r < r_{cut}, \\ 0, & r \geq r_{cut}, \end{cases}
\tag{24}
$$

where $N_o(\theta_p,\theta_q,\theta_p q,r)$ is the number of polar atom pair $(p,q)$ at distance $r$. $\theta_p$, $\theta_q$ and $\theta_{pq}$ are orientation angles of polar atoms. $R$ is the gas constant, temperature $T$ is set as $300\ K$, $r$ is the distance between an atom pair, the cutoff distance $r_{cut}$ is set to $14.5\mathring{A}$, and $\Delta r(\Delta r_{cut})$ is the bin width at $r$ $(r_{cut})$. The value of parameter $\alpha$ is proven to be $1.51$ [64]. The total dDFIRE potential is the summation of energy of all possible atom pairs $(p,q)$ in the protein structure:

$$
f_2 = E_D = \sum_{r_{pq},\theta_p,\theta_q,\theta_{pq}} \bar\mu_D(r_{pq},\theta_p,\theta_q,\theta_{pq}).
\tag{25}
$$

**Rosetta** The third objective function $f_3$ is Rosetta, a composite energy energy function based on physics and knowledge, each potentials of it is intricately designed:

$$
f_3 = E_R = \sum_i w_i E_i(\Theta_i, P),
\tag{26}
$$

where w and $\Theta$ are the weight and degree of freedom of each energy term, respectively. $P$ is the protein structure. Details on Rosetta can be referenced in [3].

**RWplus** The last objective function $f_4$ is RWplus, whose calculation is the same as that of dDFIRE. However, it emphasizes more about the potential of short-range interactions:

$$f_4 = E_{RW} = \sum_{\alpha,\beta} -kT \ln \frac{N_o(\alpha,\beta,R)}{N_e(\alpha,\beta,R)} + \sum_{A,B} -0.1\delta(A,B)kT \ln \frac{N_o(A,B,O_{AB})}{N_e(A,B,O_{AB})}, \quad (27)$$

where $k$ is the Boltzmann constant and $T$ is the temperature in units of Kelvin. $N_o(\alpha,\beta,R)$ and $N_e(\alpha,\beta,R)$ are the observed and expected number of atom pairs, with $\alpha$ and $\beta$ atom types within a distance $R$, respectively. Similarly, $N_o(A,B,O_{AB})$ and $N_e(A,B,O_{AB})$ are the observed and expected number of atom vector pairs with $(A,B)$ type within a relative orientation $O_{AB}$ respectively. The type of atom vector pairs between atom types $\alpha$ and $\beta$ is $(A,B)$. $\delta(A,B)$ is 0 when vector pairs $A$ and $B$ are not contact, and 1 vice versa. The total RWplus potential is the sum of these atom pairs' energies.

The problem formulation of PSP can be referenced in [69] [4].

## D.2 Parameter Setting

This section lists several parameters used in experiments, including the parameters in EMO algorithms, and other parameters we need in `D-PBEMO`:

- The probability and distribution of index for SBX: $p_c = 1.0$ and $\eta_c = 20$;
- The mutation probability and distribution of index for polynomial mutation operator: $p_m = \frac{1}{m}$ and $\eta_m = 20$;
- The population size for different problems can be referenced in Table A2;
- The maximum number of generation $G$ can be referenced in Table A3;
- For `I-MOEA/D-PLVF` and `I-NSGA2/LTR`, the number of incumbent candidate presented to decison maker (DM) for consultation: $\mu = 10$;
- For `I-MOEA/D-PLVF` and `I-NSGA2/LTR`, there exists the number of consecutive consultation session $\tau$: $\tau = 25$;
- The step size of reference point update $\eta$ for MOEA/D-series PBEMO algorithms are set to: $\eta = 0.3$.
- The reference points in different test problems can be referenced in Table A4. The reference point of real-world problems will be introduced in their experiment results.
- While the winning probability is usually known in advance in dueling bandits, it however does not exist in the context of MO. In this paper, to project a MO solution to an arm in the bandit setting, we define the winning probability $p_{i,j}$ of a pair of candidate solutions $\langle \mathbf{x}_i, \mathbf{x}_j \rangle, \forall i,j \in \{1,\ldots,N\}$ as [20]:

$$p_{i,j} = \mu(\Pr(\mathbf{x}_i) - \Pr(\mathbf{x}_j)), \quad (28)$$

where $\mu(a) = 1/(1 + \exp(-a))$, $\Pr$ denotes the real preference distribution of each arm. For example, we set $\Pr(\mathbf{x}) = \mathcal{N}(\mathbf{x}|\mathbf{x}^\star, \sigma^{\star 2})$, where $\mathbf{x}^\star$ is decision vector of the reference point $\mathbf{z}^*$ (Table A4) and $\sigma^\star$ is a constant value initially set as $0.1$. Note that $p_{i,j}$ can take other forms in case it satisfies the conditions in Section 2.2.

- The subset number $K$:

$$K = \begin{cases} 10 & \text{if } m = 2, \\ 8 & \text{if } m = 3, \\ 12 & \text{if } m = 5, \\ 14 & \text{if } m = 8, \\ 18 & \text{if } m = 10. \end{cases}$$

- The budget for dueling bandits algorithm $T$ is set as 100.
- The parameter $\alpha$ in $u_{i,j}$ and $l_{i,j}$ is set as $\alpha = 0.6$.

---

[4]https://github.com/zhangzm0128/PCM-Protein-Structure-Prediction

### D.3 Statistical Test

**Wilcoxon rank-sum test**  To offer a statistical interpretation of the significance of comparison results, we conduct each experiment 20 times. To analyze the data, we employ the Wilcoxon signed-rank test [61] in our empirical study.

The Wilcoxon signed-rank test, a non-parametric statistical test, is utilized to assess the significance of our findings. The test is advantageous as it makes minimal assumptions about the data's underlying distribution. It has been widely recommended in empirical studies within the EA community [19]. In our experiment, we have set the significance level to $p = 0.05$.

**Scott-Knott rank test**  Instead of merely comparing the raw $\epsilon^\star(\mathcal{S})$ and $\bar{\epsilon}(\mathcal{S})$ values, we apply the Scott-Knott test [45] to rank the performance of different peer techniques over 31 runs on each experiment. In a nutshell, the Scott-Knott test uses a statistical test and effect size to divide the performance of peer algorithms into several clusters. In particular, the performance of peer algorithms within the same cluster is statistically equivalent. Note that the clustering process terminates until no split can be made. Finally, each cluster can be assigned a rank according to the mean $\epsilon^\star(\mathcal{S})$ or $\bar{\epsilon}(\mathcal{S})$ values achieved by the peer algorithms within the cluster. In particular, since a smaller $\epsilon^\star(\mathcal{S})$ and $\bar{\epsilon}(\mathcal{S})$ value is preferred, the smaller the rank is, the better performance of the technique achieves.

## E  Experiment on Test Problem Suite

### E.1  Population Results

In this section, we show the results of our proposed method running on ZDT, DTLZ, and WFG test suites.

From the selected plots of the final non-dominated solutions obtained by different algorithms shown in Figure A7 (full results are in Figures A8 to A9), we can see that the solutions found by our `D-PBEMO` instances are more concentrated on the 'golden solution' while the others are more scattered. Furthermore, the superiority of our proposed `D-PBEMO` algorithm instances become more evident when tackling problems with many objectives, i.e., $m \geq 3$.

The population results of `D-PBEMO` running on ZDT1∼ZDT4, and ZDT6 are shown in Figure A8. The population results running on DTLZ1∼DTLZ4 ($m = \{3, 5, 8, 10\}$) are shown in Figure A10 Figure A11 Figure A12 Figure A13 respectively. The results running on WFG ($m = 3$) are shown in Figure A9.

The performance metrics are shown in Table A5 and Table A6.

### E.2  Parameter influence

In this section, we testify the influence of different hyperparameters, including KL threshold $\varepsilon$, and subset number $K$.

We set $\varepsilon$ in `D-PBEMO-MOEA/D` to three different values, $\varepsilon = \{10^{-1}, 10^{-3}, 10^{-6}\}$. The experiment results of $\epsilon^\star(\mathcal{S})$ and $\bar{\epsilon}(\mathcal{S})$ can be referenced in A9 and A10.

We set $K$ in `D-PBEMO-MOEA/D` to three different values running on ZDT test problems ($N = 100$), $K = \{2, 5, 10\}$. The experiment results of $\epsilon^\star(\mathcal{S})$ and $\bar{\epsilon}(\mathcal{S})$ can be referenced in A11, A13, A15, A17, A19, A12, A14, A16, A18, and A20.

## F  Experiments on Scientific Discovery Problems

### F.1  Experiment on Inverse RNA Design

This section shows the experiment result of our proposed method, specifically `D-PBEMO-NSGA-II`. Our method and peer algorithms run on Eterna100-vienna1 benchmark[5], which contains 100 RNA sequences. We selectively run our method on 10 short sequences $n \in [12, 36]$. For inverse RNA

---

[5]https://github.com/eternagame/eterna100-benchmarking

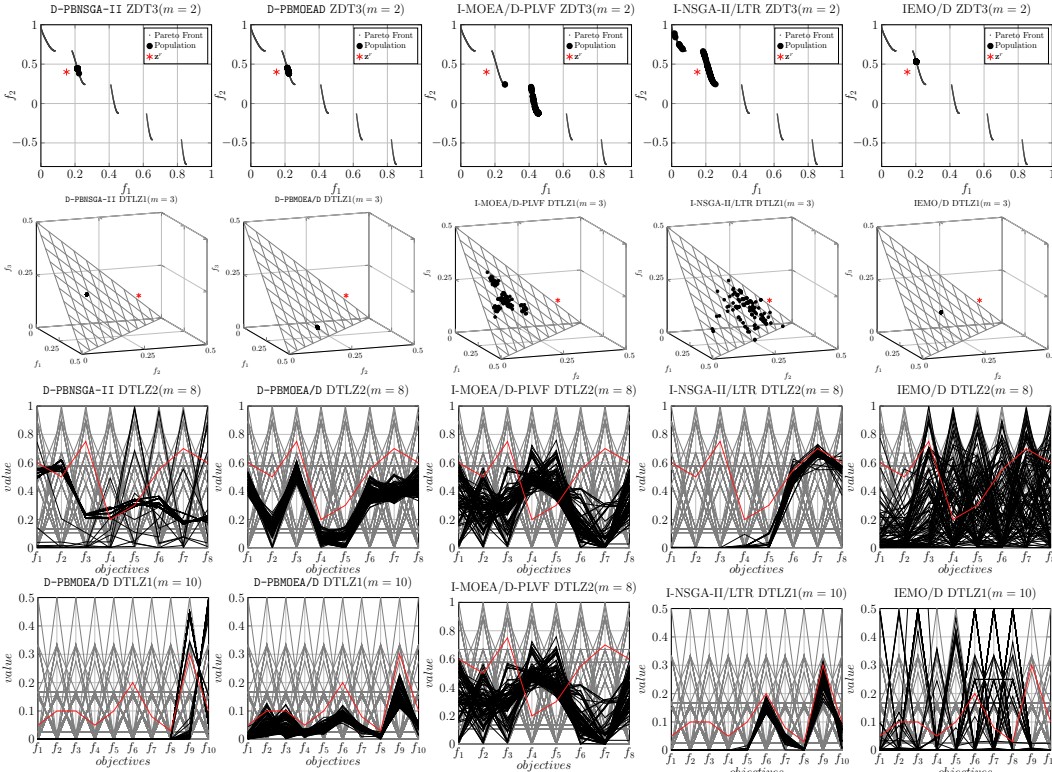

**Figure A7:** Selected plots of the final populations of a `D-PBEMO` instance and peer algorithms. The red star (∗) or line (—) represents reference point, the gray lines (—) represent PF, and the black points (●) or lines (—) represent the non-dominated solutions obtained by an algorithm.

design problems, the most important task is to predict an accurate sequence whose secondary structure is the same as target structure, $f_2 = 0$. However, in real-world scenarios, we need to tradE-off between stability and similarity. When $f_2 = 0$, the stability $f_1$ can not reach its global optimum in PF. Our `D-PBEMO` framework can do more than those singlE-objective algorithms for RNA. Users can guide the search of `D-PBEMO` algorithms and finally reach a point which sacrifices a little similarity but has better stability. So our experiment are divided into two parts. In the first part, `D-PBEMO-NSGA-II` only focuses on finding the most similar solutions, whose reference point locates on $f_2 = 0$. In another part, user prefers a solution in the middle of PF, $f_2 \in (0, 1)$. The reference points of our two-session experiments can be referenced in A21.

The experiment results of running on reference point 1 are listed in A22 and A23. The predicted secondary structures of each algorithm are shown in A14 and A15. Each algorithm repeatedly runs for 10 times. As we can see from these two tables, `D-PBEMO-NSGA-II` has best performance for 5 times from the perspective of $\epsilon^\star(\mathcal{S})$ while 6 times from $\bar{\epsilon}(\mathcal{S})$.

The experiment results of running on reference point 2 are listed in A24 and A25. As we can see from these two tables, `D-PBEMO-NSGA-II` has best performance for 8 times with reference to $\epsilon^\star(\mathcal{S})$ and $\bar{\epsilon}(\mathcal{S})$.

### F.2 Experiment on PSP

In this section we listed the results implementing our proposed method, specifically `D-PBEMO-NSGA-II` on PSP problems. We implement RMSD as the performance metric for PSP problems:

$$RMSD = \sqrt{\frac{\sum_{i=1}^{n_{atom}} d_i^2}{n_{atom}}} \tag{29}$$

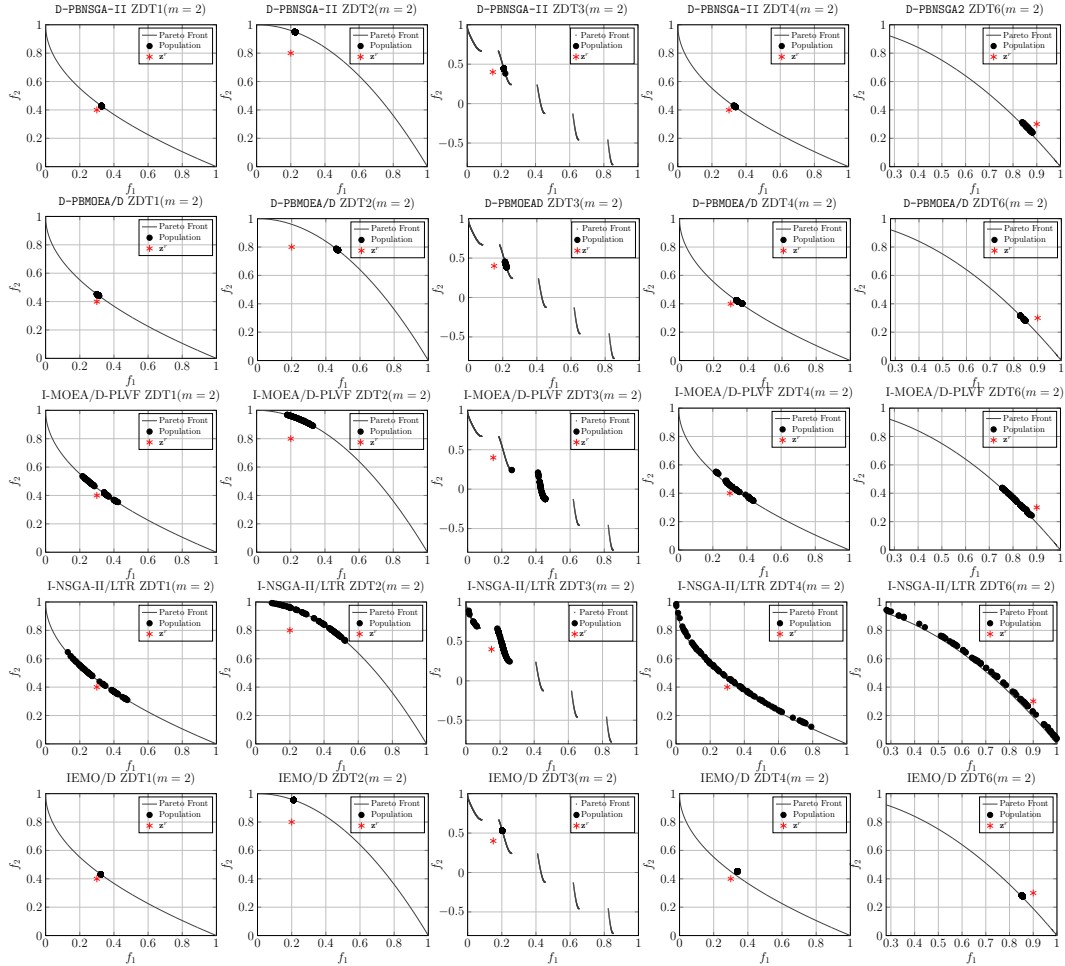

**Figure A8:** The population distributions of D-PBEMO and peer algorithms running on ZDT test suite ($m = 2$).

where $n_{atom}$ is the total number of matched atoms between the two protein structures and $d_i$ is the distance between each pair of atoms. The four energy settings and predicted results of our proposed method are in Table A26. Other parameters in PSP problems are align with [69].

The population results are shown in Figure A16 and the secondary structures are dipicted in Figure 6. The RMSD comparison results are shown in Table A27. As we can see our proposed method have better convergence and accuracy than synthetic problems. This may be caused by two reasons:

- The first one is the PSP problem is only conducted on 4-dimensional objective spaces. In synthetic problems, our proposed method shows better performance results when dimension $m \geq 3$ while peer algorithms may collapse.

- The second reason is the formulation of PSP problems. In this paper, we adopt utilizing 4 energy function to represent, which are empirically proved to be more accurate than in 1-dimensional objective function[69].

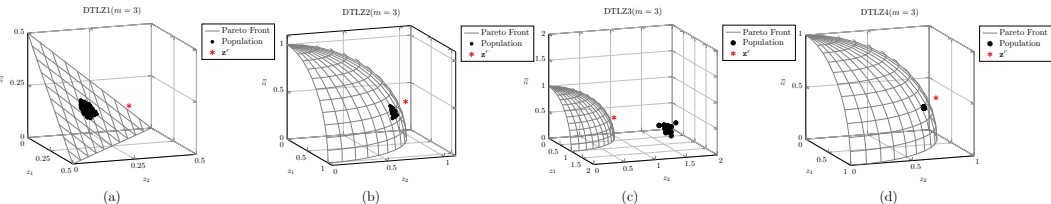

**Figure A9:** The population distribution of `D-PBEMO` algorithms and peer algorithms running on WFG test suite ($m = 3$).

**Figure A10:** The population distribution of our proposed method (i.e., `D-PBMOEA/D`) running on DTLZ test suite ($m = 3$).

**Table A1:** Lookup Table for Key Notations

| Notation | Dimension | Description |
|---|---|---|
| $\mathbf{x}$ | $\mathbb{R}^n$ | $n$-dimensional decision vector |
| $x_i$ | $\mathbb{R}$ | the $i$-th decision variable |
| $F$ | $\mathbb{R}^m$ | $m$-dimensional objective vector |
| $f_i$ | $\mathbb{R}$ | the $i$-th objective function |
| $P$ | $\mathbb{R}^{K \times K}$ | preference matrix of $K$ arms |
| $p_{i,j}$ | $[0,1]$ | winning probability of the $i$-th arm over the $j$-th arm |
| $\zeta_i$ | $[0,1]$ | the normalized Copeland score of the $i$-th arm |
| $k^*$ | $\mathbb{N}^K$ | the Copeland winner in $K$ arms |
| $B$ | $\mathbb{R}^{K \times K}$ | constructed winning matrix of $K$ arms in dueling bandits algorithm |
| $b_{i,j}$ | $[0,1]$ | number of time-slots when the $i$-th arm is preferred from the pair of the $i$-th and the $j$-th arms |
| $\tilde{p}_{i,j}$ | $[0,1]$ | approximated preference probability of the $i$-th arm over the $j$-th arm |
| $\tilde{u}_{i,j}$ | $[0,1]$ | upper confidence bound of $\tilde{p}_{i,j}$ |
| $\tilde{l}_{i,j}$ | $[0,1]$ | lower confidence bound of $\tilde{p}_{i,j}$ |
| $\alpha$ | $\mathbb{R}^+$ | parameter for confidence level |
| $t$ | $\mathbb{N}^+$ | total number of comparisons so far |
| $R_T$ | $\mathbb{R}^+$ | expected cumulative regret |
| $T$ | $\mathbb{N}^+$ | maximum number of comparisons |
| $P_s$ | $\mathbb{R}^{N \times N}$ | solution-level preference matrix |
| $p_{i,j}^s$ | $[0,1]$ | winning probability of $\mathbf{x}^i$ is preferred over $\mathbf{x}^j$ |
| $\mathbf{v}$ | $\mathbb{N}^N$ | winning time vector of population of solutions |
| $v_i$ | $\mathbb{N}$ | winning times of the $i$-th solution |
| $\boldsymbol{\ell}$ | $\mathbb{N}^N$ | losing time vector of population of solutions |
| $\ell_i$ | $\mathbb{N}$ | losing times of the $i$-th solution |
| $\tilde{\zeta}_i$ | $[0,1]$ | upper confidence Copeland score of the $i$-th solution |
| $\theta_{i,j}^{(1)}$ | $[0,1]$ | winning probability of $\tilde{\mathcal{S}}^i$ subsets $\tilde{\mathcal{S}}^j$ sampled from Thompson sampling |
| $\theta_{i,'}^{(2)}$ | $[0,1]$ | winning probability of $\tilde{\mathcal{S}}^i$ over $\tilde{\mathcal{S}}'$ sampled from Thompson sampling |
| $\Omega$ | $\mathbb{R}^n$ | feasible set in the decision space |
| $\mathcal{S}$ | finite set | population of non-dominated solutions |
| $\tilde{\mathcal{S}}^i$ | finite set | the $i$-th subset of population of non-dominated solutions |
| $\mathcal{C}^1$ | finite set | subset of population of non-dominated solutions with the highest upper confidence Copeland score |
| $\tilde{\mathcal{S}}'$ | finite set | subset most likely covers the SOI |
| $\mathcal{C}^2$ | finite set | subset of population of non-dominated solutions with lower-confident winning probability over $\tilde{\mathcal{S}}'$ |
| $\tilde{\mathcal{S}}''$ | finite set | subset potentially preferred over $\tilde{\mathcal{S}}'$ |

**Table A2:** Population size in different problems

| Problem | $m$ | $N$ |
|---|---|---|
| ZDT | 2 | 100 |
| DTLZ | 3 | 64 |
| | 5 | 128 |
| | 8 | 224 |
| | 10 | 288 |
| WFG | 3 | 64 |
| PSP | 4 | 50 |
| Inverse RNA | 2 | 100 |

**Table A3:** Maximum generation in different problems

| Problem | G |
|---|---|
| ZDT | 250 |
| DTLZ1 | $500 + 50(m - 2)$ |
| DTLZ2 | $200 + 50(m - 2)$ |
| DTLZ3 | $1000 + 50(m - 2)$ |
| DTLZ4 | $200 + 50(m - 2)$ |
| WFG | $1000 + 50(m - 2)$ |
| PSP | 300 |
| Inverse RNA | 250 |

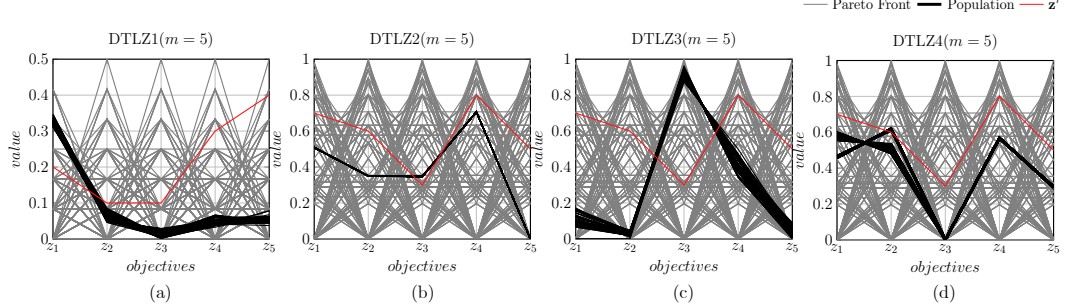

**Figure A11:** The population distribution of our proposed method (i.e., D-PBMOEA/D) running on DTLZ test suite ($m = 5$)

.

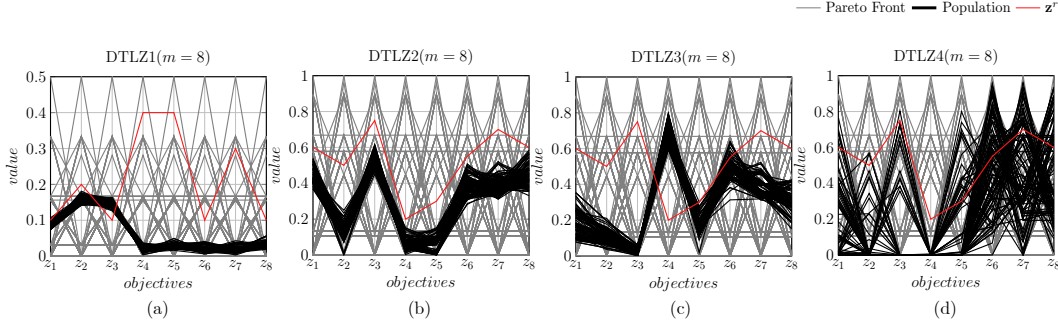

**Figure A12:** The population distribution of our proposed method (i.e., D-PBMOEA/D) running on DTLZ test suite ($m = 8$).

**Table A4:** The settings of the 'golden solution' $\mathbf{z}^*$ for different benchmark problems (represented in the objective space).

| problem | $m$ | reference point |
|---|---|---|
| ZDT1 | 2 | $(0.3, 0.4)^\top$ |
| ZDT2 | 2 | $(0.2, 0.8)^\top$ |
| ZDT3 | 2 | $(0.15, 0.4)^\top$ |
| ZDT4 | 2 | $(0.3, 0.4)^\top$ |
| ZDT6 | 2 | $(0.9, 0.3)^\top$ |
| WFG1 | 3 | $(0.2, 0.5, 0.6)^\top$ |
| WFG3 | 3 | $(0.6, 0.8, 0.8)^\top$ |
| WFG5 | 3 | $(0.3, 0.7, 0.3)^\top$ |
| WFG7 | 3 | $(0.7, 0.4, 0.4)^\top$ |
| | 3 | $(0.3, 0.3, 0.2)^\top$ |
| | 5 | $(0.2, 0.1, 0.1, 0.3, 0.4)^\top$ |
| DTLZ1 | 8 | $(0.1, 0.2, 0.1, 0.4, 0.4, 0.1, 0.3, 0.1)^\top$ |
| | 10 | $(0.02, 0.01, 0.06, 0.04, 0.01, 0.02, 0.03, 0.05, 0.08)^\top$ |
| | 3 | $(0.7, 0.8, 0.5)^\top$ |
| | 5 | $(0.7, 0.6, 0.3, 0.8, 0.5)^\top$ |
| DTLZ2 | 8 | $(0.6, 0.5, 0.75, 0.2, 0.3, 0.55, 0.7, 0.6)^\top$ |
| | 10 | $(0.3, 0.3, 0.3, 0.1, 0.3, 0.55, 0.35, 0.35, 0.25, 0.45)^\top$ |
| | 3 | $(0.7, 0.8, 0.5)^\top$ |
| | 5 | $(0.7, 0.6, 0.3, 0.8, 0.5)^\top$ |
| DTLZ3 | 8 | $(0.6, 0.5, 0.75, 0.2, 0.3, 0.55, 0.7, 0.6)^\top$ |
| | 10 | $(0.3, 0.3, 0.3, 0.1, 0.3, 0.55, 0.35, 0.35, 0.25, 0.45)^\top$ |
| | 3 | $(0.7, 0.8, 0.5)^\top$ |
| | 5 | $(0.7, 0.6, 0.3, 0.8, 0.5)^\top$ |
| DTLZ4 | 8 | $(0.6, 0.5, 0.75, 0.2, 0.3, 0.55, 0.7, 0.6)^\top$ |
| | 10 | $(0.3, 0.3, 0.3, 0.1, 0.3, 0.55, 0.35, 0.35, 0.25, 0.45)^\top$ |
| | 3 | $(0.2, 0.3, 0.6)^\top$ |
| | 5 | $(0.12, 0.12, 0.17, 0.24, 0.7)^\top$ |
| DTLZ5 | 8 | $(0.04, 0.04, 0.0566, 0.8, 0.113, 0.16, 0.2263, 0.68)^\top$ |
| | 10 | $(0, 0, 0, 0, 0.0096, 0.027, 0.082, 0.25, 0.75, 0.08)^\top$ |
| | 3 | $(0.2, 0.3, 0.6)^\top$ |
| | 5 | $(0.12, 0.12, 0.17, 0.24, 0.7)^\top$ |
| DTLZ6 | 8 | $(0.04, 0.04, 0.0566, 0.8, 0.113, 0.16, 0.2263, 0.68)^\top$ |
| | 10 | $(0, 0, 0, 0, 0.0096, 0.027, 0.082, 0.25, 0.75, 0.08)^\top$ |

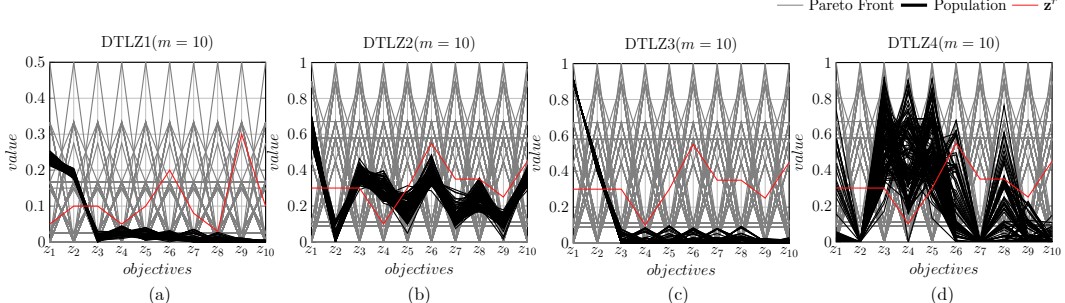

**Figure A13:** The population distribution of our proposed method (i.e., D-PBMOEA/D) running on DTLZ test suite ($m = 10$).

**Table A5:** The mean(std) of $\epsilon^\star(\mathcal{S})$ obtained by our proposed `D-PBEMO` algorithm instances against the peer algorithms.

| PROBLEM | $m$ | D-PBNSGA-II | D-PBMOEA/D | I-MOEA/D-PLVF | I-NSGA-II/LTR | IEMO/D |
|---------|-----|-------------|------------|---------------|---------------|--------|
| ZDT1 | 2 | 0.039(2.40E-7) | 0.067(1.85E-3) | 0.054(1.23E-3)† | 0.064(5.99E-4)† | 0.066(8.83E-3) |
| ZDT2 | 2 | 0.172(1.71E-3)‡ | 0.208(1.26E-3)‡ | 0.274(1.07E-2) | 0.121(4.35E-4) | 0.294(5.59E-2) |
| ZDT3 | 2 | 0.086(5.87E-4)‡ | 0.098(2.17E-3)‡ | 0.362(4.32E-2) | 0.05(4.11E-4) | 0.117(6.93E-4) |
| ZDT4 | 2 | 0.141(1.91E-2) | 0.114(7.88E-3) | 0.071(3.93E-3) | 0.078(5.34E-3) | 0.086(5.37E-3) |
| ZDT6 | 2 | 0.046(1.76E-6) | 0.055(2.16E-5) | 0.106(1.07E-2)† | 0.048(7.62E-5) | 0.052(5.77E-6)† |
| WFG1 | 3 | 2.24(6.44E-4)‡ | 2.36(3.17E-1)‡ | 2.02(3.81E-3) | 2.2(2.30E-2) | 2.48(9.35E-2) |
| WFG3 | 3 | 1.7(1.17E-1)‡ | 1.01(4.48E-2)‡ | 0.658(1.03E-3) | 0.835(2.09E-3) | 0.743(6.97E-5) |
| WFG5 | 3 | 1.29(3.74E-1) | 1.62(1.23e+00) | 2.24(2.27E-1)† | 1.59(1.86E-2)† | 2.64(2.28E-4)† |
| WFG7 | 3 | 1.1(3.78E-1) | 1.79(1.58E-1) | 2.12(3.19E-1)† | 1.5(4.39E-4) | 1.49(4.53E-3)† |
| | 3 | 0.144(1.48E-3) | 0.194(3.11E-4) | 0.171(1.04E-3)† | 0.178(4.41E-4)† | 0.171(1.11E-4)† |
| | 5 | 0.417(2.20E-2)‡ | 0.318(3.99E-4)‡ | 0.25(1.69E-2) | 0.198(4.92E-3) | 0.302(3.38E-5) |
| DTLZ1 | 8 | 0.526(7.62E-1)‡ | 0.512(2.81E-4)‡ | 0.364(3.34E-2) | 0.367(1.78E-2) | 0.473(7.06E-5) |
| | 10 | 0.5(4.91E-3)‡ | 0.25(6.59E-4)‡ | 0.208(6.88E-3) | 0.246(4.95E-3) | 0.224(2.97E-4) |
| | 3 | 0.237(1.74E-3)‡ | 0.213(1.77E-3)‡ | 0.256(1.03E-2) | 0.177(5.09E-5) | 0.176(1.67E-5) |
| | 5 | 0.669(2.78E-2)‡ | 0.507(4.02E-3)‡ | 0.504(8.40E-3) | 0.594(3.74E-2) | 0.356(1.34E-4) |
| DTLZ2 | 8 | 0.589(3.31E-2) | 0.714(2.53E-3) | 0.623(2.87E-2)† | 1.06(7.44E-3)† | 0.647(4.35E-3)† |
| | 10 | 0.675(1.07E-1) | 0.336(4.38E-3) | 0.684(2.87E-2) | 0.845(1.74E-2)† | 0.447(8.80E-3)† |
| | 3 | 0.552(8.33E-2)‡ | 0.265(6.54E-3)‡ | 1.8(2.42e+00) | 0.192(4.92E-3) | 0.912(3.99E-1) |
| | 5 | 0.702(5.80E-1)‡ | 0.498(7.69E-3)‡ | 1.13(5.84E-1) | 0.985(5.18E-1) | 0.355(6.75E-5) |
| DTLZ3 | 8 | 0.592(1.58E-1) | 0.701(3.39E-3) | 0.786(3.77E-2)† | 2.28(3.64e+00) | 0.625(4.35E-3)† |
| | 10 | 0.361(4.24E-1) | 0.385(1.01E-2) | 0.637(1.52E-2)† | 1.35(4.79E-1) | 0.431(7.60E-3)† |
| | 3 | 0.551(9.04E-2)‡ | 0.618(9.03E-2) | 0.621(9.88E-2) | 0.474(9.66E-2) | 0.553(8.92E-2) |
| | 5 | 0.877(3.90E-2)‡ | 0.612(2.75E-2) | 0.583(2.27E-2) | 1.16(1.15E-2) | 0.664(6.38E-2) |
| DTLZ4 | 8 | 0.875(2.18E-2)‡ | 0.851(1.17E-2) | 0.807(7.42E-2) | 1.5(2.09E-6) | 0.835(3.24E-2) |
| | 10 | 0.773(1.88E-1)‡ | 0.6(2.26E-2) | 0.668(1.69E-2) | 1.26(1.10E-8) | 0.556(1.65E-2) |
| | 3 | 0.336(6.22E-5) | 0.312(6.14E-7) | 0.327(1.15E-3) | 0.312(3.10E-6) | 0.321(5.68E-9)† |
| | 5 | 0.226(8.72E-1) | 0.223(1.09E-6) | 0.344(4.26E-2)† | 0.426(1.52E-2)† | 0.23(3.00E-6)† |
| DTLZ5 | 8 | 0.756(3.11E-1) | 0.721(1.12E-4) | 0.8(7.53E-3)† | 0.734(1.14E-3) | 0.736(1.42E-7)† |
| | 10 | 0.538(7.42E-1) | 0.581(1.17E-2) | 0.779(1.49E-2)† | 1.51(5.33E-2)† | 1.22(1.37E-9)† |
| | 3 | 0.444(3.49E-2)‡ | 0.425(1.06E-3) | 0.459(2.43E-3) | 1.54(1.40E-2) | 0.424(7.19E-4) |
| | 5 | 0.449(7.50E-1) | 0.33(9.20E-4) | 0.4(4.59E-2) | 3.41(2.17e+00)† | 0.33(1.40E-3) |
| DTLZ6 | 8 | 0.941(5.30E-1) | 0.735(3.99E-4) | 1.11(2.28E-1)† | 2.76(5.79E-1)† | 0.741(2.75E-4) |
| | 10 | 1.03(3.71E-1) | 0.645(1.62E-2) | 0.889(2.11E-2)† | 7.54(1.06e+00)† | 1.29(6.01E-4)† |

† denotes our proposed method significantly outperforms other peer algorithms according to the Wilcoxon's rank sum test at a 0.05 significance level;
‡ denotes the corresponding peer algorithm outperforms our proposed algorithm.

**Table A6:** The mean(std) of $\bar{\epsilon}(\mathcal{S})$ obtained by our proposed `D-PBEMO` algorithm instances against the peer algorithms.

| PROBLEM | $m$ | D-PBNSGA-II | D-PBMOEA/D | I-MOEA/D-PLVF | I-NSGA-II/LTR | IEMO/D |
|---|---|---|---|---|---|---|
| ZDT1 | 2 | 0.04(6.47E-7) | 0.126(1.41E-3) | 0.146(3.37E-3)† | 0.162(1.24E-3)† | 0.082(1.14E-2) |
| ZDT2 | 2 | 0.173(1.73E-3) | 0.683(9.54E-2) | 0.426(1.85E-2)† | 0.208(1.64E-3) | 0.366(9.23E-2) |
| ZDT3 | 2 | 0.088(5.80E-4) | 0.209(3.26E-3) | 0.593(4.02E-2)† | 0.188(2.15E-3)† | 0.129(8.78E-4)† |
| ZDT4 | 2 | 0.161(2.29E-2) | 0.166(6.78E-3) | 0.145(7.41E-3) | 0.337(2.80E-2) | 0.11(1.28E-2) |
| ZDT6 | 2 | 0.052(2.30E-4) | 0.134(4.59E-5) | 0.226(1.81E-2)† | 0.33(1.21E-3)† | 0.053(6.69E-6)† |
| WFG1 | 3 | 2.31(5.73E-4)‡ | 2.37(3.19E-1)‡ | 2.08(6.04E-3) | 2.17(3.73E-4) | 2.52(8.62E-2) |
| WFG3 | 3 | 1.7(1.17E-1)‡ | 1.15(1.66E-1)‡ | 1.12(3.28E-2) | 1.27(2.38E-3) | 0.745(6.44E-5) |
| WFG5 | 3 | 2.57(3.74E-1) | 1.64(1.22e+00) | 3.14(3.15E-1)† | 2.5(2.69E-2)† | 2.64(2.11E-4)† |
| WFG7 | 3 | 2.19(3.78E-1)‡ | 1.8(1.57e+00)‡ | 3.33(2.32E-1) | 2.9(9.81E-3) | 1.51(4.05E-3) |
| DTLZ1 | 3 | 0.223(1.42E-3) | 0.143(2.05E-2) | 0.239(2.02E-3) | 0.21(8.71E-4)† | 0.172(9.39E-5)† |
|  | 5 | 0.56(8.51E-2)‡ | 0.391(1.12E-4)‡ | 3.6(5.58E-1) | 0.265(3.57E-2) | 0.305(1.65E-5) |
|  | 8 | 3.38(8.02e+01) | 0.54(2.00E-4) | 57.6(4.90e+00)† | 0.399(1.69E-2)† | 0.548(1.40E-3) |
|  | 10 | 13.3(5.29e+02) | 0.209(4.21E-4) | 8.87(7.74E-1)† | 0.285(7.48E-3) | 0.255(9.30E-4)† |
| DTLZ2 | 3 | 0.254(3.64E-3)‡ | 0.416(2.08E-3)‡ | 0.55(9.63E-3) | 0.235(1.38E-3) | 0.176(1.66E-5) |
|  | 5 | 0.669(2.72E-2)‡ | 0.626(3.14E-3)‡ | 0.779(1.32E-2) | 0.701(3.59E-2) | 0.406(2.66E-2) |
|  | 8 | 1.25(2.68E-2) | 0.798(2.30E-3) | 1.03(2.51E-2)† | 1.12(3.55E-3)† | 0.941(7.17E-2)† |
|  | 10 | 1.65(3.64E-1) | 0.498(1.77E-2) | 0.968(1.13E-2)† | 0.905(1.41E-2)† | 0.831(4.02E-2)† |
| DTLZ3 | 3 | 0.574(8.00E-2) | 0.447(1.06E-2) | 158.0(1.42e+02)† | 0.502(1.23E-2)† | 1.04(4.56E-1)† |
|  | 5 | 6.03(5.18e+01)‡ | 0.633(6.21E-3)‡ | 19.4(7.02e+00) | 1.46(2.65e+00) | 0.429(2.51E-2) |
|  | 8 | 48.5(7.50e+03) | 0.795(7.39E-3) | 329.0(8.25e+01)† | 4.82(1.14e+02)† | 0.903(4.10E-2)† |
|  | 10 | 1.47(3.72E-1) | 0.565(4.12E-2) | 49.7(5.93e+00)† | 2.25(3.56e+00)† | 0.814(2.43E-2)† |
| DTLZ4 | 3 | 0.566(9.57E-2) | 0.716(5.49E-2) | 0.834(5.17E-2) | 0.615(4.95E-2) | 0.563(9.02E-2) |
|  | 5 | 0.989(4.83E-2)‡ | 0.684(2.42E-2) | 0.777(1.45E-2) | 1.17(8.18E-3) | 0.674(6.00E-2) |
|  | 8 | 1.21(2.31E-2) | 0.935(1.03E-2) | 1.19(4.18E-2)† | 1.5(0.00e+00)† | 1.0(4.23E-2) |
|  | 10 | 1.62(4.54E-1) | 0.683(2.30E-2) | 0.979(6.08E-3)† | 1.26(5.69E-8)† | 0.893(1.01E-2)† |
| DTLZ5 | 3 | 0.337(4.33E-5)‡ | 0.345(1.37E-4)‡ | 0.552(5.70E-3) | 0.344(2.43E-4) | 0.321(2.94E-9) |
|  | 5 | 1.54(8.14E-1)‡ | 0.261(6.29E-4)‡ | 0.909(1.48E-1) | 0.576(1.99E-2) | 0.222(1.37E-11) |
|  | 8 | 1.69(3.34E-1) | 0.758(1.83E-4) | 1.15(7.10E-2)† | 0.772(4.92E-3) | 0.776(4.04E-8)† |
|  | 10 | 2.3(3.47E-1) | 0.951(4.70E-3) | 0.992(1.11E-1) | 1.79(5.38E-2†) | 1.22(1.41E-9)† |
| DTLZ6 | 3 | 1.54(7.80E-2)‡ | 0.489(1.64E-3)‡ | 1.81(4.49E-3) | 1.83(2.18E-2) | 0.424(7.21E-4) |
|  | 5 | 8.99(8.03E-1)‡ | 0.421(2.94E-3)‡ | 1.57(4.90E-1) | 3.86(2.41e+00) | 0.332(1.46E-3) |
|  | 8 | 9.57(5.51E-1)‡ | 0.798(1.99E-3) | 3.79(1.98E-1) | 3.12(7.40E-1) | 0.798(4.12E-4) |
|  | 10 | 10.5(2.09E-1) | 1.08(7.47E-2) | 1.53(3.75E-1)† | 8.15(8.62E-1)† | 1.3(1.76E-3)† |

† denotes our proposed method significantly outperforms other peer algorithms according to the Wilcoxon's rank sum test at a 0.05 significance level;

‡ denotes the corresponding peer algorithm outperforms our proposed algorithm.

**Table A7:** The mean(std) of $\epsilon^{\star}(\mathcal{S})$ obtained by our proposed `D-PBEMO` algorithm in distillation experiments.

| PROBLEM | $m$ | D-PBNSGA-II | D-PBMOEA/D | D-PBEMO-DTS | D-PBEMO-PBO |
|---------|-----|-------------|------------|-------------|-------------|
| ZDT1 | 2 | 0.039(2.40E-7) | 0.067(1.85E-3) | 0.721(4.48E-3)† | 0.04(3.53E-6) |
| ZDT2 | 2 | 0.172(1.71E-3)‡ | 0.208(1.26E-3)‡ | 0.915(4.78E-2) | 0.153(7.07E-4) |
| ZDT3 | 2 | 0.086(5.87E-4) | 0.098(2.17E-3) | 1.16(2.38E-2)† | 0.75(4.25E-2)† |
| ZDT4 | 2 | 0.141(1.91E-2) | 0.114(7.88E-3) | 0.146(2.07E-2) | 0.107(6.81E-3) |
| ZDT6 | 2 | 0.046(1.76E-6) | 0.055(2.16E-5) | 0.13(1.41E-3) | 0.058(2.59E-5) |
| WFG1 | 3 | 2.24(6.44E-4) | 2.36(3.17E-1) | 2.26(8.56E-4) | 2.28(2.03E-2) |
| WFG3 | 3 | 1.7(1.17E-1)‡ | 1.01(4.48E-2) | 0.913(3.05E-3) | 0.926(3.82E-4) |
| WFG5 | 3 | 1.29(3.74E-1) | 1.62(1.23e+00) | 1.72(1.21E-3)† | 1.97(2.74E-3)† |
| WFG7 | 3 | 1.1(3.78E-1) | 1.79(1.58E-1) | 1.33(2.30E-3)† | 2.04(2.35E-2)† |
| DTLZ1 | 3 | 0.144(1.48E-3) | 0.194(3.11E-4) | 0.34(3.26E-3)† | 0.204(2.32E-3)† |
| | 5 | 0.417(2.20E-2) | 0.318(3.99E-4) | 0.572(1.05E-1)† | 0.327(1.93E-5) |
| | 8 | 0.526(7.62E-1)‡ | 0.512(2.81E-4)‡ | 4.23(5.36e+01) | 0.35(1.28E-3) |
| | 10 | 0.5(4.91E-3) | 0.25(6.59E-4) | 5.99(5.99e+01)† | 0.275(5.37E-2) |
| DTLZ2 | 3 | 0.237(1.74E-3)‡ | 0.213(1.77E-3)‡ | 0.56(1.89E-2) | 0.17(2.74E-4) |
| | 5 | 0.669(2.78E-2) | 0.507(4.02E-3) | 1.06(1.62E-2)† | 0.528(8.98E-5)† |
| | 8 | 0.589(3.31E-2) | 0.714(2.53E-3) | 1.42(9.76E-3)† | 0.793(1.73E-4)† |
| | 10 | 0.675(1.07E-1) | 0.336(4.38E-3) | 1.13(1.75E-3)† | 0.457(1.73E-6)† |
| DTLZ3 | 3 | 0.552(8.33E-2) | 0.265(6.54E-3) | 0.827(1.92E-2)† | 0.298(8.37E-5) |
| | 5 | 0.702(5.80E-1)‡ | 0.498(7.69E-3)‡ | 2.26(3.21e+00) | 0.351(1.29E-3) |
| | 8 | 0.592(1.58E-1) | 0.701(3.39E-3) | 133.0(8.89e+03)† | 0.709(3.62E-3)† |
| | 10 | 0.361(4.24E-1) | 0.385(1.01E-2) | 981.0(5.48e+05)† | 0.391(2.83E-3) |
| DTLZ4 | 3 | 0.551(9.04E-2) | 0.618(9.03E-2) | 0.72(3.47E-2)† | 0.563(1.20E-2) |
| | 5 | 0.877(3.90E-2) | 0.612(2.75E-2) | 0.866(2.92E-2)† | 0.698(8.33E-2) |
| | 8 | 0.875(2.18E-2) | 0.851(1.17E-2) | 1.47(4.52E-3)† | 0.988(3.71E-1)† |
| | 10 | 0.773(1.88E-1) | 0.6(2.26E-2) | 1.23(4.61E-3)† | 0.684(1.12E-4) |
| DTLZ5 | 3 | 0.336(6.22E-5) | 0.312(6.14E-7) | 0.392(5.13E-3)† | 0.328(1.20E-2) |
| | 5 | 0.226(8.72E-1) | 0.223(1.09E-6) | 1.65(2.44E-1)† | 0.36(3.38E-5)† |
| | 8 | 0.756(3.11E-1) | 0.721(1.12E-4) | 1.43(2.89E-1)† | 0.749(4.35E-3) |
| | 10 | 0.538(7.42E-1) | 0.581(1.17E-2) | 0.923(2.15E-1)† | 0.803(7.29E-2)† |
| DTLZ6 | 3 | 0.444(3.49E-2) | 0.425(1.06E-3) | 1.39(2.19E-2) | 0.421(4.91E-3) |
| | 5 | 0.449(7.50E-1) | 0.33(9.20E-4) | 2.69(6.31E-1)† | 0.383(1.48E-3) |
| | 8 | 0.941(5.30E-1) | 0.735(3.99E-4) | 4.64(1.58e+00)† | 0.798(2.75E-2)† |
| | 10 | 1.03(3.71E-1) | 0.645(1.62E-2) | 4.33(6.03E-1)† | 1.23(1.12E-4)† |

† denotes our proposed method significantly outperforms other peer algorithms according to the Wilcoxon's rank sum test at a 0.05 significance level;
‡ denotes the corresponding peer algorithm outperforms our proposed algorithm.

**Table A8:** The mean(std) of $\bar{\epsilon}(\mathcal{S})$ obtained by our proposed `D-PBEMO` algorithm in distillation experiments.

| PROBLEM | $m$ | D-PBNSGA-II | D-PBMOEA/D | D-PBEMO-DTS | D-PBEMO-PBO |
|---|---|---|---|---|---|
| ZDT1 | 2 | 0.04(6.47E-7) | 0.126(1.41E-3) | 0.73(3.89E-3)† | 0.061(8.99E-5)† |
| ZDT2 | 2 | 0.173(1.73E-3) | 0.683(9.54E-2) | 0.939(4.45E-2)† | 0.174(3.00E-3) |
| ZDT3 | 2 | 0.088(5.80E-4) | 0.209(3.26E-3) | 1.21(6.54E-3)† | 0.879(3.97E-2)† |
| ZDT4 | 2 | 0.161(2.29E-2) | 0.166(6.78E-3) | 0.231(2.66E-2) | 0.138(7.41E-3) |
| ZDT6 | 2 | 0.052(2.30E-4) | 0.134(4.59E-5) | 0.211(4.04E-4)† | 0.108(2.02E-5)† |
| WFG1 | 3 | 2.31(5.73E-4)‡ | 2.37(3.19E-1)‡ | 2.32(5.82E-4) | 2.21(7.32E-5) |
| WFG3 | 3 | 1.7(1.17E-1)‡ | 1.15(1.66E-1)‡ | 0.914(3.05E-3) | 1.27(8.38E-3) |
| WFG5 | 3 | 2.57(3.74E-1) | 1.64(1.22e+00) | 1.72(1.21E-3)† | 1.69(9.87E-5) |
| WFG7 | 3 | 2.19(3.78E-1)‡ | 1.8(1.57e+00)‡ | 1.33(2.30E-3) | 1.2(2.74E-3) |
| DTLZ1 | 3 | 0.223(1.42E-3) | 0.143(2.05E-2) | 0.393(5.06E-2)† | 2.1(3.80E-4)† |
| | 5 | 0.56(8.51E-2) | 0.391(1.12E-4) | 2.04(9.56e+00)† | 3.01(1.85E-4)† |
| | 8 | 3.38(8.02e+01) | 0.54(2.00E-4) | 11.8(1.32e+03)† | 0.571(4.80E-3) |
| | 10 | 13.3(5.29e+02) | 0.209(4.21E-4) | 8.97(1.29e+02)† | 0.269(8.92E-4)† |
| DTLZ2 | 3 | 0.254(3.64E-3) | 0.416(2.08E-3) | 0.585(2.20E-2)† | 0.499(3.41E-3)† |
| | 5 | 0.669(2.72E-2) | 0.626(3.14E-3) | 1.15(9.99E-3)† | 0.699(3.95E-3) |
| | 8 | 1.25(2.68E-2) | 0.798(2.30E-3) | 1.47(3.08E-3)† | 1.21(4.87E-5)† |
| | 10 | 1.65(3.64E-1) | 0.498(1.77E-2) | 1.16(1.88E-3)† | 1.0(1.87E-4)† |
| DTLZ3 | 3 | 0.574(8.00E-2) | 0.447(1.06E-2) | 0.86(1.66E-2)† | 0.573(1.74E-4)† |
| | 5 | 6.03(5.18e+01) | 0.633(6.21E-3) | 5.21(3.68e+01)† | 0.672(6.73E-3) |
| | 8 | 48.5(7.50e+03) | 0.795(7.39E-3) | 428.0(2.75e+05)† | 0.798(7.85E-3) |
| | 10 | 1.47(3.72E-1) | 0.565(4.12E-2) | 1170.0(4.15e+05)† | 0.609(1.78E-3)† |
| DTLZ4 | 3 | 0.566(9.57E-2) | 0.716(5.49E-2) | 0.746(3.35E-2)† | 0.578(1.03E-4) |
| | 5 | 0.989(4.83E-2)‡ | 0.684(2.42E-2) | 1.05(1.94E-2) | 0.649(2.17E-4) |
| | 8 | 1.21(2.31E-2) | 0.935(1.03E-2) | 1.49(8.32E-4)† | 1.02(6.60E-3)† |
| | 10 | 1.62(4.54E-1) | 0.683(2.30E-2) | 1.25(6.01E-4)† | 0.703(1.72E-4)† |
| DTLZ5 | 3 | 0.337(4.33E-5) | 0.345(1.37E-4) | 0.404(8.62E-3)† | 0.348(4.73E-4) |
| | 5 | 1.54(8.14E-1) | 0.261(6.29E-4) | 1.74(2.58E-1)† | 0.261(2.18E-4) |
| | 8 | 1.69(3.34E-1) | 0.758(1.83E-4) | 1.54(3.07E-1)† | 0.78(1.82E-3) |
| | 10 | 2.3(3.47E-1) | 0.951(4.70E-3) | 0.992(2.13E-1)† | 1.04(2.90E-3)† |
| DTLZ6 | 3 | 1.54(7.80E-2) | 0.489(1.64E-3) | 1.61(2.54E-2)† | 0.509(1.91E-3)† |
| | 5 | 8.99(8.03E-1)‡ | 0.421(2.94E-3) | 3.1(8.25E-1) | 0.402(2.78E-4) |
| | 8 | 9.57(5.51E-1) | 0.798(1.99E-3) | 5.03(1.44e+00)† | 0.801(9.81E-4)† |
| | 10 | 10.5(2.09E-1) | 1.08(7.47E-2) | 4.75(6.14E-1)† | 1.19(3.80E-4)† |

† denotes our proposed method significantly outperforms other peer algorithms according to the Wilcoxon's rank sum test at a 0.05 significance level;
‡ denotes the corresponding peer algorithm outperforms our proposed algorithm.

**Table A9:** The statistical comparison results of $\epsilon^\star(\mathcal{S})$ obtained by D-PBEMO with different KL threshold $\varepsilon$.

| PROBLEM | $m$ | $\varepsilon = 10^{-1}$ | $\varepsilon = 10^{-3}$ | $\varepsilon = 10^{-6}$ |
|---|---|---|---|---|
| ZDT1 | 2 | 0.04(6.35E-7) | 0.066(1.76E-3) | 0.04(1.94E-6) |
| ZDT2 | 2 | 0.188(2.02E-3) | 0.208(1.26E-3) | 0.187(1.84E-3) |
| ZDT3 | 2 | 0.072(1.29E-5) | 0.098(2.17E-3)† | 0.073(1.79E-5)† |
| ZDT4 | 2 | 0.048(1.67E-5) | 0.114(7.88E-3)† | 0.058(2.90E-4)† |
| ZDT6 | 2 | 0.055(2.48E-6) | 0.055(2.16E-5) | 0.054(4.54E-7) |
| WFG1 | 3 | 2.2(1.06E-2) | 2.36(3.17E-1) | 2.25(4.02E-2) |
| WFG3 | 3 | 1.03(4.74E-2) | 1.01(4.48E-2) | 1.03(3.72E-2) |
| WFG5 | 3 | 3.7(1.90e+00) | 3.24(1.23e+00) | 3.29(1.17e+00) |
| WFG7 | 3 | 3.59(1.53e+00) | 3.59(1.58e+00) | 2.96(7.92E-2) |
| DTLZ1 | 3 | 0.18(2.46E-4)† | 0.194(3.11E-4) | 0.182(7.28E-5) |
|  | 5 | 0.304(3.95E-4) | 0.318(3.99E-4) | 0.313(4.33E-4) |
|  | 8 | 0.511(2.67E-4) | 0.512(2.81E-4) | 0.516(2.09E-4) |
|  | 10 | 0.257(1.20E-3) | 0.25(6.59E-4) | 0.258(7.16E-4) |
| DTLZ2 | 3 | 0.211(2.04E-3) | 0.213(1.77E-3) | 0.21(8.70E-4) |
|  | 5 | 0.442(4.39E-3) | 0.507(4.02E-3)† | 0.5(6.90E-3)† |
|  | 8 | 0.757(6.38E-3)† | 0.714(2.53E-3) | 0.705(1.84E-3) |
|  | 10 | 0.569(1.29E-2)† | 0.336(4.38E-3) | 0.377(7.22E-3)† |
| DTLZ3 | 3 | 0.22(4.63E-3) | 0.265(6.54E-3) | 0.219(3.08E-3) |
|  | 5 | 0.457(3.82E-3) | 0.498(7.69E-3) | 0.467(2.76E-3) |
|  | 8 | 0.775(8.16E-3)† | 0.701(3.39E-3) | 0.725(3.35E-3) |
|  | 10 | 0.576(1.57E-2)† | 0.385(1.01E-2) | 0.371(5.07E-3) |
| DTLZ4 | 3 | 0.645(1.16E-1)† | 0.618(9.03E-2) | 0.474(1.18E-1) |
|  | 5 | 0.654(2.14E-2)† | 0.612(2.75E-2) | 0.666(2.29E-2)† |
|  | 8 | 0.926(1.60E-2)† | 0.851(1.17E-2) | 0.804(1.53E-2) |
|  | 10 | 0.663(1.57E-2)† | 0.6(2.26E-2) | 0.507(1.02E-2) |
| DTLZ5 | 3 | 0.312(8.36E-7) | 0.312(6.14E-7) | 0.312(4.70E-7) |
|  | 5 | 0.223(3.71E-6) | 0.223(1.09E-6) | 0.223(1.55E-6) |
|  | 8 | 0.718(2.03E-5) | 0.721(1.12E-4) | 0.724(1.46E-4) |
|  | 10 | 0.51(1.27E-3) | 0.581(1.17E-2)† | 0.617(1.19E-2)† |
| DTLZ6 | 3 | 0.42(1.38E-3) | 0.425(1.06E-3) | 0.418(8.85E-4) |
|  | 5 | 0.339(1.11E-3) | 0.33(9.20E-4) | 0.341(1.53E-3) |
|  | 8 | 0.743(3.77E-4) | 0.735(3.99E-4) | 0.742(9.50E-5) |
|  | 10 | 0.562(2.30E-3) | 0.645(1.62E-2) | 0.663(9.24E-3)† |

† denotes our proposed method with this $\varepsilon$ setting significantly outperforms other settings according to the Wilcoxon's rank sum test at a 0.05 significance level.

**Table A10:** Statistical comparison results of $\bar{\epsilon}(\mathcal{S})$ obtained by D-PBEMO with different $\varepsilon$ settings.

| PROBLEM | $m$ | $\varepsilon = 10^{-1}$ | $\varepsilon = 10^{-3}$ | $\varepsilon = 10^{-6}$ |
|---------|-----|-------------------------|-------------------------|-------------------------|
| ZDT1 | 2 | 0.113(6.44E-6)† | 0.128(1.40E-3) | 0.11(1.17E-5) |
| ZDT2 | 2 | 0.293(3.50E-3) | 0.683(9.54E-2) | 0.278(2.19E-3) |
| ZDT3 | 2 | 0.22(5.51E-4)† | 0.209(3.26E-3) | 0.201(3.05E-4) |
| ZDT4 | 2 | 0.121(1.40E-4) | 0.166(6.78E-3) | 0.127(3.47E-4) |
| ZDT6 | 2 | 0.139(1.16E-4)† | 0.134(4.59E-5) | 0.133(8.94E-7) |
| WFG1 | 3 | 2.22(6.62E-3) | 2.37(3.19E-1)† | 2.27(3.57E-2) |
| WFG3 | 3 | 1.15(1.63E-1) | 1.15(1.66E-1) | 1.12(1.20E-1) |
| WFG5 | 3 | 3.72(1.86e+00)† | 3.27(1.22e+00) | 3.51(1.61e+00)† |
| WFG7 | 3 | 3.6(1.51e+00) | 3.6(1.57e+00)† | 3.0(1.01E-1) |
| DTLZ1 | 3 | 0.243(4.50E-4)† | 0.243(2.05e+02) | 0.231(1.06E-4) |
| | 5 | 0.394(3.04E-4)† | 0.391(1.12E-4) | 0.384(7.81E-5) |
| | 8 | 0.561(3.85E-4) | 0.56(2.00E-4) | 0.555(2.68E-4) |
| | 10 | 0.318(2.49E-4)† | 0.314(4.21E-4) | 0.299(1.27E-4) |
| DTLZ2 | 3 | 0.416(3.42E-3) | 0.416(2.08E-3) | 0.396(5.66E-4) |
| | 5 | 0.655(3.28E-3) | 0.626(3.14E-3) | 0.627(3.67E-3) |
| | 8 | 0.929(7.71E-3)† | 0.798(2.30E-3) | 0.794(4.20E-3) |
| | 10 | 0.709(4.36E-3)† | 0.498(1.77E-2) | 0.58(3.64E-2) |
| DTLZ3 | 3 | 0.45(1.13E-2) | 0.447(1.06E-2) | 0.436(7.92E-3) |
| | 5 | 0.686(4.82E-3)† | 0.633(6.21E-3) | 0.599(1.89E-3) |
| | 8 | 0.93(5.40E-3)† | 0.795(7.39E-3) | 0.832(6.22E-3) |
| | 10 | 0.744(9.40E-3)† | 0.565(4.12E-2) | 0.504(1.32E-2) |
| DTLZ4 | 3 | 0.753(5.75E-2)† | 0.716(5.49E-2) | 0.604(6.36E-2) |
| | 5 | 0.836(1.39E-2)† | 0.684(2.42E-2) | 0.758(1.52E-2) |
| | 8 | 1.09(8.10E-3)† | 0.935(1.03E-2) | 0.916(2.05E-2) |
| | 10 | 0.829(1.24E-2)† | 0.683(2.30E-2) | 0.634(1.53E-2) |
| DTLZ5 | 3 | 0.359(2.06E-4)† | 0.345(1.37E-4) | 0.347(1.14E-4) |
| | 5 | 0.301(9.21E-3)† | 0.261(6.29E-4) | 0.265(8.00E-4) |
| | 8 | 0.83(2.45E-2)† | 0.758(1.83E-4) | 0.779(3.25E-3)† |
| | 10 | 0.964(4.45E-3) | 0.951(4.70E-3) | 0.991(1.12E-2) |
| DTLZ6 | 3 | 0.479(2.92E-3) | 0.489(1.64E-3) | 0.479(1.73E-3) |
| | 5 | 0.458(3.48E-3)† | 0.421(2.94E-3) | 0.439(2.02E-3) |
| | 8 | 0.834(3.81E-3)† | 0.798(1.99E-3) | 0.821(1.75E-2)† |
| | 10 | 1.04(1.21E-2) | 1.08(7.47E-2)† | 1.07(2.69E-2) |

† denotes our proposed method with this $\varepsilon$ setting significantly outperforms other settings according to the Wilcoxon's rank sum test at a 0.05 significance level.

**Table A11:** The statistical comparison results of $\epsilon^{\star}(\mathcal{S})$ obtained by D-PBMOEA/D with different number of subsets $K$ ($m = 2$).

| PROBLEM | $m$ | $K = 2$ | $K = 5$ | $K = 10$ |
|---------|-----|---------|---------|----------|
| ZDT1 | 2 | 0.041(9.27E-6) | 0.04(3.53E-6) | 0.041(1.10E-5) |
| ZDT2 | 2 | 0.189(1.93E-3) | 0.195(2.67E-3) | 0.178(1.55E-3) |
| ZDT3 | 2 | 0.075(6.53E-5) | 0.079(1.12E-4)† | 0.072(3.79E-5) |
| ZDT4 | 2 | 0.062(7.32E-4) | 0.054(8.05E-4) | 0.075(3.28E-3) |
| ZDT6 | 2 | 0.054(5.82E-7) | 0.054(5.20E-7) | 0.054(5.18E-7) |

† denotes our proposed method with this $K$ setting significantly outperforms other settings according to the Wilcoxon's rank sum test at a 0.05 significance level.

**Table A12:** The statistical comparison results of $\bar{\epsilon}(\mathcal{S})$ obtained by D-PBMOEA/D with different number of subsets $K$ ($m = 2$).

| PROBLEM | $m$ | $K = 2$ | $K = 5$ | $K = 10$ |
|---------|-----|---------|---------|----------|
| ZDT1 | 2 | 0.111(2.08E-5) | 0.112(2.38E-5) | 0.112(1.52E-5) |
| ZDT2 | 2 | 0.283(1.73E-3) | 0.283(2.13E-3) | 0.269(2.11E-3) |
| ZDT3 | 2 | 0.2(2.37E-4) | 0.208(1.00E-4) | 0.206(3.72E-4) |
| ZDT4 | 2 | 0.121(5.99E-4) | 0.115(6.68E-4) | 0.144(2.85E-3)† |
| ZDT6 | 2 | 0.133(3.58E-6) | 0.133(3.33E-6) | 0.135(9.38E-6)† |

† denotes our proposed method with this $K$ setting significantly outperforms other settings according to the Wilcoxon's rank sum test at a 0.05 significance level.

**Table A13:** The statistical comparison results of $\epsilon^\star(\mathcal{S})$ obtained by `D-PBMOEA/D` with different subsets $K$ ($m = 3$).

| PROBLEM | $m$ | $K = 2$ | $K = 4$ | $K = 8$ | $K = N$ |
|---|---|---|---|---|---|
| WFG1 | 3 | 1.808(1.94E-3) | 1.791(8.30E-4) | 1.817(1.52E-3)† | 2.26(8.56E-4)† |
| WFG3 | 3 | 0.907(2.87E-2) | 0.942(1.83E-2) | 0.813(1.83E-2) | 0.913(3.05E-3)† |
| WFG5 | 3 | 1.694(2.61E-4) | 1.708(3.87E-3) | 1.716(5.24E-3) | 1.72(1.21E-3) |
| WFG7 | 3 | 1.3(7.86E-4) | 1.324(1.63E-2) | 1.369(1.00E-2) | 1.33(2.30E-3) |
| DTLZ1 | 3 | 0.212(2.31E-4)† | 0.187(2.68E-4) | 0.184(1.57E-4) | 0.34(3.26E-3)† |
| DTLZ2 | 3 | 0.363(6.72E-3)† | 0.224(2.41E-3)† | 0.213(5.77E-3) | 0.56(1.89E-2)† |
| DTLZ3 | 3 | 0.402(4.42E-3)† | 0.247(5.88E-3) | 0.262(2.68E-2) | 0.827(1.92E-2)† |
| DTLZ4 | 3 | 0.741(7.07E-2) | 0.717(6.11E-2) | 0.699(1.07E-1) | 0.72(3.47E-2) |
| DTLZ5 | 3 | 0.312(1.06E-6) | 0.313(1.39E-5) | 0.313(1.06E-5) | 0.392(5.13E-3)† |
| DTLZ6 | 3 | 0.417(6.82E-4) | 0.418(1.13E-3) | 0.413(1.25E-3) | 1.39(2.19E-2)† |

† denotes our proposed method with this $K$ setting significantly outperforms other settings according to the Wilcoxon's rank sum test at a 0.05 significance level.

**Table A14:** The statistical comparison results of $\bar{\epsilon}(\mathcal{S})$ obtainted by `D-PBMOEA/D` with different $K$ ($m = 3$).

| PROBLEM | $m$ | $K = 2$ | $K = 4$ | $K = 8$ | $K = N$ |
|---|---|---|---|---|---|
| WFG1 | 3 | 2.048(1.39E-3) | 2.001(3.36E-3) | 2.026(4.10E-3) | 2.32(5.82E-4)† |
| WFG3 | 3 | 1.374(1.95E-2)† | 1.299(2.68E-2)† | 1.043(2.60E-2) | 1.914(3.05E-3)† |
| WFG5 | 3 | 2.628(8.61E-3)† | 2.278(9.09E-3) | 2.542(7.15E-1) | 2.72(1.21E-3)† |
| WFG7 | 3 | 2.498(7.02E-3) | 2.499(4.10E-1) | 2.264(7.41E-1) | 2.63(2.30E-3)† |
| DTLZ1 | 3 | 0.286(3.20E-4)† | 0.234(2.18E-4)† | 0.214(3.22E-4) | 0.393(5.06E-2)† |
| DTLZ2 | 3 | 0.678(5.52E-3)† | 0.417(3.00E-3)† | 0.321(4.72E-3) | 0.585(2.20E-2)† |
| DTLZ3 | 3 | 0.67(2.42E-3)† | 0.454(8.31E-3)† | 0.382(2.71E-2) | 0.86(1.66E-2)† |
| DTLZ4 | 3 | 0.857(2.50E-2) | 0.775(3.55E-2) | 0.742(7.41E-2) | 0.746(3.35E-2) |
| DTLZ5 | 3 | 0.373(2.14E-4)† | 0.342(2.98E-4) | 0.338(7.64E-5) | 0.404(8.62E-3)† |
| DTLZ6 | 3 | 0.503(1.14E-3)† | 0.471(2.55E-3) | 0.434(1.80E-3) | 1.61(2.54E-2)† |

† denotes our proposed method with this $K$ setting significantly outperforms other settings according to the Wilcoxon's rank sum test at a 0.05 significance level.

**Table A15:** The statistical comparison results of $\epsilon^\star(\mathcal{S})$ obtainted by `D-PBMOEA/D` results with different $K$ ($m = 5$).

| PROBLEM | $m$ | $K = 2$ | $K = 8$ | $K = 16$ | $K = N$ |
|---|---|---|---|---|---|
| DTLZ1 | 5 | 0.303(3.83E-4) | 0.317(4.75E-4)† | 0.329(6.42E-4)† | 0.572(1.05E-1)† |
| DTLZ2 | 5 | 0.606(8.00E-3)† | 0.492(4.14E-3)† | 0.436(4.15E-3) | 1.06(1.62E-2)† |
| DTLZ3 | 5 | 0.624(1.06E-2)† | 0.498(6.88E-3)† | 0.439(1.78E-3) | 2.26(3.21E+00)† |
| DTLZ4 | 5 | 0.724(1.80E-2) | 0.684(3.20E-2) | 0.657(3.27E-2) | 0.866(2.92E-2)† |
| DTLZ5 | 5 | 0.223(1.54E-6) | 0.223(1.62E-6) | 0.222(7.60E-7) | 1.65(2.44E-1)† |
| DTLZ6 | 5 | 0.33(1.34E-3) | 0.343(1.65E-3) | 0.333(1.19E-3) | 2.69(6.31E-1)† |

† denotes our proposed method with this $K$ setting significantly outperforms other settings according to the Wilcoxon's rank sum test at a 0.05 significance level.

**Table A16:** The statistical comparison results of $\bar{\epsilon}(\mathcal{S})$ obtainted by `D-PBMOEA/D` with different subsets $K$ ($m = 5$).

| PROBLEM | $m$ | $K = 2$ | $K = 8$ | $K = 16$ | $K = N$ |
|---|---|---|---|---|---|
| DTLZ1 | 5 | 0.418(1.77E-4)† | 0.393(2.48E-4)† | 0.373(1.28E-4) | 2.04(9.56E+00)† |
| DTLZ2 | 5 | 0.879(4.95E-3)† | 0.627(2.72E-3)† | 0.557(3.96E-3) | 1.15(9.99E-3)† |
| DTLZ3 | 5 | 0.878(4.22E-3)† | 0.624(5.26E-3)† | 0.578(3.56E-3) | 5.21(3.68E+01)† |
| DTLZ4 | 5 | 0.942(7.57E-3)† | 0.765(2.15E-2) | 0.72(2.42E-2) | 1.05(1.94E-2)† |
| DTLZ5 | 5 | 0.4(1.88E-3)† | 0.28(3.81E-4) | 0.27(2.75E-4) | 1.74(2.58E-1)† |
| DTLZ6 | 5 | 0.607(5.81E-3)† | 0.428(3.25E-3)† | 0.39(2.47E-3) | 3.1(8.25E-1)† |

† denotes our proposed method with this $K$ setting significantly outperforms other settings according to the Wilcoxon's rank sum test at a 0.05 significance level.

**Table A17:** The statistical comparison results of $\epsilon^\star(\mathcal{S})$ obtained by `D-PBMOEA/D` with different $K$ ($m = 8$).

| PROBLEM | $m$ | $K = 2$ | $K = 8$ | $K = 16$ | $K = N$ |
|---------|-----|---------|---------|----------|---------|
| DTLZ1 | 8 | 0.486(9.41E-5) | 0.509(2.88E-4)† | 0.516(2.53E-4)† | 4.23(5.36E+01)† |
| DTLZ2 | 8 | 0.857(4.91E-3)† | 0.698(9.91E-4) | 0.755(1.45E-2) | 1.42(9.76E-3)† |
| DTLZ3 | 8 | 0.811(5.61E-3)† | 0.708(6.81E-3) | 0.689(3.39E-3) | 133.0(8.89E+03)† |
| DTLZ4 | 8 | 0.926(9.45E-3)† | 0.841(1.65E-2) | 0.806(1.16E-2) | 1.47(4.52E-3) |
| DTLZ5 | 8 | 0.714(4.61E-9) | 0.717(3.27E-5)† | 0.721(7.73E-5)† | 1.43(2.89E-1)† |
| DTLZ6 | 8 | 0.714(1.81E-4) | 0.732(3.54E-4)† | 0.742(7.25E-4)† | 4.64(1.58E+00)† |

† denotes our proposed method with this $K$ setting significantly outperforms other settings according to the Wilcoxon's rank sum test at a 0.05 significance level.

**Table A18:** The statistical comparison results of $\bar{\epsilon}(\mathcal{S})$ obtained by `D-PBMOEA/D` with different $K$ ($m = 8$).

| PROBLEM | $m$ | $K = 2$ | $K = 8$ | $K = 16$ | $K = N$ |
|---------|-----|---------|---------|----------|---------|
| DTLZ1 | 8 | 0.58(1.52E-4)† | 0.563(1.16E-4)† | 0.554(9.18E-5) | 11.8(1.32E+03)† |
| DTLZ2 | 8 | 1.101(3.59E-3)† | 0.814(1.24E-3) | 0.83(1.33E-2) | 1.47(3.08E-3)† |
| DTLZ3 | 8 | 1.066(3.15E-3)† | 0.831(4.91E-3)† | 0.767(2.58E-3) | 428.0(2.75E+05)† |
| DTLZ4 | 8 | 1.184(7.36E-3)† | 0.947(1.12E-2) | 0.888(1.54E-2) | 1.49(8.32E-4)† |
| DTLZ5 | 8 | 0.83(8.00E-4)† | 0.757(4.27E-4) | 0.758(1.43E-4) | 1.54(3.07E-1)† |
| DTLZ6 | 8 | 0.9(2.89E-3)† | 0.802(9.58E-4) | 0.794(5.72E-4) | 5.03(1.44E+00)† |

† denotes our proposed method with this $K$ setting significantly outperforms other settings according to the Wilcoxon's rank sum test at a 0.05 significance level.

**Table A19:** The statistical comparison results of $\epsilon^\star(\mathcal{S})$ obtained by `D-PBMOEA/D` with different $K$ ($m = 10$).

| PROBLEM | $m$ | $K = 2$ | $K = 6$ | $K = 24$ | $K = N$ |
|---------|-----|---------|---------|----------|---------|
| DTLZ1 | 10 | 0.223(1.87E-4) | 0.248(4.71E-4)† | 0.244(6.34E-4)† | 5.99(5.99e+01)† |
| DTLZ2 | 10 | 0.537(2.58E-3)† | 0.437(8.59E-3) | 0.412(1.41E-2) | 1.13(1.75E-3)† |
| DTLZ3 | 10 | 0.496(3.96E-3)† | 0.488(1.17E-2)† | 0.382(6.49E-3) | 981.0(5.48E+05)† |
| DTLZ4 | 10 | 0.639(1.72E-2) | 0.611(3.25E-2) | 0.561(2.38E-2) | 1.23(4.61E-3)† |
| DTLZ5 | 10 | 0.495(3.18E-6) | 0.511(3.24E-4)† | 0.597(1.40E-2)† | 0.923(2.15E-1)† |
| DTLZ6 | 10 | 0.527(4.77E-4) | 0.562(2.03E-3)† | 0.659(1.43E-2)† | 4.33(6.03E-1)† |

† denotes our proposed method with this $K$ setting significantly outperforms other settings according to the Wilcoxon's rank sum test at a 0.05 significance level.

**Table A20:** The statistical comparison results of $\bar{\epsilon}(\mathcal{S})$ obtained by `D-PBMOEA/D` with different $K$ ($m = 10$).

| PROBLEM | $m$ | $K = 2$ | $K = 6$ | $K = 24$ | $K = N$ |
|---------|-----|---------|---------|----------|---------|
| DTLZ1 | 10 | 0.342(1.41E-4)† | 0.321(2.60E-4) | 0.323(5.62E-4) | 8.97(1.29e+02)† |
| DTLZ2 | 10 | 0.804(4.06E-3)† | 0.567(4.54E-3) | 0.562(3.63E-2) | 1.16(1.88E-3)† |
| DTLZ3 | 10 | 0.791(3.31E-3)† | 0.614(1.03E-2) | 0.544(2.29E-2) | 1170.0(4.15e+05)† |
| DTLZ4 | 10 | 0.934(2.92E-3)† | 0.722(1.80E-2) | 0.647(3.01E-2) | 1.25(6.01E-4)† |
| DTLZ5 | 10 | 0.818(1.21E-2) | 0.901(3.29E-3)† | 0.896(8.05E-3)† | 0.992(2.13E-1)† |
| DTLZ6 | 10 | 0.978(7.61E-3) | 0.981(5.44E-3) | 1.008(1.94E-2) | 4.75(6.14E-1)† |

† denotes our proposed method with this $K$ setting significantly outperforms other settings according to the Wilcoxon's rank sum test at a 0.05 significance level.

**Table A21:** RNA experiment settings.

| RNA | Eterna ID | Target Structure | Reference Point 1 | Reference Point 2 | Sample Solution | $n$ |
|---|---|---|---|---|---|---|
| 1 | 1074756 | ((((...))))。 | $(-6.3, 0)^\top$ | $(-7.1, 0.2)^\top$ | GAUAAAAUAUCA | 12 |
| 2 | 20111 | (((((......))))) | $(-9.1, 0)^\top$ | $(-13.8, 0.125)^\top$ | GGGGGGAAAAACCCCC | 16 |
| 3 | 997382 | ((....)).((....)) | $(-4, 0)^\top$ | $(-8.9, 0.6)^\top$ | CUGAAAAGAGUGAGAGC | 17 |
| 4 | 852950 | ..(((((((((.....)).))))))).. | $(-12, 0)^\top$ | $(-24, 0.23)^\top$ | AAAUGUGAAUGAA AAAUAUUAUAUAA | 26 |
| 5 | 727172 | (((((.....))..((.........))))) | $(-9, 0)^\top$ | $(-17.7, 0.33)^\top$ | GCCGCGAAAAGCAAC CGAAAAAAAAGGGGC | 30 |
| 7 | 477402 | ....((((((((.(.....).).).)))))).... | $(-24, 0)^\top$ | $(-31, 0.24)^\top$ | AAAAUACAGGGCGCGAA AGGACUCACUGUAAAAA | 34 |
| 8 | 997391 | ((....)).((.....)).((.....)).((.....)) | $(-13, 0)^\top$ | $(-28, 0.57)^\top$ | GAGAAAUCAGAGAAAUC AGUGAAAGCAGGGAAACU | 35 |
| 9 | 15819 | ((((((.(((((....))))))).)))......... | $(-15, 0)^\top$ | $(-38, 0.58)^\top$ | CGUGACAUUAUAAAAAUG AGUCGAUGAAAAAAAAAA | 36 |
| 10 | 816170 | ((((((.(((((((((....))))).)).).).)))))) | $(-26.7, 0)^\top$ | $(-45, 0.17)^\top$ | UAAAACGAGAAAAACGAA AGUUUUAUCAUGGUUUUA | 36 |

The column reference point 1 lists our first session experiment ($f_2 = 0$).
The column reference point 2 lists reference points in the second-session ($f_2 \in (0, 1)$).
The sample solution is the possible sequence for the given target structure provided by benchmark.

**Table A22:** The mean(std) of $\epsilon^\star(\mathcal{S})$ comparing our proposed method with peer algorithms on inverse RNA design problems given reference point 1

| RNA | D-PBNSGA-II | D-PBMOEA/D | I-MOEA/D-PLVF | I-NSGA2/LTR | IEMO/D |
|---|---|---|---|---|---|
| 1 | **0.289(0.0)** | 0.679(0.27) | 0.971(0.73)† | 0.35(0.05) | 0.605(0.21) |
| 2 | **0.33(0.01)** | 1.02(1.11) | 1.31(1.09)† | 1.342(1.51)† | 1.943(2.13)† |
| 3 | 2.454(11.6) | 7.89(3.39) | 7.102(13.05) | **1.332(0.19)** | 1.819(2.62) |
| 4 | 4.514(17.76) | 2.052(2.08) | **1.612(1.62)** | 1.848(0.86) | 2.96(2.05) |
| 5 | **2.871(4.43)** | 5.497(21.22) | 7.663(14.46)† | 7.557(31.47) | 6.951(11.98)† |
| 6 | **1.764(4.51)** | 4.878(11.11) | 5.699(21.21)† | 7.656(12.38)† | 4.1(10.83)† |
| 7 | 4.522(7.7) | 6.956(11.14) | 6.51(16.64) | **4.216(12.95)** | 5.284(13.19) |
| 8 | 6.604(26.97) | 12.114(23.74) | 13.638(28.74) | 5.945(15.62) | **3.959(15.97)** |
| 9 | **1.445(1.73)** | 3.656(5.75) | 4.506(10.72)† | 4.424(14.0)† | 6.731(40.42)† |
| 10 | 3.988(11.25) | 8.506(13.14) | **3.86(10.51)** | 4.988(20.48) | 7.128(23.01) |

† denotes our proposed method significantly outperforms other peer algorithms according to the Wilcoxon's rank sum test at a 0.05 significance level;
‡ denotes the corresponding peer algorithm outperforms our proposed algorithm.

**Table A23:** The mean(std) of $\bar{\epsilon}(\mathcal{S})$ of our proposed method with peer algorithms on inverse RNA design problems given reference point 1

| RNA | D-PBNSGA-II | D-PBMOEA/D | I-MOEA/D-PLVF | I-NSGA2/LTR | IEMO/D |
|---|---|---|---|---|---|
| 1 | **0.35(0.05)** | 0.685(0.26) | 1.063(0.65)† | 0.357(0.03)† | 0.605(0.21)† |
| 2 | **0.33(0.01)** | 1.259(1.14) | 1.832(1.04)† | 1.501(1.6)† | 1.945(2.13)† |
| 3 | 2.454(11.6) | 8.667(2.27) | 9.409(2.68) | **1.697(0.39)** | 3.234(8.49) |
| 4 | 4.514(17.76) | 2.569(3.85) | 1.758(1.35) | **1.849(0.86)** | 3.226(2.22) |
| 5 | **2.871(4.43)** | 6.894(15.14) | 9.261(5.33)† | 8.439(22.87)† | 7.206(12.81)† |
| 6 | **1.764(4.51)** | 5.372(10.74) | 6.451(19.45)† | 9.077(8.67)† | 4.833(8.18)† |
| 7 | 4.522(7.7) | 7.226(17.14) | 9.602(10.75) | **4.216(12.95)** | 6.579(10.64) |
| 8 | 6.604(26.97) | 12.775(24.08) | 14.764(25.17) | 7.105(13.47) | 7.131(14.81) |
| 9 | **1.46(1.74)** | 4.94(5.79) | 4.965(10.91)† | 5.332(11.64)† | 7.876(34.82)† |
| 10 | **4.001(11.16)** | 9.077(17.27) | 4.057(9.73) | 5.014(20.4) | 7.133(22.96) |

† denotes our proposed method significantly outperforms other peer algorithms according to the Wilcoxon's rank sum test at a 0.05 significance level;
‡ denotes the corresponding peer algorithm outperforms our proposed algorithm.

**Table A24:** The mean(std) of $\epsilon^\star(\mathcal{S})$ comparing our proposed method with peer algorithms on inverse RNA design problems given reference point 2

| RNA | D-PBNSGA-II | D-PBMOEA/D | I-MOEA/D-PLVF | I-NSGA2/LTR | IEMO/D |
|---|---|---|---|---|---|
| 1 | 0.459(0.07) | 0.469(0.09) | **0.333(0.09)** | 0.821(0.06) | 1.614(1.33) |
| 2 | 0.752(0.54) | **0.74(0.24)** | 2.112(1.11)† | 3.143(0.82)† | 4.131(1.38)† |
| 3 | 0.409(0.11) | **0.064(0.0)** | 2.095(2.54)† | 5.037(2.93)† | 6.343(5.02)† |
| 4 | **1.459(3.0)** | 1.72(2.1) | 4.25(5.64)† | 3.702(14.69) | 9.43(14.85)† |
| 5 | 2.247(2.19) | **0.84(0.65)** | 7.28(13.67)† | 5.405(23.99)† | 5.934(12.8)† |
| 6 | **1.846(2.53)** | 3.91(2.5) | 3.592(10.88)† | 7.571(19.21)† | 7.94(14.85)† |
| 7 | 2.612(4.65) | **1.518(1.65)** | 3.601(6.84) | 8.531(31.14)† | 10.78(33.49)† |
| 8 | 2.1(2.57) | **1.388(0.99)** | 4.953(31.37) | 7.202(50.66)† | 9.016(23.62)† |
| 9 | 5.559(25.5) | **4.071(4.5)** | 4.276(17.19) | 8.706(28.49)† | 13.123(31.61)† |
| 10 | **4.25(11.95)** | 7.42(2.38) | 8.94(13.94)† | 7.089(16.7) | 15.29(49.85)† |

† denotes our proposed method significantly outperforms other peer algorithms according to the Wilcoxon's rank sum test at a 0.05 significance level;
‡ denotes the corresponding peer algorithm outperforms our proposed algorithm.

**Table A25:** The mean(std) of $\bar{\epsilon}(\mathcal{S})$ of our proposed method with peer algorithms on inverse RNA design problems given reference point 2

| RNA | D-PBNSGA-II | D-PBMOEA/D | I-MOEA/D-PLVF | I-NSGA2/LTR | IEMO/D |
|---|---|---|---|---|---|
| 1 | 0.459(0.07) | 0.523(0.07) | **0.425(0.07)** | 0.821(0.06) | 1.614(1.33) |
| 2 | 0.753(0.54) | **0.745(0.22)** | 2.295(1.22)† | 3.143(0.82)† | 4.165(1.37)† |
| 3 | **0.409(0.11)** | 0.688(0.03) | 3.402(1.29)† | 5.072(2.92)† | 6.874(2.45)† |
| 4 | **1.462(3.0)** | 2.51(4.3) | 4.393(5.65)† | 3.797(14.2)† | 9.436(14.79)† |
| 5 | **2.247(2.19)** | 2.918(1.27) | 8.126(12.68)† | 5.535(23.02)† | 7.108(15.5)† |
| 6 | **1.846(2.53)** | 4.442(3.9) | 3.851(11.13)† | 8.347(24.1)† | 8.001(14.62)† |
| 7 | **2.616(4.64)** | 4.066(5.7) | 3.944(6.39) | 8.924(26.74)† | 11.35(31.91)† |
| 8 | **2.141(2.47)** | 3.988(2.57) | 9.281(12.21)† | 7.267(49.92) | 10.736(28.62)† |
| 9 | 5.724(24.65) | 5.729(7.45) | **4.701(15.73)** | 8.736(28.2) | 14.447(50.17) |
| 10 | **4.272(12.0)** | 7.477(2.3) | 8.954(13.95)† | 7.345(13.88) | 15.295(49.86)† |

† denotes our proposed method significantly outperforms other peer algorithms according to the Wilcoxon's rank sum test at a 0.05 significance level;
‡ denotes the corresponding peer algorithm outperforms our proposed algorithm.

**Table A26:** The difference between native and predicted protein in energy

| ID | Type | Bound | dDFIRE | Rosetta | RWplus |
|---|---|---|---|---|---|
| 1K36 | Native | 431.51 | -52.84 | 293.70 | -5059.39 |
| | Predicted | 431.75 | -41.66 | 402.33 | -3990.52 |
| 1ZDD | Native | 297.18 | -74.02 | -27.73 | -4604.18 |
| | Predicted | 328.84 | -63.03 | 63.03 | -3986.78 |
| 2M7T | Native | 269.76 | -39.51 | -10.82 | -3313.84 |
| | Predicted | 276.12 | -22.98 | 210.47 | -2111.19 |
| 3P7K | Native | 379.04 | -104.15 | -11.29 | -6140.81 |
| | Predicted | 413.47 | -91.21 | 184.17 | -3399.93 |

**Table A27:** The mean(std) of RMSD comparing our propsoed emthod with peer algorithms on PSP problems.

| ID | D-PBNSGA-II | D-PBMOEA/D | I-MOEA/D-PLVF | I-NSGA2-LTR | IEMO/D |
|---|---|---|---|---|---|
| 1K36 | 583.29(117.08) | **302.82(138.59)** | 682.23(182.63) | 597.19(284.91) | 610.62(402.31) |
| 1ZDD | 446.88(542.33) | **360.25(17.73)** | 623.14(394.14) | 450.23(582.19) | 488.28(518.42) |
| 2M7T | **350.51(8.95)** | 477.56(76.93) | 671.45(372.01) | 721.73(502.31) | 823.46(1023.54) |
| 3P7K | 719.90(1202.92) | **152.53(7.45)** | 663.29(802.99) | 692.31(823.13) | 818.93(923.87) |
| 3V1A | 687.07(497.33) | **584.43(28.34)** | 887.68(391.74) | 791.13(304.72) | 823.28(528.87) |

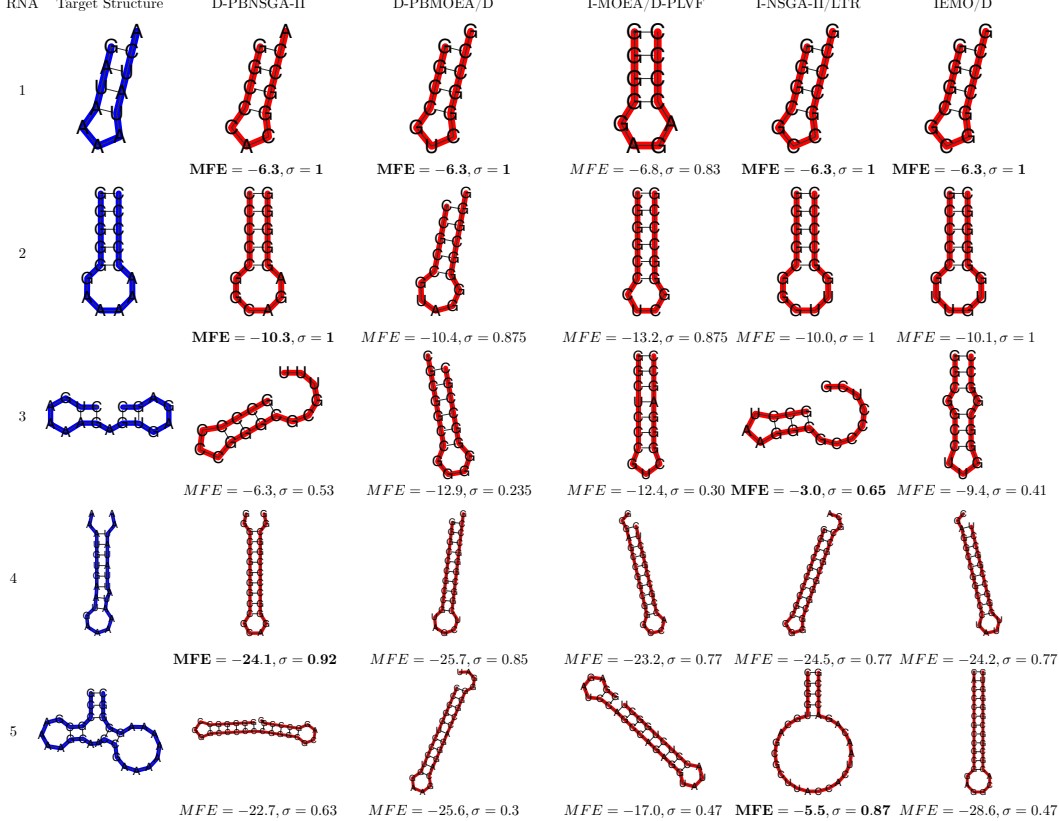

| RNA | Target Structure | D-PBNSGA-II | D-PBMOEA/D | I-MOEA/D-PLVF | I-NSGA-II/LTR | IEMO/D |
|---|---|---|---|---|---|---|
| 1 | | $\mathbf{MFE} = \mathbf{-6.3}, \sigma = \mathbf{1}$ | $\mathbf{MFE} = \mathbf{-6.3}, \sigma = \mathbf{1}$ | $MFE = -6.8, \sigma = 0.83$ | $\mathbf{MFE} = \mathbf{-6.3}, \sigma = \mathbf{1}$ | $\mathbf{MFE} = \mathbf{-6.3}, \sigma = \mathbf{1}$ |
| 2 | | $\mathbf{MFE} = \mathbf{-10.3}, \sigma = \mathbf{1}$ | $MFE = -10.4, \sigma = 0.875$ | $MFE = -13.2, \sigma = 0.875$ | $MFE = -10.0, \sigma = 1$ | $MFE = -10.1, \sigma = 1$ |
| 3 | | $MFE = -6.3, \sigma = 0.53$ | $MFE = -12.9, \sigma = 0.235$ | $MFE = -12.4, \sigma = 0.30$ | $\mathbf{MFE} = \mathbf{-3.0}, \sigma = \mathbf{0.65}$ | $MFE = -9.4, \sigma = 0.41$ |
| 4 | | $\mathbf{MFE} = \mathbf{-24.1}, \sigma = \mathbf{0.92}$ | $MFE = -25.7, \sigma = 0.85$ | $MFE = -23.2, \sigma = 0.77$ | $MFE = -24.5, \sigma = 0.77$ | $MFE = -24.2, \sigma = 0.77$ |
| 5 | | $MFE = -22.7, \sigma = 0.63$ | $MFE = -25.6, \sigma = 0.3$ | $MFE = -17.0, \sigma = 0.47$ | $\mathbf{MFE} = \mathbf{-5.5}, \sigma = \mathbf{0.87}$ | $MFE = -28.6, \sigma = 0.47$ |

**Figure A14:** The comparison results of `D-PBEMO` against the other three stat-of-the-art PBEMO algorithms on inverse RNA design problems. In particular, the target structure is a sample of possible solution represented in blue color while the predicted one obtained by different optimization algorithms are highlighted in red color. In this graph, the reference point is set as $\sigma = 1$. The closer $\sigma$ is to 1, the better performance achieved by the corresponding algorithm. When the $\sigma$ share the same biggest value, the smaller $MFE$ the better the performance is.

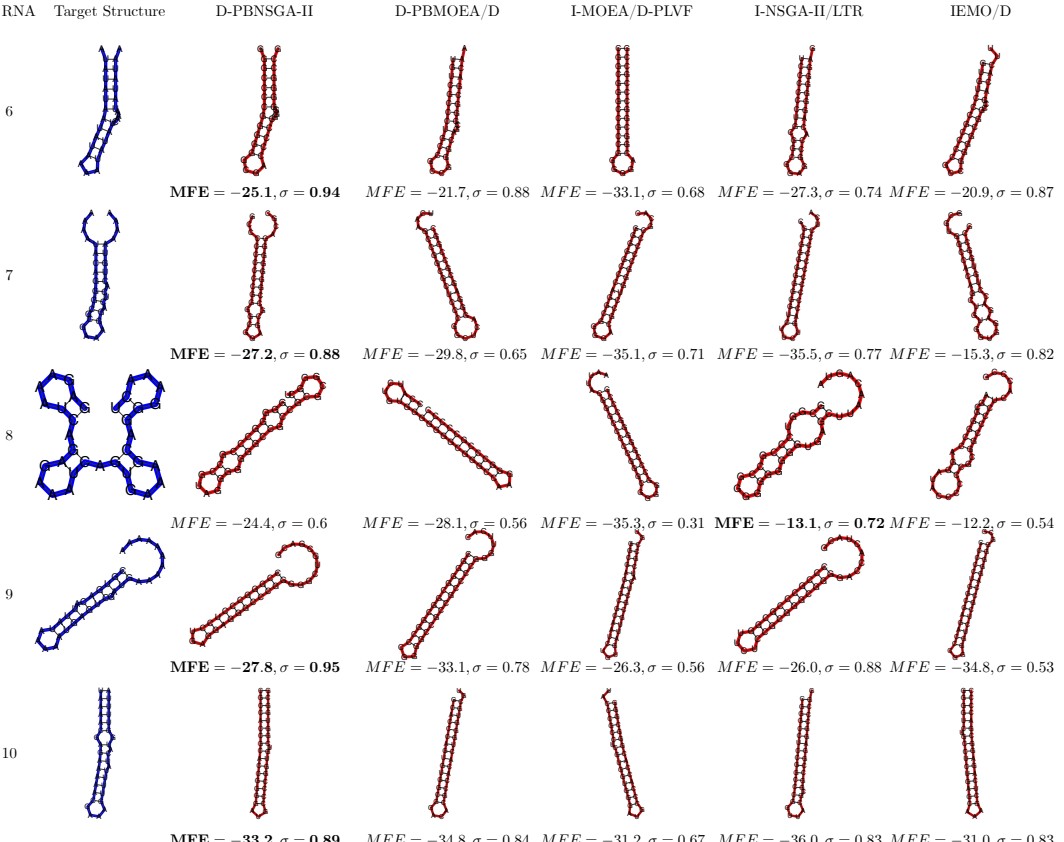

**Figure A15:** The comparison results of `D-PBEMO` against the other three state-of-the-art PBEMO algorithms on inverse RNA design problems. In particular, the target structure is a sample of possible solution represented in blue color while the predicted one obtained by different optimization algorithms are highlighted in red color. In this graph, the reference point is set as $\sigma = 1$. The closer $\sigma$ is to 1, the better performance achieved by the corresponding algorithm. When the $\sigma$ share the same biggest value, the smaller $MFE$ the better the performance is.

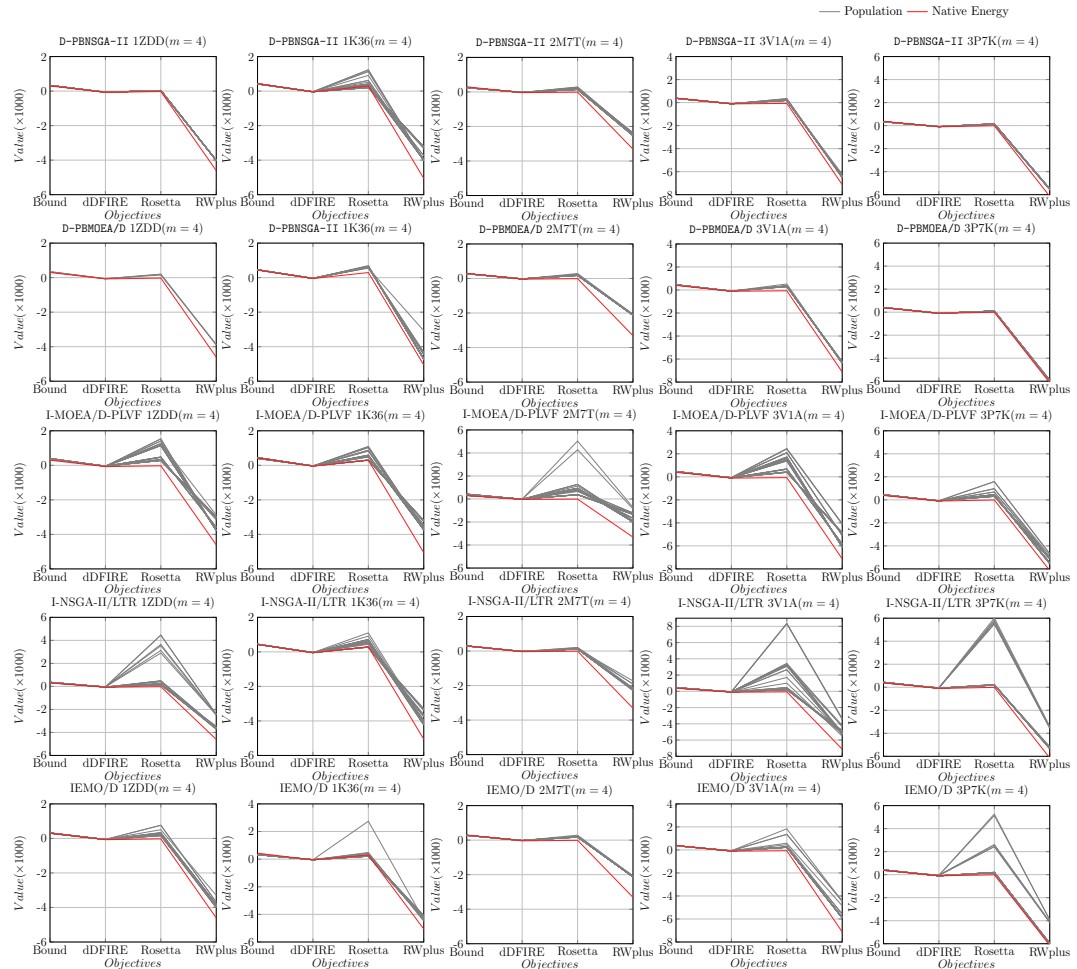

**Figure A16:** The population distribution of D-PBEMO and peer algorithms running on PSP problems ($m = 4$).

