# OpenReview forum: "Direct Preference-Based Evolutionary Multi-Objective Optimization with Dueling Bandits"
_NeurIPS.cc/2024/Conference — NeurIPS 2024 poster_

### Official Review · Reviewer_dDig · 2024-07-09

**Soundness:** 3
**Presentation:** 4
**Contribution:** 3
**Rating:** 6
**Confidence:** 4

**Summary:**

This article focuses on the preference-based evolutionary multiobjective optimization. First of all, such problems are widely found in real engineering application scenarios, thus becoming one of the most popular research directions in the field of multi-objective optimization.

Overall, in terms of the presentation of the article, this article is well-structured and clearly expressed. From the aspect of methodological design, this article is somewhat innovative. However, from the aspect of experimental design, this article seems to have some defects.

**Strengths:**

The designed method aims to address the issues exists in both consultation and preference elicitation modules.

To be specifically, the authors design a clustering-based stochastic dueling bandits algorithm to overcome the problem of convergence due to a large amount of preference comparisons. Additionally, the authors applies the Gaussian mixture distribution to leverage the preference learned from the consultation session.

**Weaknesses:**

1-In Section 4.1, I am curious as to what criteria the authors used to select the benchmark problems. For example, why didn't the authors choose the same series of benchmark problems as DTLZ7, WFG2, WFG4, etc.?
2-In Section 4.2, I am concerned that the comparison algorithm used in the experiment is SOTA in the domain.
3-Two variants, DPNSGA-II and DP MOEA/D, were designed in the article. However, the discussion between these two variants seems to be a bit thin. What are their respective strengths and weaknesses?
4-Similarly, I found the discussion in the experimental section to be overly simplistic, seeming to restate the results of the experiment at face value. But it doesn't analyze the reasons behind the results too much.
5-Population size, number of search iterations, and other such parameter settings do not seem to be given by the authors.

**Questions:**

1-In Section 4.1, I am curious as to what criteria the authors used to select the benchmark problems. For example, why didn't the authors choose the same series of benchmark problems as DTLZ7, WFG2, WFG4, etc.?
2-In Section 4.2, I am concerned that the comparison algorithm used in the experiment is SOTA in the domain.
3-Two variants, DPNSGA-II and DP MOEA/D, were designed in the article. However, the discussion between these two variants seems to be a bit thin. What are their respective strengths and weaknesses?
4-Similarly, I found the discussion in the experimental section to be overly simplistic, seeming to restate the results of the experiment at face value. But it doesn't analyze the reasons behind the results too much.
5-Population size, number of search iterations, and other such parameter settings do not seem to be given by the authors.

**Limitations:**

The experimental chapter needs to be further upgraded.

---

> ### Author Rebuttal · Authors · 2024-08-07
>
> **Response to weaknesses and questions**
>
> We address the reviewer’s concerns one by one as follows.
>
> 1. The choice of benchmark test problems follow the existing literature [1] and it also adheres to the criteria outlined in [2,3]. This ensures that the chosen benchmark test problems have different Pareto-optimal front (PF) shapes and various characteristics (e.g., multiple local optima and concave/convex PF, etc) while we omitted some benchmark problems having duplicated properties. In addition, we included more real-world problems (protein structure prediction and RNA inverse design) to promote AI4Science.
>
>     We do not consider DTLZ7 and WFG2 because they are with disconnected PF segments. As discussed in [1], it is difficult for decision-makers to elicit their preferences in the disconnected regions, which also represent infeasible parts. Further, since WFG4 shares the same characteristics with WFG7 (i.e., separable, unimodal and with a concave PF), we consider WFG7 in our experiments.
>
> 2. In particular, according to the experimental results reported in a recent paper [4], we chose three most competitive algorithms (i.e., I-MOEA/D-PLVF, I-NSGA-II/LTR, and IEMO/D) as our peer algorithms. We believe the chosen algorithms represent the current state-of-the-art in preference-based evolutionary multi-objective optimization (EMO). Further, to demonstrate the flexibility of our framework, we chose a preferential Bayesian optimization [5], which was designed only for single-objective optimization problems, as a baseline and adapt it to our D-PBEMO framework by replacing its preference learning part by our consultation module. As the results shown in Section 4.3, this variant also works well for tackling multi-objective optimization problems.
>
> 3. Because our proposed D-PBEMO framework is algorithm-agnostic. That is to say any existing EMO algorithm can be used in the optimization module with minor adaptation. For proof-of-concept purpose, we choose NSGA-II and MOEA/D, two influential EMO algorithms in the literature, as two instances. As reported in the EMO literature [6], NSGA-II performs better on 2-objective problems, while MOEA/D scales well to many-objective problems ($m\ge3$). This characteristic is retained in our D-PBNSGA-II and D-PBMOEA/D algorithms.
>
> 4. We apologize for the succinct discussion on the experimental results. This is partially because our proposed D-PBNSGA-II and D-PBMOEA/D have shown constantly superior performance in almost all comparisons, as well as the strict page limit. We promise to strengthen this part in the camera-ready version. Specifically, we plan to focus on discussing the different characteristics of D-PBNSGA-II and D-PBMOEA/D for handling low- and high-dimensional problems, respectively. These characteristics are related to the internal mechanisms of of NSGA-II and MOEA/D.
>
>     i) NSGA-II is a representative algorithm based on Pareto dominance in its environmental selection. Since its diversity maintenance strategy mainly depends on the distances among solutions, it is relatively robust and fast to find a reasonably good Pareto-optimal front (PF) approximation. However, because NSGA-II relies on Pareto dominance, it does not scale well to many-objective problems.
>
>     ii) In contrast, the search of MOEA/D is mainly determined by the weight vectors. It is naturally more scalable to many-objective cases.
>
>     The above properties are all reflected in the performance of D-PBNSGA-II and D-PBMOEA/D.
>
> 5. All experiment parameter settings, including population size, number of search iterations, and other parameter settings, are provided in our Appendix D.2.
>
> [1] Li, Ke, et al. "Does preference always help? A holistic study on preference-based evolutionary multiobjective optimization using reference points." *IEEE Transactions on Evolutionary Computation* 24(6): 1078-1096, 2020.
>
> [2] Huband, P. et al. "A review of multiobjective test problems and a scalable test problem toolkit," in IEEE Transactions on Evolutionary Computation, 10(5):477-506, 2006, doi: 10.1109/TEVC.2005.861417.
>
> [3] Zapotecas-Martínez, Saúl, et al. "A review of features and limitations of existing scalable multiobjective test suites." *IEEE Transactions on Evolutionary Computation* 23(1): 130-142, 2018.
>
> [4] Li, Ke, et al. "Interactive evolutionary multiobjective optimization via learning to rank." *IEEE Transactions on Evolutionary Computation* 27(4): 749-763, 2023.
>
> [5] J. González, et al. "Preferential bayesian optimization."  In ICML’17: Proc. of the 34th international conference on Machine learning, volume 70, pages 1282–1291. PMLR, 2017.
>
> [6] Zhang, Qingfu, and Hui Li. "MOEA/D: A multiobjective evolutionary algorithm based on decomposition." *IEEE Transactions on evolutionary computation* 11(6): 712-731, 2007.

---

> ### Comment · Reviewer_dDig · 2024-08-08
>
> I think the author did a great job of responding to all of my comments. From my personal point of view, I have no other problems.
>
> Of course, I wish the author had placed some of the replies to my comments in the final version, as it would have prevented others from having similar doubts.
>
> Based on the author's and other reviewers' comments, I think this article is of good quality and I would like to increase my rating for this article from 6 (Weak Accept) to 7 (Accept).

---

> > ### Author Response · Authors · 2024-08-08
> > **Response to the Reviewer dDig**
> >
> > We sincerely appreciate the reviewer's positive feedback on our efforts and your kind help to elevate your rating. Meanwhile, we for sure will carefully revise our final version to make sure it stands at the highest quality possible.
> >
> > Last but not the least, we would like to take this opportunity to thank yours and the other reviewers' constructive suggestions on our work. All of them give us insights about how to improve the quality of our work, as well as how to push this line of research forward.

---

> > ### Author Response · Authors · 2024-08-13
> > **Response to Official Comment by Reviewer dDig**
> >
> > Dear Reviewer dDig,
> >
> > Thank you very much again for your positive feedback and confirmation on our effort. We would like to respectively ask whether you would like to raise your score from 6 (Weak Accept) to 7 (Accept) as mentioned at the end of your comment? Really sorry for chasing this, because the deadline is approaching.
> >
> > Thank you very much again.
> >
> > Best regards
> > Authors

---

> > > ### Comment · Reviewer_dDig · 2024-08-14
> > > **Raising my score from 6 (Weak Accept) to 7 (Accept)**
> > >
> > > After evaluating the comments of the other reviewers and the authors' responses, I determined to raise the score from 6 to 7.

---

> > > > ### Author Response · Authors · 2024-08-14
> > > > **Raising my score from 6 (Weak Accept) to 7 (Accept)**
> > > >
> > > > Thank you very much for confirming our effort!
> > > >
> > > > Best regards,
> > > > Authors

---

### Official Review · Reviewer_FUFf · 2024-07-10

**Soundness:** 2
**Presentation:** 1
**Contribution:** 2
**Rating:** 5
**Confidence:** 3

**Summary:**

This paper focused on the problem of multi-objective optimization in the dueling bandits settings.  A clustering-based stochastic dueling bandits algorithm was developed and analysis. The performance is further validated via experiments.

**Strengths:**

- A new framework via combining dueling bandits and evolutionary multi-objective optimization was presented.
- The proposed algorithm has been applied to real-world problem, i.e., protein structure prediction, which is interesting and promising.

**Weaknesses:**

- In dueling bandits literature, one of the most widely assumed winner or the most general winner is the Condorcet winner. In this paper, for example, in definition 2.1, the authors defined the Copeland winner. What is the intuition behind this or in other words, does your framework requires this specific winner, or can it be generalized to the general Condorcet winner (or other winners such as Borda Winner, Neumann Winner that are widely used in the dueling bandits literature)?
- In your algorithms and theorem, $\alpha>0.5$ is assumed. Is this a natural option in practice?
- The proposed D-PBEMO framework consists of three modules as shown in Figure 1(a), while the regret is characterized for the consultation modules. Not sure if I miss anything, can this regret be claimed as that of D-PBEMO?
- The writing of the paper can be significantly improved. It is hard to follow the paper with many mathematical notations not rigorously defined, and some details are missing.

**Questions:**

See weakness above.

**Limitations:**

N/A.

---

> ### Author Rebuttal · Authors · 2024-08-07
>
> **Response to W1: Choice of winner**
>
> We address the reviewer’s concerns from the following three aspects.
>
> 1. The Copeland winner is more universally applicable than the Condorcet winner. The Copeland winner includes the Condorcet winner. That is to say the Condorcet winner is always a Copeland winner, whereas the Copeland winner is not necessarily a Condorcet winner. In some scenarios, a Condorcet winner may even not exist, but a Copeland winner always does.
> 2. Following the first bullet point, in our multi-objective optimization scenario, a Condorcet winner may not exist. Specifically, user preferences, which are represented as reference points or golden points in our experimental settings, may not be reachable because sometimes user preferences lie beyond the PF (see some examples in Figure A7 on the ZDT3 test problem in our Appendix E.1). In such cases, the non-dominated solution set found by the evolutionary multi-objective optimization (EMO) algorithm may contain multiple solutions that are the same closest to the golden point. These solutions are optimal arms and are Copeland winners but not Condorcet winners. Because the Condorcet winner should be unique by definition, the Condorcet winner is not suitable for our multi-objective optimization context.
> 3. As for the reviewer mentioned two other winners, our justifications are as follows.
> - For the Borda winner, it belongs to the family of positional voting where the preferences are elicited as a full rank regarding all solutions. As reported in [1], such preference elicitation method can be cognitive intensive for humans. Note that preference-based EMO itself is an optimization-cum-decision-making process, which involves multiple runs of consultations with humans. Therefore, one of our design principles is to reduce the human's cognitive load as much as we can during each consultation. Given this justification, we think the Borda winner can in principle be applied in our consultation module, but it is not recommended, at least not as good as the dueling bandits using pairwise comparisons in our D-PBEMO.
> - For the Neumann winner, it is designed for scenarios involving potential clones of arms [2]. However, because each subset (i.e., arm) is different in our D-PBEMO, the preference elicitation module will suspend preference learning before subsets become clones. Further, the Neumann winner is originally designed for contextual dueling bandits, it is not directly applicable to our EMO context which is stochastic.
>
> Given the above justifications, we believe our choice of Copeland winner in our current version of D-PBEMO is rational.
>
> [1] Zintgraf, Luisa M., et al. "Ordered preference elicitation strategies for supporting multi-objective decision making." AAMAS'18: 1477-1485, (2018).
>
> [2] Dudík, Miroslav, et al. "Contextual dueling bandits." *Conference on Learning Theory*. PMLR, 2015.
>
> **Response to W2: $\alpha$ setting**
>
> We confirm to the reviewer that $\alpha > 0.5$ is a natural option in practice. The requirement $\alpha > 0.5$ in the double Thompson sampling proof can be traced back to the RUCB paper [3]. This parameter range ensures the exploration capability in Thompson sampling or UCB processes, which is a fundamental assumption to guarantee the algorithm’s convergence.
>
> [3] Zoghi, Masrour, et al. "Relative upper confidence bound for the k-armed dueling bandit problem." *ICML'14*: 10-18, (2014).
>
> **Response to W3: Regret analysis**
>
> We confirm that the reviewer understood well. The regret analysis is for the consultation module, i.e., the clustering-based stochastic dueling bandits algorithm used for preference learning. We justify this further from the following three aspects.
>
> 1. The regret represents the uncertainty in the consultation module. In this paper, we are not yet to provide an uncertainty quantification for the D-PBEMO algorithm. This is mainly attributed to the stochastic nature of the reproduction operators (i.e., crossover and mutation) which introduce additional uncertainty difficult to quantify.
> 2. Further, we believe our regret analysis for the preference learning in the context of preference-based EMO is an original contribution, which has not yet been addressed so far to the best of our knowledge.
> 3. Note that our proposed D-PBEMO framework is algorithm agnostic. That is to say any EMO algorithm can be applied with minor modification in our optimization module. In this paper, we applied two most influential EMO algorithms as the baseline for a proof-of-concept purpose. Because the convergence property highly depends on the algorithmic behavior of the underlying EMO algorithm (partially justified in our first bullet point), it is hard to provide a universal convergence analysis applied for all EMO algorithms. Yet, we also believe this is part of our future endeavours.
>
> **Response to W4: Writing of the paper**
>
> We apologize for our presentation when tackling many mathematical notions. In the camera-ready version, we will take two actions to improve the readability of this paper.
>
> 1. Carefully double check all mathematical notions used in this paper and prepare a lookup table at the beginning of the Appendix.
> 2. We will also augment some important definitions and preliminary knowledge statement about dueling bandits to be more self-contained.

---

> > ### Comment · Reviewer_FUFf · 2024-08-10
> > **Thank you for the rebuttal**
> >
> > Thank you for the clarifications. I will increase the rating.

---

> > > ### Author Response · Authors · 2024-08-10
> > > **Response to the Reviewer FUFf**
> > >
> > > Thank you very much for confirming our effort. We also sincerely appreciate your constructive suggestions on improving the quality of our work.

---

### Official Review · Reviewer_hDtX · 2024-07-11

**Soundness:** 2
**Presentation:** 2
**Contribution:** 2
**Rating:** 5
**Confidence:** 2

**Summary:**

Preference-based evolutionary multi-objective optimization (PBEMO) methods involve optimization (explore the space), consultation (learn human preference), and elicitation (guide evolutionary search). Existing PBEMO methods may suffer from inaccurate reward models, which is likely to happen given that human feedback exhibit a lot of randomness. The authors propose to directly leverage human feedback without using a reward model to guide the evolutionary search. Specifically, given that human feedback from relative comparison is better than absolute labels, the authors employ dueling bandits to compute preference metrics.

**Strengths:**

The authors prove the regret bound of proposed algorithm, which is better than that of Thompson sampling based dueling bandit algorithms. Empirical evaluations on 33 settings showcase the effectiveness of proposed approach.

**Weaknesses:**

In step 1 of consultation module, one needs to choose K, which is the number of subsets to partition S into. The choice of K would greatly affect how close the solutions are within a subset. Thus, it is important to discuss methods to choose K and provide justifications.

The runtimes are not reported.

**Questions:**

Could the authors provide some insights on the advantage of starting with a coarse-grained representation, which may yield an initially inaccurate SOI, compared to having a set of Pareto-optimal candidate solution upfront?

**Limitations:**

The limitations are not clear. The scope of the claims should be discussed. I encourage the authors to create a separate limitations section (see NeurIPS paper checklist).

---

> ### Author Rebuttal · Authors · 2024-08-07
>
> **Response to W1: Choice of $K$**
>
> We agree with the reviewer that $K$ can impact the crowdedness of solutions within a subset. In particular, the larger the $K$ is, the smaller the distances between solutions within each subset are. As for the choice  and sensitivity of $K$, our justifications are presented in the **Author Rebuttal** attached with a PDF containing additional experiment results.
>
> **Response to W2: Runtimes**
>
> Because we are not sure whether the *runtimes* referred by the reviewer is the CPU wall clock time (actual execution time) or the run time analysis used to analyze the convergence of an evolutionary algorithm, we provide discussions on both aspects.
>
> 1. In terms of CPU wall clock time, our D-PBEMO algorithms are the fastest one in the experiments. However, we are concerned about whether it is a fair comparison if we report the CPU wall clock time results. This is mainly attributed to the programming languages used to implement different algorithms. In particular, our clustering-based stochastic dueling bandits algorithm is implemented in C++ and it runs super fast (usually within $2$  seconds). In contrast, the other peer algorithms are mainly implemented in Python. Therefor, they are often much slower than ours. Further, algorithm like I-NSGA-II/LTR [1] involves neural network training that can even make itself very slow (often over $10$ minutes).
>
> 2. As for the run time analysis, while there have been some attempts on the evolutionary multi-objective optimization (EMO) algorithms (e.g., [2]), it is far from mature compared to the rich literature for single-objective optimization, not to mention the preference-based EMO. Further, the run time analysis highly depends on the baseline algorithm while our D-PBEMO framework is EMO algorithm agnostic. In this paper, our key theoretical contribution is the regret bound for the preference learning part (i.e., the consultation module) when considering multiple conflicting objectives. We believe this contribution is original and will be valuable to multiple communities including but not limited to preference learning, multi-objective decision-making, and EMO. As part of our future works, we plan to further analyze the complexity for EMO when using the preference learned from our consultation module.
>
> [1] Li, Ke, et al. "Interactive evolutionary multiobjective optimization via learning to rank." *IEEE Transactions on Evolutionary Computation* 27(4): 749-763, 2023.
>
> [2] Bian, Chao, et al. "A General Approach to Running Time Analysis of Multi-objective Evolutionary Algorithms." *IJCAI'18*: 1405-1411, (2018).
>
> **Response to Q1: Advantage of starting with coarse-grained representation**
>
> We respectively check whether the reviewer is concerned about the statement in our *Remark 1*. If this is the case, we address the reviewer’s question from the following three aspects.
>
> 1. In EMO, solutions at the early stages of the evolutionary search are often far from the Pareto-optimal front (PF). Further, the search direction of EMO is not deterministic. Instead, given the diversity and spread requirements of an evolutionary population, the search direction is rather stochastic in the early stages. Therefore, it can be misleading and noisy if we ask decision-makers to elicit their preferences regarding such solutions.
> 2. On the other hand, we do not intend to wait until the end of EMO, i.e., when obtaining a set of solutions well approximate the PF (this also corresponds to the reviewer mentioned "$\cdots$ *a set of Pareto-optimal candidate solution upfront*"). This represents the posteriori decision-making whose drawbacks are discussed in Appendix A.1. In particular, it is hard to guarantee that we can have the solution(s) meet the decision-makers’ preference upfront given the dense characteristics of PF. Therefore, it is a paradox if the underlying EMO algorithm ends up with no solutions lying in the region of interest. This is not uncommon since the range of PF can be too huge to have a full coverage when having many objectives. Differently, since preference-based EMO is an optimization-cum-decision-making process, it is designed to search for the solution of interest interactively, rather than approximating the whole PF.
> 3. Our strategy in D-PBEMO is to start consulting decision-makers neither too early nor too late as justified in the above two bullet points. This is because: 1) such coarse-grained PF sufficiently represents the PF shape, and 2) the search direction of an evolutionary population is largely determined. In this case, we believe decision-makers can already elicit meaningful preferences regarding such solutions. This hypothesis is also empirically validated through our experiments, i.e., our proposed D-PBEMO instances outperform the other peer algorithms. As some experiments in [3, 4], it is suggested to elicit decision-makers' preferences in the later half of an EMO process. Here, to make our comparison fair, we heuristically set the timing for consulting decision-makers as the middle of EMO for all peer algorithms.
>
> [3] Lai, Guiyu, et al. "Empirical studies on the role of the decision maker in interactive evolutionary multi-objective optimization." *CEC'21*: 185-192, (2021).
>
> [4] Marquis, Jon, et al. "Impact of number of interactions, different interaction patterns, and human inconsistencies on some hybrid evolutionary multiobjective optimization algorithms." *Decision Sciences* 46(5): 981-1006, 2015.

---

> > ### Author Response · Authors · 2024-08-10
> > **Additional justification of limitations**
> >
> > Dear **Reviewer hDtX**,
> >
> > We just realized that our response to your **Limitations** concern was missing. Please find our response as follows.
> >
> > ```
> > Due to the space limitations, we only briefly discussed the limitations in the conclusion section of our current manuscript. In the camera-ready version, we will add a dedicated section to discuss the limitations. We list some examples as follows.
> >
> > 1. The regret analysis of our proposed clustering-based stochastic dueling bandits is for the optimal subset, i.e., the region of interest on the PF. It is not yet directly applicable to identify the exact optimal solution of interest. As part of our future work, we will work on efficient algorithms and theoretical study on the best arm identification in the context the preference-based EMO.
> > 2. This paper only analyzes the regret of the consultation module. How to further analyze the convergence of the D-PBEMO as a whole remains unknown. This will also lead to the next step of our research. In particular, if it is successful, we may provide a radically new perspective to analyze the convergence of evolutionary multi-objective optimization algorithms.
> > ```
> > Hope this can address your concern at this point. If you have any concerns and questions, we are more than happy to have further discussion. Thank you very much for your efforts and help.

---

> > > ### Comment · Reviewer_hDtX · 2024-08-13
> > >
> > > I thank the authors for their response. The authors addressed my concerns on the choice of K, runtime, and limitations. I increased my score.

---

> > > > ### Author Response · Authors · 2024-08-13
> > > > **Response to Official Comment by Reviewer hDtX**
> > > >
> > > > Thank you very much for confirming our efforts. Anyway, if you have any other concerns or questions, we are happy to engage in further discussions :-)

---

### Official Review · Reviewer_PNDi · 2024-07-13

**Soundness:** 3
**Presentation:** 2
**Contribution:** 3
**Rating:** 6
**Confidence:** 1

**Summary:**

One challenge and potential advantage of multi-objective optimization (MO) is to adapt the dynamic human preferences while outputting an optimum. Although Preference-based evolutionary MO (PBEMO) is a promising framework, current approaches are inefficient and may not interpret the decision makers' true intentions accurately. One reason is that the decision makers' true intentions were not precisely "expressed" and "acknowledged" by the predefined reward model. This paper, intending to solve this reason, designs a framework that directly leverages human feedback, using a clustering-based stochastic dueling bandits algorithm.

**Strengths:**

1. This paper tackles an interesting and important problem, and the method is novel. In particular, it is certainly valuable to directly express the human feedback into the framework, as human feedback is the critical information for preferences outputs.

2. The application of multiple arms and Copeland winner as the main decision-making criteria indeed makes sense to me. Although different, it resembles the rank computation in many RLHF schemes to some extent.

**Weaknesses:**

Due to my very limited knowledge in this area, I might not be able to find valuable weaknesses in this paper.

**Questions:**

In Section 3.2 "Preference Elicitation Module", why is Equation (4) the resulted mixture distribution? If it was computed, were there any justifications that this shall be the result? More explanations on this may be appreciated because, to the best of my understanding, this part plays a critical role in justifying the framework since it directly works on the preferences.

**Limitations:**

Yes.

---

> ### Author Rebuttal · Authors · 2024-08-07
>
> **Response to Q1: Is the Gaussian mixture model computed**
>
> Thank you for this question. First of all, the Gaussian mixture model is not computed. Instead, it is a model assumption which we contend to be reasonable to represent user preference distribution within the solution of interest (SOI) region. We justify this from the following three aspects.
>
> 1. A key assumption of using the Gaussian mixture model is that user preference distribution within the SOI region follows a Gaussian distribution. However, we do not enforce such Gaussian distribution assumption to the other areas out side of the SOI region.
> 2. Theoretically, a Gaussian mixture model can simulate any distribution. Therefore, using a Gaussian mixture model to simulate user preference distributions is more general compared to other methods for expressing preferences, such as Gaussian process for ranking used in Bayesian optimization [1].
> 3. In addition, Gaussian mixture model is potent to be used to derive the uncertainty of the preference elicitation model.
>
> [1] Zintgraf, Luisa M., et al. "Ordered preference elicitation strategies for supporting multi-objective decision making." *AAMAS'18*: 1477-1485, (2018).

---

> > ### Comment · Reviewer_PNDi · 2024-08-13
> > **Thanks for the response**
> >
> > I appreciate the concise response by the authors. However, due to my very limited knowledge in this area, I will keep my score and confidence level.

---

> > > ### Author Response · Authors · 2024-08-13
> > > **Response to "Thanks for the response"**
> > >
> > > We appreciate the reviewer's support in this work.

---

### Author Rebuttal · Authors · 2024-08-07

# Response to reviewer hDtX about the choice of $K$
We agree with the reviewer hDtX that $K$ can impact the crowdedness of solutions within a subset. In particular, the larger the $K$ is, the smaller the distances between solutions within each subset are. As for the choice of $K$, we justify this from two perspectives:

1. As shown in Theorem 3.3, $K$ is involved in the regret of our clustering-based stochastic dueling bandits algorithm. A larger $K$ results in a larger uncertainty in preference learning for the same number of comparisons, leading to a slower convergence rate. However, it also narrows the final region of interest (ROI). There is no definitive guideline for selecting $K$ for different problems. Users can adjust $K$ to an appropriate value based on the accuracy of preference learning.
2. Further, we have conducted a sensitivity study on $K$ for $2$-objective problems in Section 4.4. In particular, we set the population size as $100$ and $K\in\{2,5,10\}$. From the results therein we can see that our proposed D-PBEMO framework is not sensitive to the choice of $K$. In addition, we have conducted additional experiments on problems with more objectives. The results can be found in the attached PDF file.
As the dimensionality increases, the population size will increase. Generally, with larger populations, a higher $K$ tends to yield better results, aligning with our intuition. Furthermore, our significance analysis across 20 repeated experiments reveals that the optimal $K$ does not show significant differences in performance.
In summary, $K$ does not significantly impact the performance of our proposed D-PBEMO framework. For most problems, we do not recommend choosing a very small/large $K$ (e.g., $K=2$, $K=N$), as it may inefficiently narrow down the ROI.

---

### Author Response · Authors · 2024-08-12
**We are keen on engaging in further discussions**

Dear reviewers,

Thank you very much for your time and efforts on engaging in this discussion. Meanwhile, we sincerely appreciate those reviewers who confirm our contributions and response, as well as elevating their scores.

If you have any further concerns, please feel free to let us know. We are keen on engaging in further discussions.

Best regards,

Authors

---

### Decision · Program_Chairs · 2024-09-25

**Decision:**

Accept (poster)

**Comment:**

This paper focuses on the application of dueling bandits to evolutionary multi-objective optimization. In particular, reference-based evolutionary multi-objective optimization (PBEMO) methods involve optimization (explore the space), consultation (learn human preference), and elicitation (guide evolutionary search). Existing PBEMO methods may suffer from inaccurate reward models, which is likely to happen given that human feedback exhibit a lot of randomness. The authors propose to directly leverage human feedback without using a reward model to guide the evolutionary search. Specifically, given that human feedback from relative comparison is better than absolute labels, the authors employ dueling bandits to compute preference metrics and to choose the winner using the Copeland score.

In summary the authors have done a great job explaining the unclear parts during the rebuttal phase, convincing several reviewers to raise their final scores. While the paper does not contribute that much to the core ML literature, the application of dueling bandits to the multi-objecive optimisation domain is significant.